# Functional synapses between neurons and small cell lung cancer

Small cell lung cancer (SCLC) is a highly aggressive type of lung cancer, characterized by rapid proliferation, early metastatic spread, frequent early relapse and a high mortality rate[1–3]. Recent evidence has suggested that innervation has an important role in the development and progression of several types of cancer[4,5]. Cancer-to-neuron synapses have been reported in gliomas[6,7], but whether peripheral tumours can form such structures is unknown. Here we show that SCLC cells can form functional synapses and receive synaptic transmission. Using in vivo insertional mutagenesis screening in conjunction with cross-species genomic and transcriptomic validation, we identified neuronal, synaptic and glutamatergic signalling gene sets in mouse and human SCLC. Further experiments revealed the ability of SCLC cells to form synaptic structures with neurons in vitro and in vivo. Electrophysiology and optogenetic experiments confirmed that cancer cells can receive NMDA receptor- and GABA$_A$ receptor-mediated synaptic inputs. Fitting with a potential oncogenic role of neuron–SCLC interactions, we showed that SCLC cells derive a proliferation advantage when co-cultured with vagal sensory or cortical neurons. Moreover, inhibition of glutamate signalling had therapeutic efficacy in an autochthonous mouse model of SCLC. Therefore, following malignant transformation, SCLC cells seem to hijack synaptic signalling to promote tumour growth, thereby exposing a new route for therapeutic intervention.

SCLC constitutes approximately 15% of all lung cancer cases[1–3]. Frontline treatment, consisting of cisplatin, etoposide and immune-checkpoint blockade plus optional prophylactic cranial irradiation, induces response rates of greater than 60%[2,3]. However, these responses are largely transient, resulting in a median overall survival of around 12 months[1,2].

SCLC is characterized by nearly universal biallelic loss of *TP53* and *RB1* (refs. 8,9). Several studies have shown that pulmonary neuroendocrine cells (PNECs) are a permissive cell type of origin for SCLC, but other cell types can also give rise to SCLC in mice following *Trp53* and *Rb1* loss, especially when *Myc* is concomitantly overexpressed[10–13]. These non-neuroendocrine lung epithelial cells acquire a PNEC-like phenotype and express neuroendocrine markers[11]. PNECs develop from lung epithelial progenitors of endodermal lineage and are innervated by different types of nerve fibres originating from the nodose, jugular and dorsal root ganglia[14–17].

Three molecular subtypes of SCLC, driven by the transcription factors ASCL1 (SCLC-A), NEUROD1 (SCLC-N) or POU2F3 (SCLC-P), have been described. A fourth subtype is variably described as inflamed (SCLC-I) or YAP1 expressing (SCLC-Y)[18–20]. SCLC-A and SCLC-N contain electrically active cells that can fire action potentials[21], whereas SCLC-P and SCLC-Y show lower neuroendocrine differentiation[18].

Recent evidence has suggested that innervation impacts tumour initiation and plasticity[4,5,22,23]. For instance, glutamate spillover from the synaptic cleft of neuron-to-neuron synapses was reported to stimulate breast cancer cells located in a perisynaptic position[24]. Direct synaptic contacts between presynaptic neurons and postsynaptic glioma cells were also reported to increase proliferation and invasion[5–7]. By contrast, no bona fide synapses have thus far been described between neurons and cancers that arise outside the central nervous system.

## Synaptic genes influence mouse SCLC

To search for genes and pathways that contribute to SCLC tumorigenesis in vivo, we performed a *piggyBac* insertional mutagenesis screen in the *Rb1$^{fl/fl}$Trp53$^{fl/fl}$* (RP) SCLC model[25]. Expression of the *piggyBac* transposase in *Rosa26$^{LSL.PB}$* (L) mice[26] is prevented by a *loxP*-STOP-*loxP* cassette (LSL) (Extended Data Fig. 1a,b). We crossed RPL mice with *ATP1-S2* (S) mice (carrying 20 transposon copies on chromosome 10) or *ATP-H39* (H) mice (carrying 80 transposon copies on chromosome 5). Depending on the integration site, the transposons can intercept and block transcription or activate expression of different isoforms through the CAG promoter[26]. SCLC was induced by intratracheal instillation with Ad-CMV-Cre adenovirus[25,27] (Extended Data Fig. 1c,d). We sequenced genomic DNA from 106 tumours derived from 14 untreated mice, 117 tumours from 24 mice treated with cisplatin and etoposide, and 90 tumours from 20 mice treated with the anti-PD-1 antibody RMP1-14 (Extended Data Fig. 1e).

Initial examination did not reveal any gene with a significantly different number of insertions between untreated, cisplatin + etoposide-exposed and anti-PD1-treated tumours or between primary and metastatic tumours (Supplementary Tables 1–3). Therefore, all samples were pooled for subsequent analyses. The significantly transposon-targeted genes in our *piggyBac* screen were essentially distributed across the

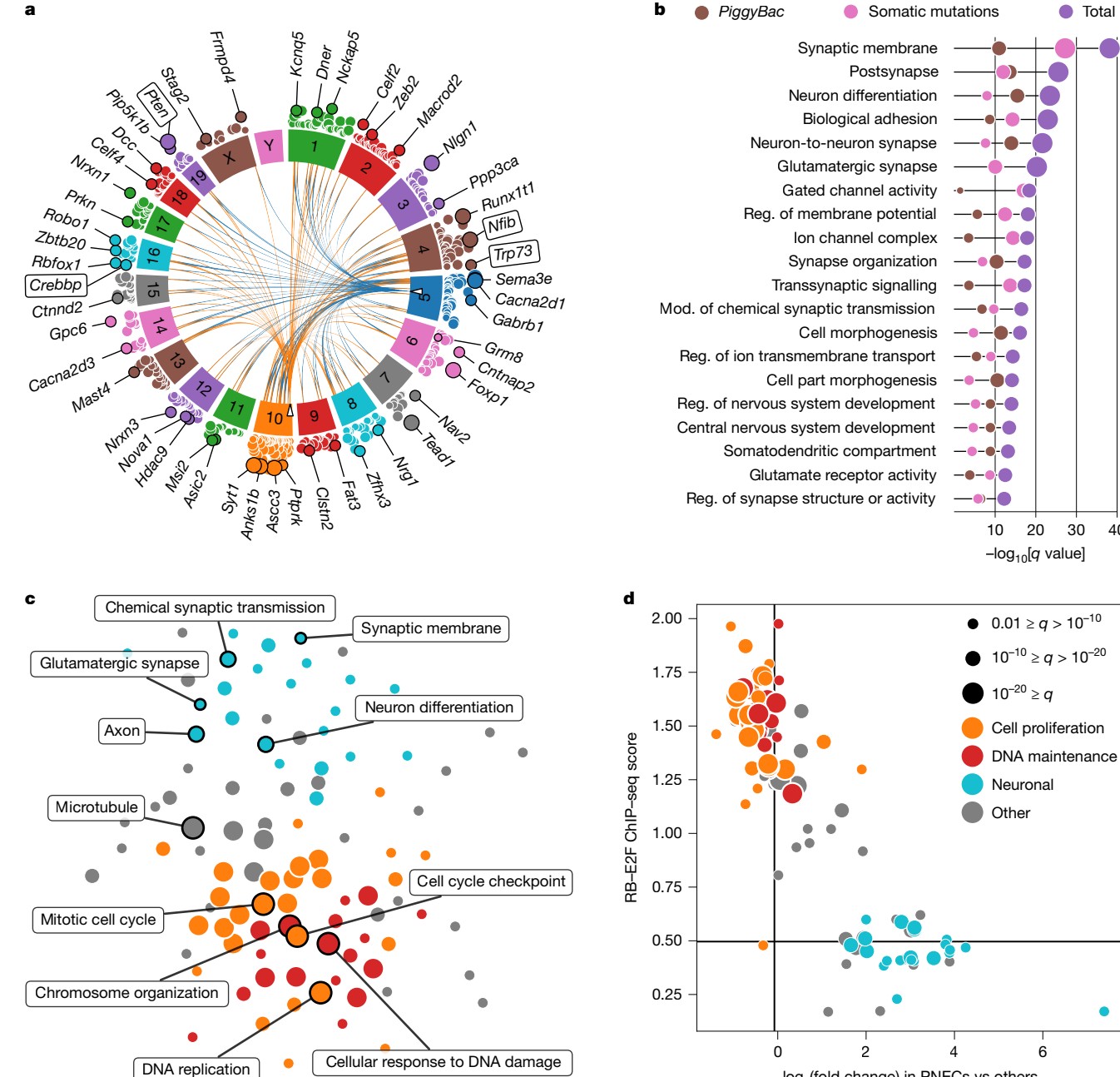

**Fig. 1 | Genome-wide analysis of SCLC across species. a**, Circos plot displaying the transposon integration pattern of an unbiased *piggyBac* insertional mutagenesis screen in 303 mouse tumours. The chord plot in the centre shows the transpositions from the donor loci (empty triangles) on chromosomes 5 and 10 to the 100 genes with the most significant enrichment in transposon insertions. The middle layer shows the chromosome labels. The scatterplot in the outer layer includes all genes with a significant enrichment in transposon insertions ($q < 0.1$, Poisson distribution with false discovery rate (FDR) correction). Selected genes are annotated, and genes previously linked to SCLC have label boxes. **b**, Top 20 most significantly enriched GO terms in the *piggyBac* dataset and human genetic data. Significance was determined by two-sided Fisher's exact test with FDR correction. Mod., modulation; reg., regulation. **c**, Force-directed graph of GO analysis, showing gene sets

enriched for genes upregulated in SCLC compared with other types of cancer from the TCGA dataset and with healthy tissue types from the GTEx dataset. Significance was determined by two-sided Fisher's exact test with FDR correction. **d**, Scatterplot of the gene sets in **c**. On the $y$ axis is the RB–E2F score, calculated using ChIP–seq data from the CISTROME database. A high score indicates strong ChIP–seq signal in experiments with antibodies against RB1, RBL2, E2F1, E2F2, E2F3, E2F4 or E2F5 near the promoter of the upregulated genes included in the gene set. On the $x$ axis is the fold change in expression on the $\log_2$ scale for PNECs versus other lung cell types in published scRNA-seq data. A high fold change indicates that the upregulated genes in the gene set are also upregulated in healthy PNECs. Significance was determined by two-sided Fisher's exact test with FDR correction.

entire genome (Fig. 1a and Supplementary Table 4), and our screen returned genes with known roles in SCLC, such as *Crebbp*, *Pten*, *Nfib* and *Trp73* (refs. 8,28–30; Extended Data Fig. 1f–i). Unexpectedly, we also identified several genes associated with the formation of synapses, such as *Nrxn1*, *Nlgn1*, *Dcc* and *Reln*[31–34] (Extended Data Fig. 1j–m).

## Synaptic genes are mutated in human SCLC

To cross-validate the hits derived from our *piggyBac* screen, we re-analysed sequencing data from 456 human SCLC samples[8,35–40]. The specimens included cell lines, primary tumours and metastases

from both chemotherapy-naive and chemotherapy-exposed patients (Extended Data Fig. 2a). These different SCLC samples were similar in their mutation profiles (Extended Data Fig. 2b–d) and had aberrations in genes with known roles in SCLC, such as *TP53*, *RB1*, *CREBBP* and *PTEN* (Extended Data Fig. 2e–h and Supplementary Table 5). We also identified a significant number of mutations in several genes that were recurrently targeted by *piggyBac* transposon integration, including *NRXN1*, *NLGN1*, *DCC* and *RELN* (Extended Data Fig. 2i–l and Supplementary Table 5). Overall, the *piggyBac* and human datasets were highly overlapping ($P = 7.2 \times 10^{-37}$, Fisher's exact test).

Notably, the rate of transposon insertions in mice and the rate of mutations in human samples showed an opposite correlation to gene expression, suggesting that these two datasets ideally complement each other (Extended Data Fig. 3a). In agreement with this notion, mutations in genes that were significantly mutated in human samples but not identified in the *piggyBac* screen were enriched for non-conserved nucleotides, whereas genes that were identified in both datasets had mutations that were depleted of non-conserved nucleotides, suggesting a functional role for the genes identified in both datasets (Extended Data Fig. 3b). We further confirmed the validity of our screen using the $Rb1^{fl/fl}Trp53^{fl/fl}Rbl2^{fl/fl}R26^{LSL-tdTomato}H11^{LSL-Cas9}$ SCLC model[41], combined with lentiviral delivery of Cre and single-guide RNAs (sgRNAs) against *Reln*, a gene identified in both datasets. Fully in line with our cross-species discovery approach, two distinct sgRNAs targeting *Reln* resulted in significantly larger tumours compared with non-targeting control sgRNAs (Extended Data Fig. 3c–g).

We next asked which Gene Ontology (GO) gene sets were significantly enriched in the human sequencing datasets and in the *piggyBac* screen (Fig. 1b, Extended Data Fig. 3h and Supplementary Tables 6 and 7). Unexpectedly, the vast majority of the enriched terms were related to neuronal phenotypes and synaptic functions, such as 'synaptic membrane', 'glutamatergic synapse', 'glutamate receptor activity', 'GABAergic synapse' and 'transsynaptic signalling'. Therefore, the only clear genetic signal we identified at the network level in 456 human and 313 mouse tumours was related to neuronal and synaptic functions.

## Expression of synaptic genes in SCLC

To probe the relevance of these synaptic genes, we analysed transcriptome data derived from tumour specimens and normal tissue. We collected raw expression data from the datasets in refs. 8,40 and re-analysed them using The Cancer Genome Atlas (TCGA) transcriptome pipeline, to identify gene sets with expression that was enriched in SCLC compared with 33 distinct cancer entities. We similarly deployed the Genotype-Tissue Expression (GTEx) pipeline to ask which gene sets were specifically enriched in SCLC transcriptomes compared with those derived from 27 healthy tissue types (Fig. 1c, Extended Data Fig. 4a–h and Supplementary Tables 8–11). Using this approach, we identified several gene sets involved in DNA replication, cell cycle checkpoint signalling, chromosome organization and the DNA damage response (Fig. 1c). Individual genes identified in the *piggyBac* and human genetic datasets, such as *NRXN1*, *NLGN1*, *DCC* and *RELN*, were highly expressed (Extended Data Fig. 4e–h). Notably, we also identified several of the same gene sets that were enriched at the genetic level, such as 'synaptic membrane', 'glutamatergic synapse', 'chemical synaptic transmission' and 'neuron differentiation', among others (Fig. 1c).

To further characterize the gene sets that are enriched in SCLC tumours, we derived an RB–E2F score, using chromatin immunoprecipitation and sequencing (ChIP–seq) data from the CISTROME database[42]. A high score indicates a strong, ChIP–seq-verified presence of RB1, RBL2, E2F1, E2F2, E2F3, E2F4 or E2F5 near the promoter of the upregulated genes included in the gene set. We also plotted gene expression profiles derived from PNECs versus other lung-resident cell populations on a $\log_2$ scale, deploying a previously published dataset[43], with a high fold change indicating that the upregulated genes in a given gene set

are specifically upregulated in PNECs. This analysis indicated that the SCLC-specific expression of neuronal and synaptic gene sets is part of the PNEC-like SCLC phenotype, whereas the high expression of genes associated with cell cycle regulation and genome maintenance seems to be largely driven by RB–E2F signalling (Fig. 1d).

To confirm that expression of the SCLC-specific gene sets was driven by cancer cells, we performed single-nucleus RNA sequencing (snRNA-seq) on six tumours collected from RP mice and re-analysed available human single-cell RNA sequencing (scRNA-seq) data[44]. In both species, the gene sets specifically enriched in cancer cells were dominated by cell proliferation and neuronal gene sets, resulting in a nearly identical pattern to our analysis of bulk RNA (Fig. 1c and Extended Data Fig. 4i–o).

Therefore, two signals are evident in human and mouse SCLC at the expression level: (1) the high expression of cell cycle gene sets downstream of the RB–E2F axis and (2) the high expression of neuronal and synaptic gene sets, which are part of the PNEC-like phenotype of SCLC cells and substantially overlap with the GO terms we identified at the genetic level.

## Neuronal processes contact SCLC cells

The observation that neuronal and synaptic gene sets constituted the strongest and most consistent signal in our *piggyBac* screen and in human SCLC prompted us to investigate a physical interaction between SCLC cells and neurons. We first asked whether neuron–cancer contacts could be detected in lung sections isolated from tumour-bearing RP mice. Interestingly, vesicular glutamate transporter 1 (VGluT1)-, P2X purinoceptor 3 (P2X3)- and growth-associated protein 43 (GAP43)-positive nerve fibres were detectable in a subset of healthy PNECs, clustered into neuroepithelial bodies (NEBs; Fig. 2a and Extended Data Fig. 5a–c), and in small SCLC tumours (Fig. 2a,b and Extended Data Fig. 5d). Conversely, larger tumours mostly lacked intralesional nerve fibres (Extended Data Fig. 5e) and, when present, GAP43- and synaptophysin (SYP)-positive fibres were observed at the tumour border (Fig. 2c). Calcitonin gene-related peptide (CGRP)-positive, substance P (SP)-positive and SYP-positive fibres were also profusely present near, but not within, tumours (Fig. 2a,b and Extended Data Fig. 5d–h). Using RP mice that additionally carried an enhanced green fluorescent protein (eGFP)-marked allele ($Rb1^{fl/fl}Trp53^{fl/fl}Rosa26^{Cas9-EGFP}$; RPC mice), we detected VGluT1-positive fibres arborizing between neuroendocrine cells from the initial stages of transformation up to the formation of small and medium-sized tumours (Fig. 2d–f). The presence of nerve fibres within small RP-derived SCLC tumours was corroborated through electron microscopy, where vesicle-enriched axon-like fibres appeared in close proximity to tumour cells (Extended Data Fig. 5i–k). We also detected nerve fibres immunoreactive for neurofilaments and SYP at the border or in the vicinity of human SCLC tumours (Extended Data Fig. 5l–p).

To assess the formation of contacts between SCLC cells and neurons in vivo, we transplanted DsRed-expressing RP tumour cells into the hippocampus of recipient *Thy1*-eGFP transgenic mice, in which excitatory neuronal subsets express eGFP. Using confocal microscopy, we found that by 10–12 days after transplantation the cancer cells located in the periphery of the tumour were profusely contacted by eGFP-positive boutons and axonal bundles (Fig. 2g–i). Most of these eGFP-positive boutons were strongly immunoreactive for the excitatory presynaptic marker VGluT1 (Fig. 2i).

We next established co-culture experiments of human SCLC cells with mouse cortical neurons. Human COR-L88 cells, of the SCLC-A subtype, were profusely contacted by VGluT1-positive neuronal processes (Extended Data Fig. 6a). Lastly, we demonstrated that these points of contact on cancer cells mostly occurred with neuronal axons marked by phosphorylated neurofilaments (anti-SMI-312 antibody) and not with dendrites immunoreactive for MAP2 (Extended Data Fig. 6b–d).

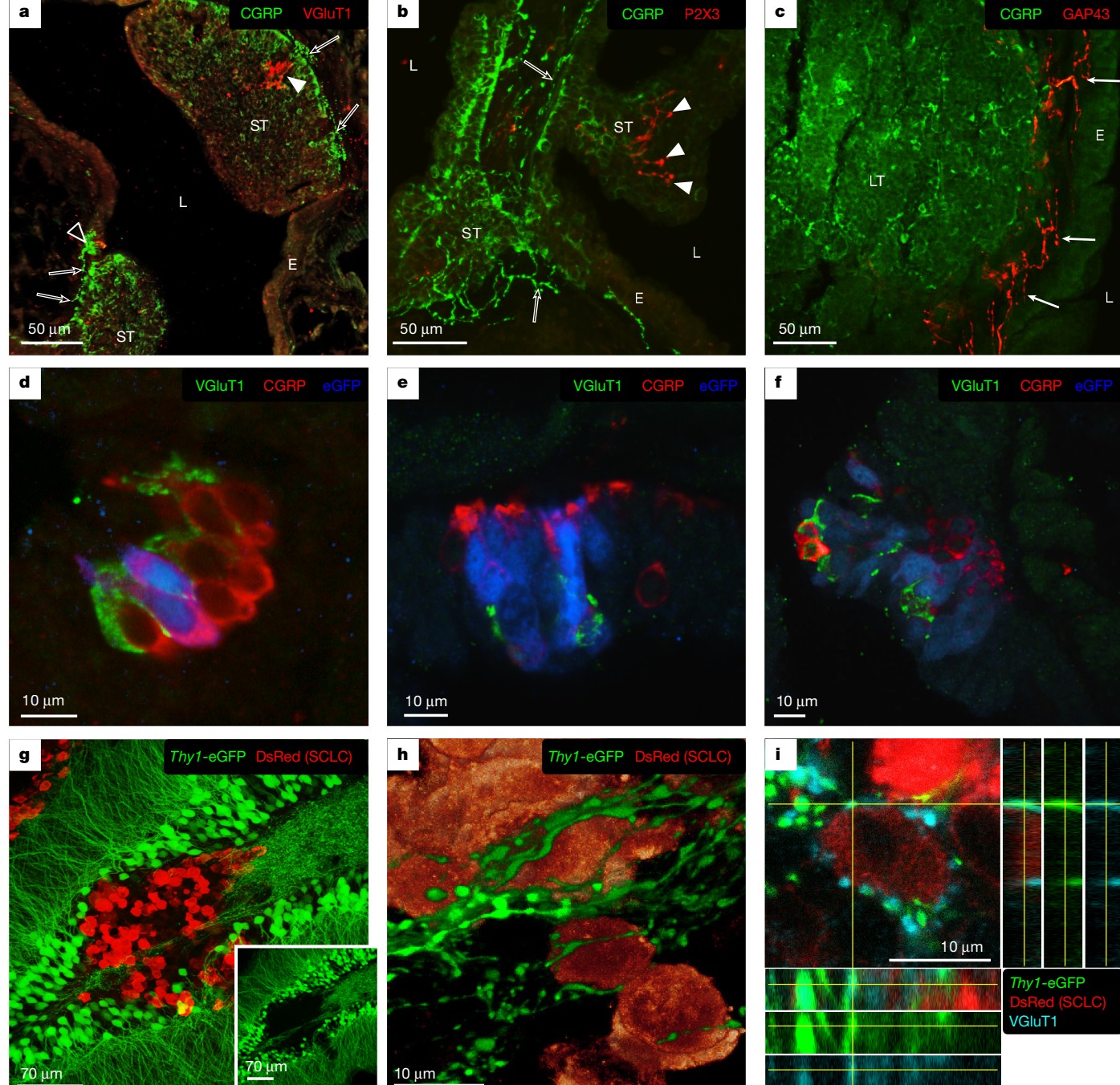

**Fig. 2 | Detection of nerve fibres in mouse SCLC tumours. a**, Confocal image of an intrapulmonary airway from an RP mouse. Two small tumours (ST) and a normal NEB (open arrowhead) are visualized with CGRP (green) and can be observed to bulge in the airway lumen (L). VGluT1-immunoreactie nerve terminals (red) are detected contacting the NEB and arborizing (arrowhead) in one of the tumours. CGRP-positive nerve fibres (open arrows) can be observed at the base of the tumours and NEB. E, epithelium. **b**, P2X3-positive nerve terminals (red, arrowheads) can be seen to arborize between the CGRP-positive (green) neuroendocrine cells of a small tumour. CGRP-positive nerve fibres (open arrows) can be observed at the base of the tumour. **c**, Immunolabelling of a CGRP-positive (green) large tumour (LT). The connective tissue between the tumour and the epithelium harbours many GAP43-positive nerve fibres (red, arrows), which do not appear to penetrate the tumour mass.

**d**, Confocal image of an NEB in an RPC mouse. Two cells are positive for eGFP (blue), indicating successful recombination and incipient transformation. VGluT1-positive fibres (green) arborize between the neuroendocrine cells (red). **e**, Initial proliferation of eGFP-positive neuroendocrine cells (blue) in an NEB. VGluT1-positive fibres (green) arborize between the transforming cells. **f**, Small SCLC tumour positive for eGFP (blue) and CGRP (red). VGluT1-positive fibres (green) arborize between the tumour cells. **g**, Immunolabelling of SCLC cells (expressing DsRed) transplanted into the hippocampus of *Thy1*-eGFP mice. The inset shows that the core of the tumour is devoid of eGFP-positive fibres. **h**, 3D reconstruction of SCLC cells located in the tumour periphery surrounded by eGFP-positive axonal varicosities. **i**, Co-localization analysis of eGFP- and VGluT1-positive boutons contacting a DsRed-expressing SCLC cell.

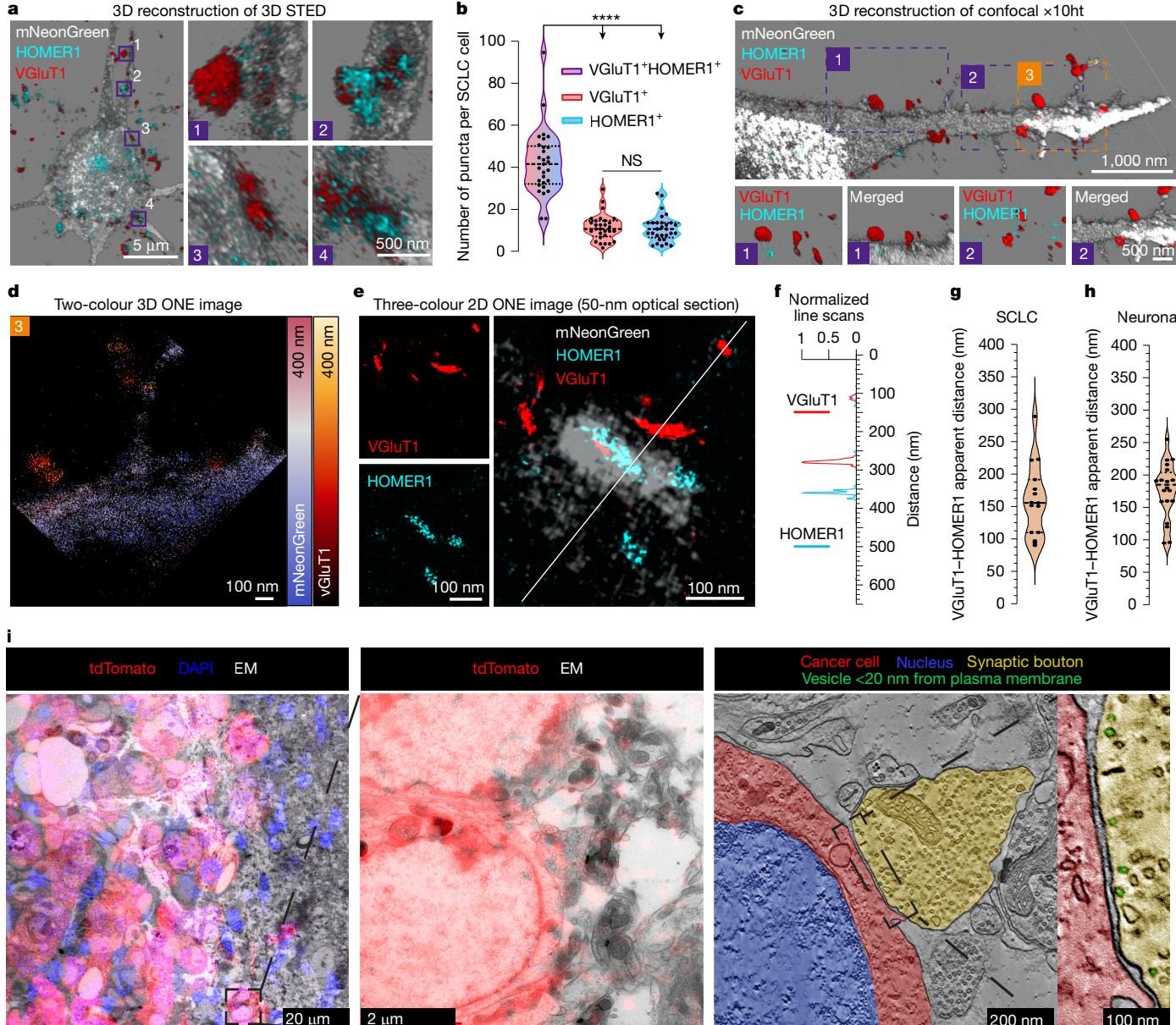

**Fig. 3 | Structural evidence for bona fide synapses in SCLC cells. a**, 3D STED images of SCLC (expressing mNeonGreen)–neuron co-cultures stained for presynaptic VGluT1 and postsynaptic HOMER1. The magnified views on the right show regions of marker co-localization. **b**, Analysis of the number of VGluT1 and HOMER1 single-positive and double-positive puncta per SCLC cell. $n = 29$ cells derived from three independent cultures and two x10ht experiments. Kruskal–Wallis one-way ANOVA test, ****$P < 0.0001$. NS, not significant. **c**, Overview of a representative 3D-reconstructed confocal image of an SCLC cell in a neuronal co-culture subjected to x10ht. Bottom panels depict magnified regions of contact between the neuron (VGluT1 positive) and SCLC cell (HOMER1 positive). **d**, Two-colour 3D ONE image of region 3 in **c**. **e**, Three-colour 2D ONE image of a representative putative synapse showing presynaptic (VGluT1-positive) and postsynaptic (HOMER1-positive) markers at points of contact between neurons and SCLC cells. **f**, Line scan of the neuron–SCLC contact in **e** showing the distance between VGluT1- and HOMER1-positive puncta. **g**, VGluT1–HOMER1 apparent distance measured in neuron–SCLC cell contacts. $n = 15$ contacts. **h**, VGluT1–HOMER1 apparent distance measured in neuron–neuron contacts. $n = 20$ contacts. **i**, CLEM of SCLC cells (expressing tdTomato) grafted into the mouse hippocampus. The left two panels depict the registered overlay between the fluorescence signal and electron microscopy (EM) image. The third panel shows the electron tomogram of an identified synaptic contact. The tomogram (single slice) depicts a presynaptic bouton (yellow pseudocolour) filled with vesicles contacting a tdTomato-positive cancer cell (red pseudocolour). Blue pseudocolour indicates the nucleus. The rightmost panel shows an enlarged view of the synaptic cleft and a pool of vesicles located within 20 nm of the plasma membrane (green pseudocolour).

These data show that SCLC cells have the ability to form contacts with neurons, both in vivo and in vitro.

## Neuron-to-cancer synapses in SCLC

To investigate the nature of these contacts, we performed confocal and stimulated emission depletion (STED) microscopy of SCLC cells

in five distinct experimental settings. First, in co-cultures of SCLC cells and cortical neurons, immunostaining for glutamatergic vesicles (anti-VGluT1) and the postsynaptic protein HOMER1 revealed co-localizing formations at the contacts between neurons and cancer cells (Fig. 3a,b). Second, we identified similar contacts in co-cultures with human induced pluripotent stem (iPS) cell-derived cortical neurons, which were characterized by expression of the presynaptic

protein Bassoon in neurons and HOMER1 in cancer cells (Extended Data Fig. 6e). Third, in co-cultures with mouse nodose ganglia, which physiologically innervate PNECs and are the most likely origin of the VGluT1-positive fibres observed in tumours in vivo[14] (Fig. 2a,d–f), we again identified juxtaposition of HOMER1 and VGluT1 on cancer cells (Extended Data Fig. 6f–i). Fourth, we detected HOMER1-positive post-synaptic structures in cancer cells in close proximity to eGFP-positive axonal boutons in brain allografts (Extended Data Fig. 7a). Lastly, we detected HOMER1–VGluT1 proximity at the interface of Cre-exposed, recombined eGFP-positive cancer cells in lung sections from autochthonous RP mice (Extended Data Fig. 7b–d).

We next conducted tenfold expansion microscopy (x10ht), reaching approximately 25-nm resolution[45]. Three-dimensional (3D) reconstruction of cortical neuron co-cultures showed a spatial organization consistent with synaptic structures, with VGluT1-positive puncta outside cancer cells juxtaposed to HOMER1 immunoreactivity in cancer cells (Fig. 3c). To visualize synapses in even greater detail, we used one-step nanoscale expansion (ONE) microscopy[46]. 3D and two-dimensional (2D) ONE images showed clear separation of the pre- and postsynaptic elements, with their localization resembling that in canonical synapses between neurons (Fig. 3d,e). Notably, the distance between the VGluT1- and HOMER1-positive puncta was comparable to that observed for neuron-to-neuron synapses in the same cultures (Fig. 3e–h).

We further characterized these synaptic contacts through electron microscopy and correlative light electron microscopy (CLEM) in brain allografts and co-cultures. Electron tomograms and 3D reconstructions of DsRed- or tdTomato-positive SCLC cells confirmed the presence of synaptic boutons filled with vesicles contacting the plasma membrane of cancer cells (Fig. 3i). Detailed examination of 280 cell perimeters located at the periphery of the allografts in ultrathin sections revealed that an average of 8.2% of the cancer cells exhibited synapses with axonal boutons (Extended Data Fig. 7e). We also identified additional ultrastructural hallmarks of stereotypical synapses, including the presence of a synaptic cleft and a pool of vesicles close to the presynaptic membrane (Fig. 3i and Extended Data Fig. 7f).

## Neuron-to-cancer neurotransmission

To assess the functionality of cancer–neuron synapses, we next conducted electrophysiological recordings of cancer cells in co-culture with cortical neurons. Although whole-cell patch-clamp recordings of COR-L88 monocultures did not show any spontaneous inputs, the same cells developed spontaneous postsynaptic currents (sPSCs) when co-cultured with neurons (Extended Data Fig. 8a,b). These currents were reduced when the co-cultures were treated with the voltage-gated sodium channel blocker tetrodotoxin (TTX), with the AMPA receptor antagonist 6-cyano-7-nitroquinoxaline-2,3-dione (CNQX), with the NMDA receptor antagonist D-2-amino-5-phosphonopentanoate (D-AP5) or with the glutamate release inhibitor riluzole[47], but not with the GABA receptor inhibitor bicuculline (Extended Data Fig. 8c). Similar to COR-L88 cells, the H524 cell line (SCLC-N) exhibited no sPSCs in monoculture (Extended Data Fig. 8d). However, we detected sPSCs in H524 cells co-cultured with cortical neurons when measuring with a holding potential of +40 mV. The majority of these currents could be inhibited with D-AP5 (Fig. 4a,b and Extended Data Fig. 8e). In two cells, a small fraction of the currents remained after D-AP5 exposure, presented a shape consistent with GABA$_A$ receptor-mediated currents and could be inhibited with the GABA$_A$ receptor blocker gabazine (Gbz; Fig. 4a,b). We also identified examples of synaptic events when measuring at 0 mV, a voltage at which mainly GABA$_A$-mediated chloride currents are observable (Extended Data Fig. 8f). Notably, optogenetic stimulation of co-cultured neurons expressing channelrhodopsin-2 (ChR2) elicited postsynaptic events in SCLC cells measured at +40 mV, which could be abolished with D-AP5, further corroborating the existence of

direct synaptic glutamatergic transmission (Fig. 4c,d). In one cell, we identified the presence of both evoked NMDA receptor- and GABA$_A$ receptor-mediated currents, further suggesting that cancer cells in co-culture are able to form functional synaptic contacts with both glutamatergic and GABAergic neurons (Extended Data Fig. 8g).

Furthermore, ex vivo patch-clamp recordings in slices from brain allografts revealed detectable biphasic sPSCs in a fraction of recorded SCLC cells (Extended Data Fig. 8h,i). Treatment of the slices with TTX, CNQX, D-AP5 or a combination of CNQX, D-AP5 and bicuculline reduced the occurrence of sPSCs, although the difference did not reach statistical significance (Extended Data Fig. 8i). Nevertheless, these data indicate that SCLC cells engage in synaptic transmission in brain tissue.

To substantiate these findings, we performed retrograde monosynaptic rabies virus (RABV) tracing experiments in SCLC cells using a replication-incompetent EnvA-pseudotyped G-gene-deficient virus, ΔG RABV-GFP, which can only infect cells expressing the avian viral receptor TVA. Following initial infection, cells that complement expression of the RABV glycoprotein (G) are able to transmit the virus to their first-order presynaptic partners[48]. After transduction of SCLC cells with a DsRed retrovirus encoding G and TVA (G-TVA), we co-cultured them with cortical neurons and added RABV-GFP to the cultures (Fig. 4e). In line with retrograde RABV-GFP spread from DsRed-positive SCLC 'starter cells' to synaptically connected neurons, we detected DsRed and GFP double-positive SCLC cells surrounded by clusters of GFP-positive neurons, which also displayed strong VGluT1 immunoreactivity (Fig. 4e and Extended Data Fig. 9a–c). Time-lapse experiments of these co-cultures showed that neurons acquired GFP fluorescence within 48 h of the appearance of DsRed and GFP double-positive SCLC starter cells (Extended Data Fig. 9d). Assessment of GFP-positive neurons in co-cultures with SCLC cells lacking any prior retroviral transduction or expressing only TVA and/or DsRed (but not G) as negative controls revealed a low and quantifiable level of spurious labelling by RABV under our conditions (Fig. 4f). By contrast, co-expression of G in SCLC cells resulted in a net increase in neuronal labelling of tenfold or more, corroborating the reliability of this transsynaptic approach (Fig. 4f). Analysis of co-cultures with either COR-L88 (SCLC-A) or DMS273 (SCLC-N) cells identified a connectivity ratio of 3 to 12 neurons per starter cancer cell (Fig. 4g).

We next conducted transsynaptic tracing experiments in brain allografts in vivo, by stereotactically co-injecting G- and TVA-expressing or TVA-only-expressing SCLC cells and EnvA-pseudotyped ΔG RABV-GFP into the mouse hippocampus (Fig. 4h). In animals injected with G-TVA-encoding virus, DsRed and GFP double-positive SCLC starter cells were typically surrounded by GFP-positive axonal fibres (Fig. 4h and Extended Data Fig. 9e). In line with this, GFP-positive neurons were found in the regions (hippocampus and subiculum) adjacent to grafted G-TVA-expressing SCLC cells, whereas, in control experiments with cancer cells expressing exclusively TVA, neuronal labelling was absent or minor (Fig. 4i and Extended Data Fig. 9f). Classification of neurons according to their morphology and layer positioning in traced anatomical regions near the injection area identified both putative excitatory and inhibitory neurons, further indicating that SCLC cells can be innervated by distinct neuronal subtypes in vivo (Fig. 4h,j and Extended Data Fig. 9e). These experiments indicate that SCLC cells are capable of forming functional synapses with neurons in vitro and in vivo.

## Neurons stimulate SCLC proliferation

To test whether SCLC cells derive a growth advantage when kept in co-culture with neurons, we compared the proliferative capacity of DsRed-expressing human COR-L88 cells seeded at low density and maintained either alone (monoculture) or in co-culture with cortical neurons, followed by 5-ethynyl-2′-deoxyuridine (EdU) labelling 2 h before analysis. While only a few scattered EdU-positive SCLC cells were

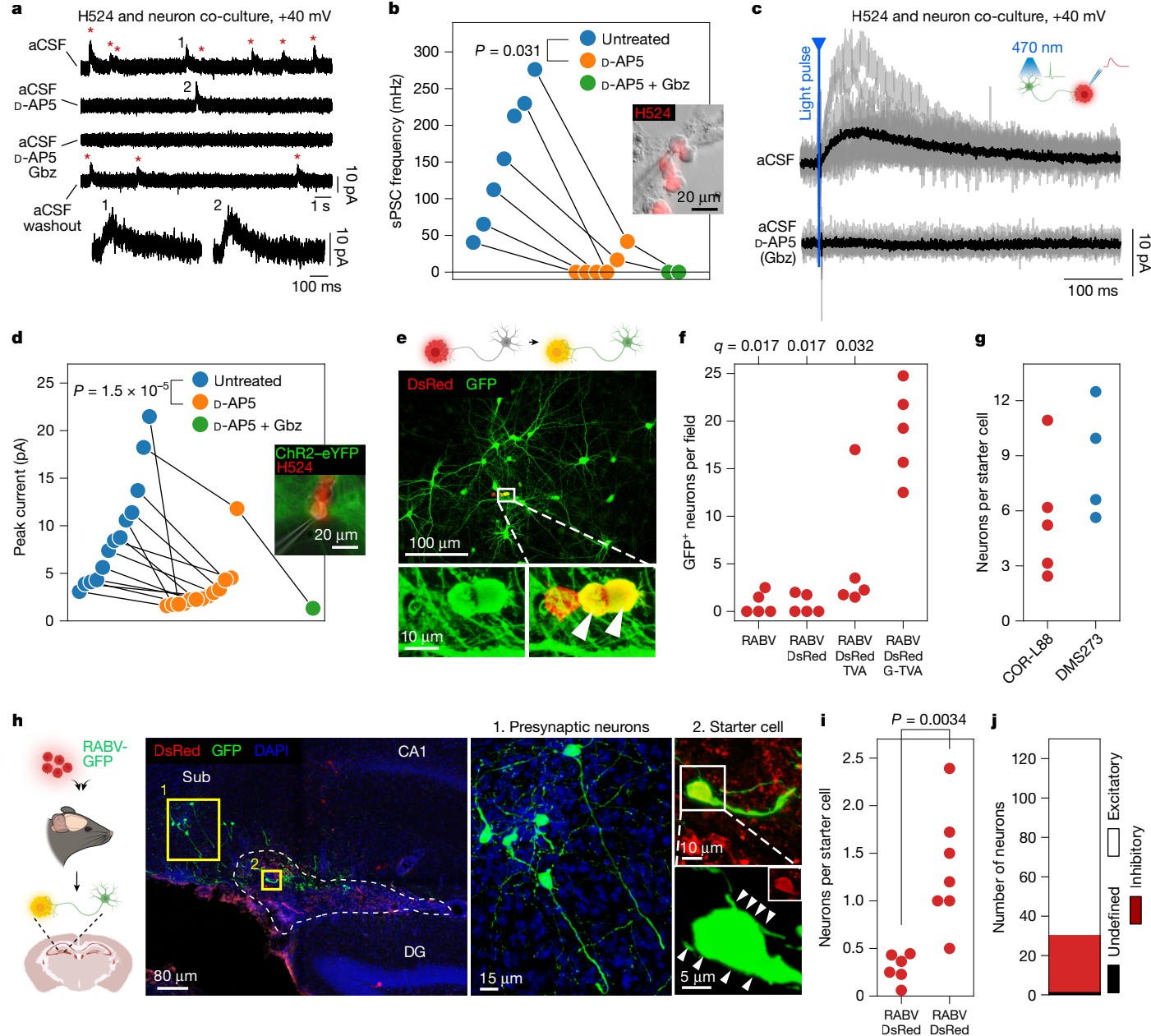

**Fig. 4 | Neuron-to-SCLC synapses are functional. a**, Whole-cell voltage-clamp traces in artificial cerebrospinal fluid (aCSF; control) and following treatment with NMDA receptor (D-AP5) and GABA$_A$ receptor (Gbz) blockers. Representative of seven cells across three experiments. Red asterisks or numbers mark individual events. **b**, Frequency of currents in H524 cells co-cultured with cortical neurons (untreated or exposed to D-AP5 alone or together with Gbz). Current frequency is compared before and after addition of D-AP5 (paired two-sided Wilcoxon test, $n = 6$ treated cells). Inset, example of a patched H524 cell. **c**, Whole-cell voltage-clamp traces of SCLC cells (grey) after a blue-light pulse (5 ms) to stimulate ChR2–enhanced yellow fluorescent protein (eYFP)-expressing neurons. The effects of D-AP5 ($n = 12/13$) or D-AP5 + Gbz ($n = 1/13$), compared to aCSF, are shown. **d**, Amplitude of evoked currents in H524 cells co-cultured with ChR2–eYFP-expressing cortical neurons after optogenetic stimulation. The amplitude before and after addition of D-AP5 is compared (two-sided paired Wilcoxon test, $n = 13$). Inset, example of a patched tdTomato-expressing H524 cell. **e**, Retrograde tracing of neurons monosynaptically connected to SCLC cells expressing DsRed, G and TVA after addition of EnvA-pseudotyped ($\Delta G$) RABV-GFP. Lower panels, magnified views of double-positive SCLC starter cells (arrowheads). **f**, Quantification of RABV-GFP-mediated neuronal labelling following SCLC transduction with virus encoding TVA alone or together with G ($n = 5$ biological replicates). All conditions are compared to the full experimental system (RABV, DsRed, G, TVA). $q$ values were obtained by two-sided Mann–Whitney test with FDR correction. **g**, Connectivity ratio per COR-L88 and DMS273 starter cell ($n = 4$–5 biological replicates). **h**, Retrograde tracing of neurons monosynaptically connected to G-TVA- and DsRed-expressing SCLC cells grafted into the mouse hippocampus. Right panels, magnified views of GFP-positive presynaptic excitatory neurons. GFP-only-positive axonal fibres contacting SCLC cells are indicated (arrowheads). CA1, cornus ammonis; DG, dentate gyrus; Sub, subiculum. **i**, Connectivity ratio per SCLC starter cell in mice grafted with TVA- or G-TVA-expressing SCLC cells ($n = 6$–7 mice per condition), $P$ value obtained by two-sided Mann–Whitney test. **j**, Number of traced GFP-positive neurons classified as excitatory or inhibitory ($n = 7$ mice).

found in monocultures, SCLC cells in co-cultures frequently appeared as larger proliferating clusters (Fig. 5a,b). This effect was significantly, but not completely, reduced when the co-cultures were treated with

TTX, suggesting that the proliferative advantage is mediated by both neuronal activity-dependent and neuronal activity-independent mechanisms (Fig. 5c,d).

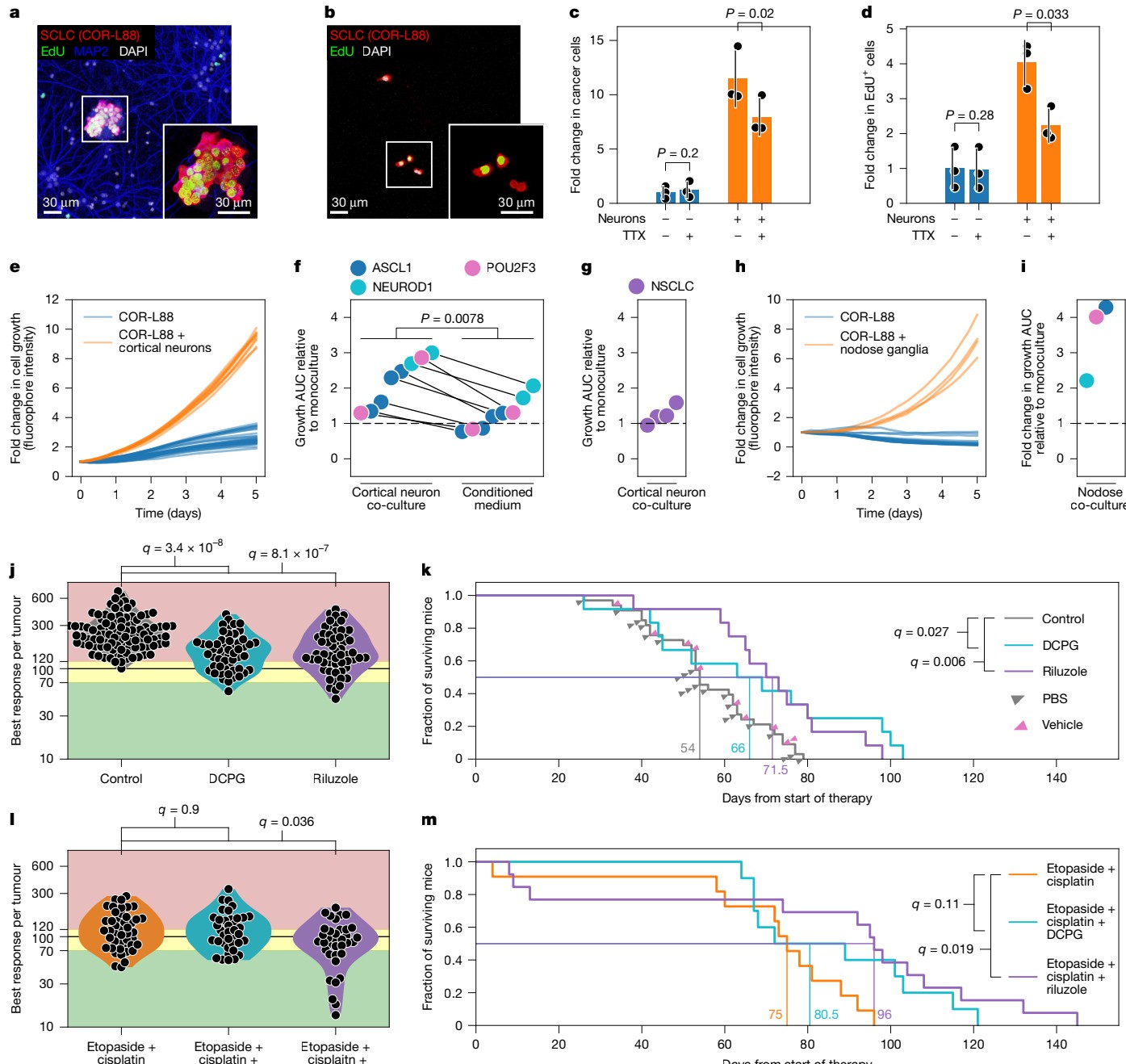

**Fig. 5 | Glutamatergic signalling constitutes an actionable target in SCLC. a,b**, DsRed-expressing COR-L88 SCLC cells cultured for 3 days with (**a**) or without (**b**) cortical neurons. **c,d**, Fold change in total (**c**) and EdU-positive (**d**) COR-L88 cells cultured for 3 days with or without neurons and/or TTX (*n* = 3; two-sided paired *t* test; centre, mean; error bars, s.d.). **e**, Growth curves of COR-L88 cells cultured with or without cortical neurons from one experiment (*n* = 10 and *n* = 30 wells). **f**, Quantification of live-cell imaging of SCLC cell lines (*n* = 8; in order: H526, H1836, H146, H69, COR-L88, DMS273, H211, H524). Cancer cells were cultured with cortical neurons or cortical neuron-conditioned medium. Proliferation was normalized to the growth of monocultures in the same plates. *P* value were obtained by two-sided Wilcoxon signed-rank test. AUC, area under the curve. **g**, Growth quantification of NSCLC cell lines (*n* = 4; in order: HOP62, HCC44, H2291, H1975) in co-culture with cortical neurons, normalized to the growth of monocultures. **h**, Growth curves of COR-L88 cells

cultured with or without nodose ganglia from two experiments (*n* = 4 and *n* = 16 wells). **i**, Growth quantification of SCLC cell lines (*n* = 3; in order: DMS273, H211, COR-L88) co-cultured for 5 days with nodose ganglion explants, relative to monocultures. **j**, Response of tumours in mice treated with vehicle (*n* = 102), DCPG (*n* = 54) or riluzole (*n* = 57), expressed as percentage of the initial volume. *q* values were obtained by two-sided Mann–Whitney test with FDR correction. **k**, Overall survival of RP mice treated with DCPG (*n* = 12), riluzole (*n* = 12) or the relative control (*n* = 33). *q* values were obtained by two-sided log-rank test with FDR correction. **l**, Response of tumours in mice treated with etoposide + cisplatin alone (*n* = 45) or combined with DCPG (*n* = 38) or riluzole (*n* = 36), expressed as percentage of the initial volume. *q* values as in **j**. **m**, Overall survival of RP mice treated with etoposide + cisplatin alone (*n* = 11) or combined with riluzole (*n* = 13) or DCPG (*n* = 10). *q* values as in **k**. See Extended Data Fig. 10 for individual replicates of **f, g** and **i**.

To evaluate whether neuronal co-culture stimulates proliferation in all SCLC subtypes, we monitored the growth of eight distinct cell lines with live-cell imaging: COR-L88, H1836, H69 and H146

(all SCLC-A), DMS273 and H524 (SCLC-N), and H211 and H526 (SCLC-P). All lines, including the SCLC-P lines, derived a significant proliferation advantage when co-cultured with cortical neurons (Fig. 5e,f and

Extended Data Fig. 10a). Some cell lines (COR-L88, H69, DMS273, H524 and H211) also derived a minor proliferation advantage when cultured in conditioned medium derived from neuronal cultures. However, for all cell lines, physical presence of the neurons conferred a significantly stronger proliferation advantage (Fig. 5f and Extended Data Fig. 10a). The proliferative advantage appeared to be specific for neurons, as it vastly exceeded that observed in high-density monocultures (Extended Data Fig. 10a) and it was even stronger in four of the cell lines than that conferred by co-culture with fibroblasts, which have been shown to strongly promote SCLC growth[49] (Extended Data Fig. 10a). The effects seemed to be largely specific for SCLC, as four non-small cell lung cancer (NSCLC) lines (H1975, HCC44, HOP62 and H2291) derived only a minor proliferation advantage when co-cultured with neurons (Fig. 5g and Extended Data Fig. 10b). Finally, increased proliferation also occurred when co-culturing SCLC cells with mouse nodose ganglia (Fig. 5h,i and Extended Data Fig. 10c–e). These data indicate that all major SCLC subtypes derive a growth benefit when co-cultured with neurons. This advantage is at least partially dependent on neuronal activity and physical proximity.

## Targeting glutamate signalling in SCLC

Given the formation of functional synapses between SCLC cells and glutamatergic neurons in vitro and in vivo (Figs. 3 and 4), we next sought to target the glutamatergic system therapeutically. The SCLC samples that we analysed at the expression level can be broadly classified into classic SCLC with strong neuroendocrine features and variant SCLC with lower expression of neuroendocrine features, using a lung-specific neuroendocrine score[50] (Extended Data Fig. 10f). Expression of genes in the GO term 'glutamatergic synapse' was particularly high in the *ASCL1*- and *NEUROD1*-expressing subtypes (Extended Data Fig. 10g), suggesting that these subtypes in particular might benefit from interference with the glutamatergic system.

Among the possible molecular targets in this system are the glutamate receptors, which we also identified as individual genes targeted by transposon insertion in our *piggyBac* screen (*Grid1*, *Grik2*, *Grin3a*, *Grm1*, *Grm3*, *Grm5* and *Grm8*), in human mutation data (*GRIA1*, *GRIA2*, *GRIA3*, *GRIA4*, *GRID2*, *GRIK2*, *GRIK3*, *GRIK4*, *GRIN2A*, *GRIN2B*, *GRIN3A*, *GRM1*, *GRM3*, *GRM5* and *GRM8*) and at the expression level in human samples (*GRIA2*, *GRIN3A*, *GRIK3*, *GRIK5*, *GRM2*, *GRM4* and *GRM8*). Prominent among them was *GRM8*, a gene encoding an inhibitory metabotropic glutamate receptor that has been shown to counteract glutamate signalling by negatively regulating cyclic AMP-dependent sensitization of inositol 1,4,5-trisphosphate receptors, thereby limiting glutamate-induced calcium release from the endoplasmic reticulum[51]. *GRM8* has been identified as an ASCL1 and NEUROD1 ChIP–seq target in human SCLC cell lines[52], and expression of GRM8 correlates with the expression of ASCL1 in cell lines reported in SCLC-CellMiner[53] and is reduced in autochthonous SCLC mice in which *Ascl1* is deleted specifically in cancer cells[12]. In our datasets, *GRM8* showed expression specifically in SCLC and a few other tumour types (Extended Data Fig. 10h), a statistically significant enrichment of both non-synonymous mutations and more severe loss-of-function mutations (Extended Data Fig. 10i), and a significant number of *piggyBac* insertions (Extended Data Fig. 10j). Re-analysis of the scRNA-seq data from ref. 44 confirmed that *GRM8* is specifically expressed in SCLC cells (Extended Data Fig. 10k,l). We also found specific expression of *Grm8* in our mouse SCLC snRNA-seq dataset, although at a substantially reduced fraction compared with the human dataset (Extended Data Fig. 10m). This specific expression suggests that GRM8 can be targeted, while the enrichment in loss-of-function mutations suggests that the activity of GRM8 is detrimental to SCLC tumours. On the basis of these data, we selected (*S*)-3,4-dicarboxyphenylglycine (DCPG) and riluzole, two compounds with predicted anti-glutamatergic effects, for preclinical testing. DCPG is a potent and selective agonist of GRM8 (ref. 54), while

riluzole is a US Food and Drug Administration-approved inhibitor of glutamate release that inhibited sPSCs in our co-culture experiments (Extended Data Fig. 8c).

To evaluate the efficacy of DCPG and riluzole in vivo, we exposed tumour-bearing RP mice to DCPG, riluzole or vehicle. Responses were evaluated every 2 weeks by magnetic resonance imaging (MRI). Whereas all tumours from vehicle-treated mice progressed during treatment, the response was significantly improved both in DCPG- and in riluzole-treated animals, with several tumours showing short-term stable disease or slower growth and a small subset of tumours showing modest shrinkage (Fig. 5j and Extended Data Fig. 11a–j). Mice treated with DCPG and riluzole also showed significantly improved survival, with a median survival of 66 days (DCPG) and 71.5 days (riluzole), compared to 54 days in the control group (Fig. 5k). To further confirm these findings, we tested the efficacy of riluzole and DCPG in a second cohort of mice using CGRP-driven expression of Cre, which has been shown to more selectively induce transformation in PNECs[10–13]. In this cohort, DCPG did not significantly improve response or survival, whereas treatment with riluzole resulted in significantly improved response and a significant survival advantage, compared with vehicle control (Extended Data Fig. 11k,l).

We then compared mice with CMV-induced expression of Cre treated with cisplatin and etoposide with mice that received this chemotherapy plus DCPG or riluzole. Tumours exposed to chemotherapy alone showed a mixed response, which included shrinkage, stable disease and progressive disease. In the context of chemotherapy, inclusion of DCPG did not significantly improve the response of SCLC tumours (Fig. 5l and Extended Data Fig. 12a–d,g–j). By contrast, chemotherapy in combination with riluzole resulted in a significantly improved response, with almost all tumours showing partial response or stable disease and slower growth for more than 2 months (Fig. 5l and Extended Data Fig. 12a,b,e–j). Similarly, inclusion of DCPG in the frontline chemotherapy regimen in our mice did not result in significantly improved survival, whereas addition of riluzole resulted in a significant improvement in survival of 21 days (Fig. 5m).

Thus, targeting glutamatergic signalling has preclinical activity against SCLC both alone and in combination with frontline chemotherapy in vivo.

## Discussion

We performed an in vivo insertional mutagenesis screen in a mouse model of SCLC and cross-validated our findings through the re-analysis of genetic and expression data from human SCLC. Unexpectedly, almost all gene sets we identified at the genetic level were related to a neuron-like phenotype in general and to synapses and glutamatergic signalling in particular.

Our co-culture and transplantation experiments revealed a striking ability of SCLC cells to form synapses and receive direct neurotransmitter-mediated inputs. These data reveal that functional, bona fide synapses can form between neurons and cancer cells of extracranial origin.

We speculate that the ability to form synapses is part of the PNEC-like phenotype of SCLC. In line with this notion, we detected fibres known to innervate PNECs, such as VGluT1- and P2X3-positive vagal fibres[14], in a subset of small SCLC tumours in mouse lungs. Here we also detected co-localization of presynaptic VGluT1 and postsynaptic HOMER1 in cancer cells, suggesting that synapses may also form in this primary setting. The precise nature and functionality of these contacts in the lung, as well as a potential role in cancer initiation, remain to be determined.

All SCLC cell lines we tested derived a growth advantage when co-cultured with neurons. This advantage was at least partially dependent on direct neuronal innervation and neuronal activity. However, the advantage was not fully abolished by TTX, suggesting the presence of additional, action potential-independent contributions to the

proliferation of cancer cells. For example, we detected a small effect of neuron-conditioned medium in vitro. We also cannot exclude a paracrine contribution in vivo, as we observed several CGRP-positive, SP-positive and GAP43-positive nerve fibres around autochthonous RP tumours, which could potentially engage in paracrine communication with the tumours. Similarly, although we focused mainly on glutamatergic contacts, the 'GABAergic synapse' GO term was also identified in our genetic screen and the potential for GABAergic communication between neurons and SCLC cells was corroborated by our electrophysiological and tracing experiments.

As SCLC is characterized by a high degree of inter- and intratumoral heterogeneity and plasticity[2,13,55,56], the general exploitability of glutamate-targeting strategies and potential therapy sequencing algorithms remain to be defined. Our data indicate that SCLC may be capable of hijacking neuronal programmes, such as the ability to form synapses, to derive a growth advantage. As we show for anti-glutamatergic drugs, investigation of these neuronal phenotypes may hold the key to finally providing more effective therapies to patients with SCLC.

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

Vignesh Sakthivelu[1,2,3,43], Anna Schmitt[2,4,43], Franka Odenthal[5,43], Kristiano Ndoci[6,43], Marian Touet[2,7,43], Ali H. Shaib[8,43], Abdulla Chihab[6,43], Gulzar A. Wani[6], Pascal Nieper[1,2], Griffin G. Hartmann[9,10], Isabel Pintelon[11,12], Ilmars Kisis[1,2,3], Maike Boecker[1,2,3], Naja M. Eckert[1], Manoela Iannicelli Caiaffa[1,2,3], Olta Ibruli[1,2], Julia Weber[13], Roman Maresch[13], Christina M. Bebber[1,6], Ali Chitsaz[1,2,3], Anna Lütz[1,2,3], Mira Kim Alves Carpinteiro[1,2,3], Kaylee M. Morris[5], Camilla A. Franchino[5], Jonas Benz[6], Laura Pérez-Revuelta[6], Jorge A. Soriano-Campos[6], Maxim A. Huetzen[2,6,14,15], Jonas Goergens[2,6,14,15], Milica Jevtic[6], Hannah M. Jahn-Kelleter[6], Hans Zempel[15,16], Aleksandra Placzek[6], Alexandru A. Hennrich[17], Karl-Klaus Conzelmann[17,45], Hannah L. Tumbrink[1,18], Pascal Hunold[1,15], Joerg Isensee[19,20], Lisa Werr[21], Felix Gaedke[6], Astrid Schauss[6], Marielle Minère[6,22], Marie Müller[6,22], Henning Fenselau[6,22,23], Yin Liu[24], Alena Heimsoeth[1,15,18], Gülce S. Gülcüler Balta[1,2,3], Henning Walczak[6,25,26], Christian Frezza[27,28], Ron D. Jachimowicz[2,6,14,15], Julie George[1,29], Marcel Schmiel[30], Johannes Brägelmann[1,3,15], Tim Hucho[19,20], Silvia von Karstedt[1,6,15], Martin Peifer[1,15], Alessandro Annibaldi[15], Robert Hänsel-Hertsch[1,6,15,16], Thorsten Persigehl[31], Holger Grüll[31], Martin L. Sos[1,32,33], Guido Reifenberger[34], Matthias Fischer[15,21], Dirk Adriaensen[11], Reinhard Büttner[30], Julien Sage[9,10], Inge Brouns[11], Roland Rad[13], Roman K. Thomas[1,30], Max Anstötz[35,44 ✉], Silvio O. Rizzoli[8,36,37,44 ✉], Matteo Bergami[6,15,38,39,44 ✉], Elisa Motori[5,6,15,44 ✉], Hans Christian Reinhardt[7,40,41,42,44 ✉] & Filippo Beleggia[1,2,3,44 ✉]

[1]Department of Translational Genomics, Faculty of Medicine and University Hospital Cologne, University of Cologne, Cologne, Germany. [2]Department I of Internal Medicine, Faculty of Medicine and University Hospital Cologne, University of Cologne, Cologne, Germany. [3]Mildred Scheel School of Oncology Aachen Bonn Cologne Düsseldorf (MSSO ABCD), Faculty of Medicine and University Hospital Cologne, University of Cologne, Cologne, Germany. [4]Institute of Virology, Faculty of Medicine and University Hospital Cologne, University of Cologne, Cologne, Germany. [5]Institute of Biochemistry, University of Cologne, Cologne, Germany. [6]Cologne Excellence Cluster on Cellular Stress Response in Aging-Associated Diseases (CECAD), University of Cologne, Cologne, Germany. [7]Department of Hematology and Stem Cell Transplantation, University Hospital Essen, Essen, Germany. [8]Department of Neuro- and Sensory Physiology, University Medical Center Göttingen, Göttingen, Germany. [9]Department of Pediatrics, Stanford University, Stanford, CA, USA. [10]Department of Genetics, Stanford University, Stanford, CA, USA. [11]Laboratory of Cell Biology and Histology, Department of Veterinary Sciences, University of Antwerp, Antwerp, Belgium. [12]Antwerp Centre for Advanced Microscopy (ACAM), University of Antwerp, Antwerp, Belgium. [13]Institute of Molecular Oncology and Functional Genomics, School of Medicine, Technical University of Munich, Munich, Germany. [14]Max Planck Institute for Biology of Ageing, Cologne, Germany. [15]Center for Molecular Medicine Cologne (CMMC), Faculty of Medicine and University Hospital Cologne, University of Cologne, Cologne, Germany. [16]Institute of Human Genetics, Faculty of Medicine and University Hospital Cologne, University of Cologne, Cologne, Germany. [17]Virology, Max von Pettenkofer Institute, Faculty of Medicine and Gene Center, Ludwig Maximilians University, Munich, Germany. [18]Molecular Pathology, Institute of Pathology, Faculty of Medicine and University Hospital Cologne, University of Cologne, Cologne, Germany. [19]Translational Pain Research, Department of Anesthesiology and Intensive Care Medicine, Faculty of Medicine and University Hospital Cologne, University of Cologne, Cologne, Germany. [20]Pain Center, Department of Anesthesiology and Intensive Care Medicine, Faculty of Medicine and University Hospital Cologne, University of Cologne, Cologne, Germany. [21]Department of Experimental Pediatric Oncology, University Children's Hospital of Cologne, Medical Faculty, University of Cologne, Cologne, Germany. [22]Research Group Synaptic Transmission in Energy Homeostasis, Max Planck Institute for Metabolism Research, Cologne, Germany. [23]Policlinic for Endocrinology, Diabetes and Preventive Medicine, University Hospital Cologne, Cologne, Germany. [24]Janelia Research Campus, Howard Hughes Medical Institute, Ashburn, VA, USA. [25]Cell Death, Inflammation and Immunity Laboratory, Institute of Biochemistry I, Center for Biochemistry, Faculty of Medicine, University of Cologne, Cologne, Germany. [26]Centre for Cell Death, Cancer and Inflammation, UCL Cancer Institute, University College London, London, UK. [27]Institute for Metabolomics in Ageing, University of Cologne, Faculty of Medicine and University Hospital Cologne, Cologne Excellence Cluster on Cellular Stress Responses in Aging-Associated Diseases (CECAD), Cologne, Germany. [28]Institute of Genetics, University of Cologne, Faculty of Mathematics and Natural Sciences, Cologne Excellence Cluster on Cellular Stress Responses in Aging-Associated Diseases (CECAD), Cologne, Germany. [29]Department of Otorhinolaryngology, Head and Neck Surgery, University Hospital Cologne, Cologne, Germany. [30]Institute of Pathology, Faculty of Medicine and University Hospital Cologne, University of Cologne, Cologne, Germany. [31]Institute for Diagnostic and Interventional Radiology, Faculty of Medicine and University Hospital Cologne, University of Cologne, Cologne, Germany. [32]Department of Translational Oncology, German Cancer Consortium (DKTK), partner site Munich, a partnership between DKFZ and Ludwig-Maximilians-Universität München (LMU), Munich, Germany. [33]Department of Medicine III, LMU University Hospital Munich, Munich, Germany. [34]Institute of Neuropathology, Medical Faculty, Heinrich Heine University and University Hospital Düsseldorf, Düsseldorf, Germany. [35]Institute of Anatomy II, Medical Faculty, University Hospital Düsseldorf, Heinrich Heine University, Düsseldorf, Germany. [36]Center for Biostructural Imaging of Neurodegeneration, University Medical Center Göttingen, Göttingen, Germany. [37]Cluster of Excellence 'Multiscale Bioimaging: from Molecular Machines to Networks of Excitable Cells' (MBExC), University of Göttingen, Göttingen, Germany. [38]Institute of Genetics, University of Cologne, Cologne, Germany. [39]Faculty of Medicine and University Hospital Cologne, University of Cologne, Cologne, Germany. [40]West German Cancer Center, University Hospital Essen, Essen, Germany. [41]DKTK, partner site Essen, University Hospital Essen, Essen, Germany. [42]Center for Molecular Biotechnology, University Hospital Essen, Essen, Germany. [43]These authors contributed equally: Vignesh Sakthivelu, Anna Schmitt, Franka Odenthal, Kristiano Ndoci, Marian Touet, Ali H. Shaib, Abdulla Chihab. [44]These authors jointly supervised this work: Max Anstötz, Silvio O Rizzoli, Matteo Bergami, Elisa Motori, Hans Christian Reinhardt, Filippo Beleggia. [45]Deceased: Karl-Klaus Conzelmann. ✉e-mail: max.anstoetz@hhu.de; srizzol@gwdg.de; matteo.bergami@uk-koeln.de; elisa.motori@uni-koeln.de; christian.reinhardt@uk-essen.de; filippo.beleggia@uk-koeln.de

## Methods

### Mice

This study was performed in accordance with FELASA recommendations and with European Union and German guidelines. The experiments were approved by the local ethics committee on animal experiments (Landesamt für Natur, Umwelt und Verbraucherschutz Nordrhein-Westfalen). Mice were housed in groups of up to five animals per cage and supplied with standard pellet food and water ad libitum with a 12-h light/12-h dark cycle, while the temperature was controlled to 21–22 °C with a relative humidity of 45–65%. Animals were regularly examined for body condition, body weight, accelerated breathing, behaviour, tumour size (<1.5 cm in diameter) and neurological symptoms. In compliance with the respective animal permissions, animals were killed before or immediately after reaching a severe burden. Mice of both sexes were included. For animal experiments performed at Stanford University, mice were maintained according to practices approved by the US National Institutes of Health, the Stanford Institutional Animal Care and Use Committee and the Association for Assessment and Accreditation of Laboratory Animal Care. The study protocol was approved by the Stanford Administrative Panel on Laboratory Animal Care (protocol 13565).

### Cell lines

Mouse cell lines (AVR424.3 and RP1462) were isolated from mouse tumours in the RP line and identified by genotyping. Human cell lines (COR-L88, H1836, H69, H146, DMS273, H524, H211, H526, H1975, HCC44, HOP62, H2291 and HEK293T) were gifts from R. Thomas, University Hospital of Cologne, and identified through STR profiling. All cell lines were tested for mycoplasma contamination.

### Statistics and reproducibility

The statistical tests used are reported in the figure legends and specific methods sections. No measurements were performed more than once on the same sample. Statistical analyses were performed with Python v3.8, v3.9 and v3.10 with the packages pandas v1.1.4 and numpy v1.20. Whenever necessary, correction for multiple testing was performed with the FDR using the Python package statsmodel v0.12.2 with the method 'Benjamini/Hochberg'. Pearson and Spearman correlation coefficients and the corresponding $P$ values were calculated using scipy v1.6.3. Statistical analysis of survival was performed with lifelines v0.25.6. The packages matplotlib v3.4.2 and seaborn 0.11.0 were used for visualization. The micrographs depicted are representative of repeated experiments, as detailed in the figure legends or as follows: Fig. 2a, 10 experiments; Fig. 2b, 2 experiments; Fig. 2c, 9 experiments; Fig. 2d–i, 3 experiments; Fig. 3i, 3 experiments; Fig. 4e, 4 experiments; Fig. 4h, 7 experiments; Fig. 5a,b, 3 experiments; Extended Data Fig. 1c,d, 3 experiments; Extended Data Fig. 3c,d, 7 experiments; Extended Data Fig. 3e, 5 experiments; Extended Data Fig. 5a, 10 experiments; Extended Data Fig. 5b, 2 experiments; Extended Data Fig. 5c,d, 7 experiments; Extended Data Fig. 5e,f, 3 experiments; Extended Data Fig. 5g, 5 experiments; Extended Data Fig. 5h, 3 experiments; Extended Data Fig. 5i–k, 2 experiments; Extended Data Fig. 6a, 3 experiments; Extended Data Fig. 6e, 2 experiments; Extended Data Fig. 6f–i, 4 experiments; Extended Data Fig. 7a, 2 experiments; Extended Data Fig. 7b–d, 3 experiments; Extended Data Fig. 7f, 2 experiments; Extended Data Fig. 8a, 40 experiments; Extended Data Fig. 9a, 4 experiments; Extended Data Fig. 9b,c, 4 experiments; Extended Data Fig. 9d, 3 experiments; Extended Data Fig. 9e, 7 experiments; Extended Data Fig. 9f, 6 experiments; Extended Data Fig. 10c,d, 9 experiments.

### SCLC tumour induction

To induce lung tumour formation and, when present, activation of the *piggyBac* transposition system or the *Cas9-EGFP* allele, 8- to 12-week-old mice of both sexes were anaesthetized with ketavet (100 mg kg$^{-1}$) and xylazine (20 mg kg$^{-1}$) by intraperitoneal injection, followed by intratracheal instillation of replication-deficient adenovirus encoding Cre recombinase (Adeno-Cre, $2.5 \times 10^7$ plaque-forming units (PFU)). Viral vectors were provided by the University of Iowa Viral Vector Core (http://www.medicine.uiowa.edu/vectorcore).

### MRI

An Achieva 3.0-T clinical MRI system (Philips Healthcare) in combination with a dedicated mouse solenoid coil (Philips Healthcare) was used for imaging. Animals were anaesthetized using isoflurane (2.5%), and T2-weighted MR images were acquired in the axial plane using a turbo-spin echo sequence (repetition time, 3,819 ms; echo time, 60 ms; field of view, $40 \times 40 \times 20$ mm$^3$; reconstructed voxel size, $0.13 \times 0.13 \times 1.0$ mm$^3$; number of average, 1). MR images (DICOM files) were analysed in a blinded fashion by determining and calculating regions of interest (ROIs) using Horos software v3.0 with the package Export Rois v2.0.

### *PiggyBac* transposition system in SCLC

For activation of transposition in an SCLC mouse model, we used the following alleles, as detailed in Extended Data Fig. 1: *Rosa26*$^{LSL-PB}$, *ATP1-S2*, *ATP1-H39*, *Rb1*$^{flox}$ and *Trp53*$^{flox}$ (refs. 25,26). Mice were kept on a mixed C57BL/6–Sv/129 background. The *Trp53*$^{flox}$ allele was genotyped with primers Trp53fw (CACAAAAACAGGTTAAACCCAG) and Trp53rv (AGCACATAGGAGGCAGAGAC). The *Rb1*$^{flox}$ allele was genotyped with primers RB1_F3 (GAAGCCATTGAAATCTACCTCCCTTGCCCTGT), RB1_F_4 (ACTCATGGACTAGGTTAAGT), RB1_R_1 (TGCCATCAATGCCCGG TTTAACCCCTGT) and RB1_R_2 (AGCATTTTATATGCATTTAATTGTC). The *ATP1* alleles were genotyped using primers ATP-F (CTCGTTAATCGCC GAGCTAC) and ATP-R (GCCTTATCGCGATTTTACCA). The *Rosa26*$^{LSL-PB}$ knock-in allele was genotyped using primers BpA5F (GCTGGGGATG CGGTGGGCTC) and Rosa3R (GGCGGATCACAAGCAATAATAACCTGT AGTTT). The wild-type *Rosa26* allele was detected with primers Rosa5F (CCAAAGTCGCTCTGAGTTGTTATCAG) and Rosa3R (GGCGGATCACAAG CAATAATAACCTGTAGTTT). To study SCLC formation, all four mouse lines were imaged following adenoviral instillation, as described above. After reaching the termination criteria, mice were killed and single tumour nodules were isolated and used for DNA extraction. Analysis of transposon mobilization at the donor locus and splinkerette-PCR amplification of transposon insertion sites were performed as previously described[26,57].

### Treatment of *piggyBac* mice

Starting 5 months after tumour induction, tumour growth was monitored through biweekly MRI as described above until termination criteria were reached. Following tumour detection (minimum tumour size of 3 mm$^3$), RPLS and RPLH mice were treated with either a combination of cisplatin and etoposide or the anti-PD-1 antibody RMP1-14. Compound solutions were prepared and injected as follows: etoposide (Hexal) was administered on days 1, 2 and 3 of a 14-day cycle, intraperitoneally, at a concentration of 10 mg kg$^{-1}$. Cisplatin (Accord) was administered intraperitoneally on day 1 of a 14-day cycle at a concentration of 5 mg kg$^{-1}$. The anti-PD-1 antibody RMP1-14 (BioXCell) was administered intraperitoneally 2 days per week (250 µg per administration).

### Deletion of *Reln* in the RPR2 model of SCLC

Generation of the *Rb1*$^{fl/fl}$*Trp53*$^{fl/fl}$*Rbl2*$^{fl/fl}$*Rosa26*$^{LSL-tdTomato/LSL-tdTomato}$*H11*$^{LSL-Cas9/LSL-Cas9}$ (RPR2;TC) mice used in this study has been described previously[41]. Forty-eight hours before lentivirus delivery, naphthalene (Sigma-Aldrich, 184500) was dissolved in corn oil vehicle (Sigma-Aldrich, C8267) at a concentration of 50 mg ml$^{-1}$ and administered to mice (8- to 12-week-old males and females) through intraperitoneal injection at a dosage of 200 mg kg$^{-1}$. Mice were then instilled with Lenti-sgRNA/Cre viruses ($1.5 \times 10^6$ PFU for each condition) through intratracheal delivery to generate lung tumours. Five months after

tumour induction, tissues were dissected from mice after they were killed and perfused with 10% neutral-buffered formalin (NBF). Lungs were inflated with 10% NBF and fixed in 10% NBF overnight. Tissues were transferred to 70% ethanol before paraffin embedding and processing. Quantification of lung tumour number and area on sections stained with haematoxylin and eosin was performed in a blinded fashion using ImageJ v1.54h. sgRNAs targeting *Reln* (Reln_a756, GACCCCA TCTAAGCCAAACGG; Reln_a894, GAACTGGACATACATAGTAT) and a non-targeting guide (GCGAGGTATTCGGCTCCGCG) were cloned into the pLL3 backbone[58] (https://www.addgene.org/browse/article/15541/). Each Lenti-sgRNA/Cre virus was packaged separately in HEK293T cells through cotransfection with polyethylenimine alongside pCMV-VSV-G (Addgene, 8454) envelope plasmid and pCMV-dR8.2 dvpr (Addgene, 8455) packaging plasmid. The medium was replaced 24 h after transfection. Virus-containing supernatant was collected at 48- and 72-h time points following transfection, filtered using 0.45-µm syringe filters, concentrated by ultracentrifugation at 25,000 RPM for 90 min at 4 °C, resuspended in PBS and titered using LSL-YFP mouse embryonic fibroblasts as previously described[59].

## Reference genomes and gene definitions

The reference genome used for all human analyses was TCGA GRCh38. d1.vd1, with the exception of the comparison of human RNA-seq data to GTEx data, which was performed using the GTEx v8 reference (Homo_sapiens_assembly38_noALT_noHLA_noDecoy_ERCC.fasta). The reference genome used for all mouse analyses was Ensembl version GRCm38.102, with the exception of the analysis of snRNA-seq data, which was performed using Ensembl reference GRCm39.110. The gene annotation for analyses of human genetic data was GENCODE annotation v22, while the gene annotation for analyses of mouse data was GENCODE annotation vM23 (ref. 60). Both GENCODE annotations were filtered first to include only transcripts marked as 'protein coding' and subsequently to include only the 17,153 genes for which a one-to-one orthologue could be identified between mouse and human using the HCOP 15-column orthology table (downloaded on 6 January 2020 from the HGNC database[61]). The gene annotation for analysis of TCGA expression data was GENCODE annotation v22, and the gene annotation used for analysis of GTEx expression data was GENCODE annotation v26.

## Analysis of *piggyBac* insertions

Sequencing reads that contained internal transposon sequences were excluded, and the remaining reads were aligned against the GRCm38 reference using BWA v0.7.15 and samtools v1.3.1. Aligned reads that did not align to the consensus TTAA target sequence were excluded. At each TTAA locus in each sample, reads derived from the same fragment, identified by the identical position of the read ends, were collapsed. TTAA loci were kept if five or more different fragments were identified. Germline insertions were identified by the presence of ten or more different fragments at a TTAA locus in tail or ear samples. These TTAA loci were excluded from analysis in the whole cohort and the sequences 1 Mb upstream and downstream were masked from analysis of tumours from the affected mice. The 10-Mb regions encompassing the donor loci were also masked from analysis (chromosome 5:50000000–70000000 for the RPLH line and chromosome 10:0–10000000 for the RPLS line). Insertions detected in more than one tumour were assigned to the tumour with the highest number of fragments. For each of the 17,153 protein-coding genes present in both the human and mouse genomes, we defined the included genomic range as the union of all the transcripts for the gene from the transcription start site (TSS) to the stop codon. The statistical analysis included two steps. First, at the sample level, the Poisson distribution was used to calculate the one-sided probability of seeing at least as many transposon fragments as were actually present. The rate used for the Poisson distribution was based on the total insertion rate within genes on each chromosome of each sample, on the total number of TTAA sites within genes on the chromosome

and on the number of TTAA sites within each gene. We then calculated FDR-corrected $q$ values for each sample and each gene. We obtained a total of 11,208 genes (an average of 37 genes per sample) that were significant at a cutoff of $q < 0.05$ at the sample level. To calculate the statistical significance of genes at the cohort level, we again used the Poisson distribution with a rate derived from distributing the 11,208 hits evenly across all non-masked genes of all samples and calculated the one-sided probability of seeing at least as many insertions in a given gene. We then calculated the FDR-corrected $q$ value at the cohort level for each gene.

## Analysis of *piggyBac* subcohorts

We used a two-sided permutation test to compare the distribution of the transposon insertions in different subcohorts: untreated versus chemotherapy, untreated versus immunotherapy and lung tumours versus metastatic tumours. For each comparison, the union of samples included in the comparison was shuffled 1,000,000 times, while maintaining the same number of samples from each mouse line in each subcohort (RPLH and RPLS). For each gene, we then counted the number of iterations in which the absolute difference in the fraction of samples carrying an insertion was greater than in the real configuration. We calculated the FDR-corrected $q$ value for each gene.

## Simulation and annotation of possible human mutations

For each gene included in the filtered GENCODE annotation v22, all possible single-nucleotide substitutions were simulated, annotated using ANNOVAR v2018Apr16 (ref. 62) with the filtered GENCODE annotation v22 and divided into three categories: synonymous (no predicted change in the protein sequence), severe (causing a premature stop, loss of the starting ATG site, a frameshift or a nucleotide change in one of the two intronic bases flanking each side of an exon) and non-synonymous (any other predicted change in the protein sequence). For each simulated variant in each gene, only the most severe consequence among all the transcripts associated with the gene was kept. All simulated variants were also annotated using the total population frequency in non-cancer samples from the gnomAD v2.1.1 GRCh38 liftover exome and the gnomAD v3 genomes and excluded if they were found in more than 1 in 10,000 samples. On the basis of this simulation, the number of possible non-synonymous or severe variants for each gene was used for calculation of the expected number of mutations in each gene.

## Data collection of human somatic mutations

Sample information and mutations were downloaded from the supplementary tables of the respective papers or from the Cancer Cell Line Encyclopedia (CCLE) website (Cell_lines_annotations_20181226. txt and CCLE_DepMap_18q3_maf_20180718.txt; https://portals.broadinstitute.org/ccle/). Where needed, the mutations were mapped to the TCGA GRCh38 reference (GRCh38.d1.vd1.fa) using the liftOver v385 tool from the UCSC database (http://genome.ucsc.edu; ref. 63). The resulting 177,983 mutations were annotated as described above for the simulated variants; 613 mutations were excluded from analysis (517 mapped to mitochondrial genes and 96 could not be mapped to primary chromosomes in h38). The remaining 177,370 variants were left-aligned using GATK LeftAlignAndTrimVariants v4.1.3.0 (ref. 64). A total of 28 samples were excluded from analysis because they shared five or more mutations with a sample from a more recent study, leaving 456 samples.

## Statistical analysis of the human cohort

Samples sharing five or more mutations were merged (e.g., samples sequenced both before and after treatment). In total, 439 samples and 117,353 non-synonymous mutations were used for analysis. We used the Poisson distribution to estimate the one-sided probability of observing at least as many mutations by chance in each gene. To obtain the rate for the Poisson distribution for each gene, we divided the total number of

non-synonymous mutations, counting each sample at most twice per gene, by the total number of possible non-synonymous mutations within the 17,153 protein-coding genes present in both the human and mouse genomes. For each gene, we then multiplied this value by the number of non-synonymous mutations that were theoretically possible in the gene (see simulations above). The rate therefore represented the expected number of non-synonymous mutations under a uniform distribution model. For each gene, we then calculated the probability of observing at least as many mutations as were actually present. We corrected the resulting $P$ values for multiple testing using the FDR to derive the $q$ value for each gene. Finally, we repeated this analysis but included only severe mutations (stop gain, start loss, frameshift and canonical splicing) to derive the probability of observing at least as many severe mutations as were actually present. The same analysis was performed on subsets of the whole cohort to compare the statistical significance across subcohorts. Mutations in selected genes were plotted on the corresponding proteins with annotations derived from the UniProt Knowledgebase (v2022_5; https://www.uniprot.org/; accessed 14 June 2022)[65].

### Analysis of evolutionary conservation of mutated nucleotides
PhyloP conservation tracks across 470 mammalian genomes were downloaded from UCSC[63,66]. Genes were divided into those that were non-significant ($q > 0.1$ in the human mutation dataset), significant in human data only ($q < 0.1$ in the human mutation dataset but $q > 0.1$ in the *piggyBac* dataset) and significant in both ($q < 0.1$ in both datasets). For each gene in the three groups, the median of the phyloP scores for all mutated nucleotides was calculated. The significant groups were compared with the non-significant group using a two-sided Mann–Whitney test, followed by FDR correction.

### Comparison of expression data to the TCGA database
SCLC RNA-seq data from two different studies[8,40] and RNA-seq data from neuroblastoma samples[67] were re-analysed using the TCGA pipeline. In brief, STAR v2.4.2a was used to align reads to the GRCh38 reference using GENCODE annotation v22. HTSeq v0.6.1p1 was then used to quantify expression at the gene level. Raw counts were converted to transcripts per million (TPM) using the median length of all transcripts for each gene, as reported in GENCODE annotation v22. TPM + 1 values were then log scaled and used for further analysis. Expression data were downloaded from the Genomic Data Commons Data Portal (https://portal.gdc.cancer.gov). The TPM values of SCLC samples were compared with the TPM values of individual types of tumours in TCGA using a two-sided Mann–Whitney test, and the fold change for each gene was calculated as the median of the SCLC $\log_2(TPM + 1)$ values minus the median of the TCGA cohort $\log_2(TPM + 1)$ values.

### Comparison of expression data to the GTEx database
SCLC RNA-seq data from two different studies[8,40] were re-analysed using GTEx pipeline v8. In brief, STAR v2.5.3a was used to align reads to the GRCh38 reference using GENCODE annotation v26. RNA-SeQC v1.1.9 was then used to quantify expression at the gene level. Raw counts were converted to TPM using the median length of all transcripts for each gene, as reported in GENCODE annotation v26. TPM + 1 values were then log scaled and used for further analysis. Expression data were downloaded from the GTEx database (https://gtexportal.org). Tissues with fewer than 30 samples were excluded (fallopian tube, bladder, cervix uterus). The TPM values of SCLC samples were compared with the TPM values of the individual tissues in GTEx using a two-sided Mann–Whitney test, and the fold change for each gene was calculated as the median of the SCLC $\log_2(TPM + 1)$ values minus the median of the GTEx tissue $\log_2(TPM + 1)$ values.

### snRNA-seq
Sucrose buffer (1 M; 1 M sucrose, 10 mM Tris-HCl (pH 8) and 3 mM magnesium acetate), lysis buffer 1 (5 mM CaCl$_2$, 3 mM magnesium acetate,

2 mM of 0.5 M EDTA (pH 8, RNase-free), 0.5 mM EGTA (ThermoFisher), 1× cOmplete, EDTA-free protease inhibitor cocktail (Sigma), 1 mM dithiothreitol (Roth), 0.1 mM phenylmethylsulfonyl fluoride (Roth) and 1.6 U ml$^{-1}$ mouse RNase inhibitor (NEB)), lysis buffer 2 (lysis buffer 1, 0.4% (v/v) Triton X-100 (Sigma) and 4 U ml$^{-1}$ mouse RNase inhibitor), lysis buffer 3 (lysis buffer 1 and lysis buffer 2 in a 1:1 ratio and 5.7 U ml$^{-1}$ mouse RNase inhibitor) and resuspension buffer (D-PBS with MgCl$_2$ and CaCl$_2$ plus 12 U ml$^{-1}$ mouse RNase inhibitor) were prechilled on ice for at least 1 h before isolation. Snap-frozen RP tumours were thawed in a 60-mm dish on ice and sharply minced with a precooled scalpel. Subsequently, the minced tissue was transferred to a gentleMACS M-tube (Miltenyi) and the 60-mm dish was rinsed with lysis buffer 1, which was then added to the gentleMACS M-tube. The tissue was dissociated using programme 'Protein-M-tube 1.0' of gentleMACS (Miltenyi). Lysis buffer 2 was added to the M-tube, followed by inversion. The lysed tissue was filtered through a 40-µm cell strainer prewetted with lysis buffer 1. Centrifugation (5 min, 450$g$, 4 °C, with breaks; Eppendorf) was conducted to pellet the nuclei. Next, the supernatant was discarded and the nuclei were resuspended in lysis buffer 3 and kept on ice. Sucrose buffer was drawn into a 25-gauge needle and syringe and ejected underneath the nuclear suspension, followed by centrifugation (5 min, 450$g$, 4 °C, with breaks). The upper phase was removed, and the nuclei were gently resuspended in resuspension buffer and filtered through a 15-µm cell strainer. Fixation of the nuclei, barcoding of single nuclei, amplification of barcoded cDNA and preparation of sequencing libraries were carried out according to the Evercode WT Mega v2.1.1 user manual (Parse Biosciences). Libraries were sequenced at the Cologne Center for Genomics using an Illumina NovaSeq 6000 instrument at an average depth of 216,310,743.5 reads per sample.

### Processing of mouse snRNA-seq data
Raw sequencing data were processed using Parse scRNA-seq pipeline v1.1.1, which included alignment to the GRCm39.110 reference using STAR v2.7.10b and demultiplexing of cells to the corresponding samples based on the first barcode. The resulting raw count matrices and cell annotation files, together with the Ensembl GRCm39.110 gene annotations, were assembled into an Anndata object using scanpy v1.9.3. Cell detection and background removal were performed using Cellbender v0.3.0 with standard settings. Doublet filtering was performed using doubletdetection v4.2 with a voter threshold of 0.5 and a $P$-value threshold of 0.001. Low-quality cells were filtered out using a two-step protocol. First, we excluded cells that had fewer than 25 protein-coding genes with at least 3 raw counts. Then, we log scaled four quality-control metrics and calculated the median and the median absolute deviation (MAD) for each. These metrics included the percentage of counts mapped to mitochondrial transcripts, the percentage of counts mapped to ribosomal transcripts, the percentage of counts included in the top ten most expressed genes and the total number of protein-coding genes. We excluded cells that had a value greater than 3 MAD from the median for each of these metrics, as well as cells with a value lower than 3 MAD from the median for the total number of genes. We also excluded genes that were not protein coding and genes that were not expressed in any cell. For clustering and visualization, the remaining counts were converted to transcripts per 10,000 (tp10k) by dividing by the median length of the transcripts for each gene and normalizing to 10,000 using scanpy.pp.normalize_total with the option to exclude highly expressed genes. The tp10k values were converted to a $\log_{10}(tp10k + 1)$ scale, and the most variable genes were selected using scanpy.pp.highly_variable_genes with standard settings. Principal-component analysis was performed using 100 components, and these were batch corrected with Harmony using scanpy.pp.harmony_integrate with standard settings. Neighbours were calculated using scanpy.pp.neighbors with 100 neighbours and using the cosine distance. Leiden clusters were calculated using scanpy.tl.leiden with a resolution of 0.5. Coarse connectivity of the manifold

was calculated using PAGA with scanpy.tl.paga and used as the starting point for uniform manifold approximation and projection (UMAP) embedding with scanpy.umap using standard settings. Markers of expected cell types were identified in the literature and used for cell type calling at the cluster level.

## UMAP visualization of gene sets in mouse scRNA-seq data
The sum of the $\log_{10}(\text{tp10k} + 1)$ values was calculated for included genes in the gene sets 'glutamatergic synapse' and 'synaptic membrane' and normalized and clipped to the range 0–1, with 0 being the mean score of the cluster with the lowest score and 1 being the mean score of the cluster with the highest score.

## Re-analysis of scRNA-seq data from patients with SCLC
Published scRNA-seq data[44] were obtained from https://cellxgene.czi-science.com/collections containing preprocessed gene expression values, annotations of cell types, SCLC subtypes and UMAP embeddings. Samples marked as NSCLC were excluded. Individual cells marked as neuroendocrine or NSCLC were also excluded.

## Gene set analysis with GO
The GO architecture and annotations were downloaded from the GO website (v2020-09-10; http://geneontology.org)[68,69]. For each term annotation of each gene, the gene was additionally annotated with all its parent terms. For each dataset of interest, the identified genes were compared to all GO terms that included at least 10 and at most 1,000 genes using a two-sided Fisher's exact test and FDR correction. *PiggyBac* hits ($n$ = 504) and human mutation hits ($n$ = 991) were included if they had a $q$ value of less than 0.1 and at least one GO annotation. Genes highly expressed in SCLC were first filtered to include only genes with at least one GO annotation and with a $q$ value of less than 0.1 in at least 90% of the comparisons (30/33 tumours or 25/27 healthy tissue samples). The remaining genes were then ranked by the median $\log_2$-transformed fold change across all comparisons and only the top 1,000 genes were included. Genes from the mouse snRNA-seq and re-analysed human scRNA-seq datasets[44] were selected by comparing the pseudobulk counts of SCLC cells to the pseudobulk counts of other cells using a two-sided Fisher's exact test followed by FDR correction. Genes were included if they had a $q$ value of less than 0.1 in the majority of the samples and a median fold change of at least 2. The remaining genes were ranked by median fold change and the top 1,000 genes were included in the analysis. Force-directed graphs were generated with datashader v0.12.1 using the ForceAtlas2 layout. For this analysis, up to 100 GO terms were included as nodes if they were significantly enriched ($q$ < 0.1) for genes in the datasets and if they were not a perfect subset or overset of a GO term with a more significant overlap. An edge was present between two GO terms if the genes included in the terms significantly overlapped ($q$ < 0.1 by two-sided Fisher's exact test and FDR correction). GO terms identified at the expression level both versus cancer types and versus healthy tissue types were further cross-referenced with ChIP–seq data downloaded from the CISTROME database (http://cistrome.org/db; accessed 27 November 2019)[42] and with scRNA-seq data from healthy human lung samples from ref. 43. The ChIP–seq peaks from experiments using antibodies against RB1, RBL2, E2F1, E2F2, E2F3, E2F4 and E2F5 were downloaded from the CISTROME database to derive an RB–E2F score. For each gene, the peaks were merged across samples and replicates and their fold change over background was added across samples and replicates. Peaks with a total fold change of at least 10 were included in the analysis. The regulatory potential was calculated for all target genes whose TSS was within 100 kb of the peak, using the distance between the peak and the TSS, as described in ref. 70. The regulatory potential was multiplied by the total fold change, and this score was added for all peaks near a TSS. For each target gene, the transcript with the highest score was kept. The scores for each transcription factor were normalized between 0 and

1 and then added together to derive the RB–E2F score for each target gene. The score for each enriched GO term was calculated as the mean score across genes included in the GO term and in the SCLC dataset. scRNA-seq data, as well as the corresponding metadata from ref. 43, were downloaded from Synapse (Synapse:syn21560406). Cell type annotations were obtained from the metadata. Cells were divided into two groups: cells annotated as neuroendocrine and all others. Unique molecular identifiers in each group were added and converted to TPM. The TPM + 1 values were then log scaled, and the $\log_2$-transformed fold change between PNECs and other lung-resident cells was calculated as the difference in the two log-scaled values for each gene. For each GO term, we calculated the mean fold change for genes included in the GO term and in the SCLC dataset.

## Neuroendocrine and 'glutamatergic synapse' expression scores
The neuroendocrine score was calculated using the 50 marker genes identified in ref. 50 as the correlation between the log ratio described in the publication and the expression levels of the genes in individual SCLC samples. To calculate the expression score for genes in the GO term 'glutamatergic synapse', the log-scaled expression values of the genes in the gene set were first normalized between the median of the tumour type or tissue with the highest expression (normalized to 1) and the median of the tumour type or tissue with the lowest expression (normalized to 0). The score was then calculated as the mean of the normalized expression for all genes in the gene set.

## Virus production
Retroviruses encoding DsRedExpress2 and those encoding the RABV glycoprotein and TVA800 (the glycosylphosphatidylinositol-anchored form of the TVA receptor), as well as GFP-encoding EnvA-pseudotyped RABV, were described previously[71].

## Cell line maintenance
SCLC and NSCLC cell lines were maintained in culture in RPMI 1640 (Life Technologies) supplemented with 10% fetal bovine serum (FBS; Gibco) and 1% penicillin/streptomycin (Gibco).

## Isolation of mouse cortical neurons
Mouse embryos (embryonic day 13.5–16.5) were isolated following cervical dislocation of the anaesthetized pregnant mother as previously described[72]. In brief, cortices were dissected in Hank's buffered salt solution (Gibco) supplemented with HEPES (10 mM; Gibco), and dissociated by means of enzymatic digestion for 15 min at 37 °C by incubating the tissue in DMEM high-glucose GlutaMAX (Gibco) containing papain (20 U ml$^{-1}$; Merck) and cysteine (1 µg ml$^{-1}$; Merck), followed by mechanical trituration in medium supplemented with 10% FBS (Gibco).

## Generation of human cortical neurons
Neurons were grown for at least 4 weeks before using them in co-cultures. Human cortical neurons were derived from the WTC11 human iPS cell line carrying a doxycycline-inducible *Ngn2* transgene[73] and were cultivated as previously described[74]. In brief, iPS cells were cultured on GelTrex-coated plates (1×; ThermoFisher Scientific) in StemMACS iPS-Brew XF (Miltenyi). When reaching confluence, the cultures were passaged with Versene passaging solution (ThermoFisher Scientific) and seeded in thiazovivine (Axon Medchem)-supplemented iPS-Brew for 1 day. Cells were grown at 37 °C and 5% CO$_2$ in a humidified incubator. Differentiation into cortical neuronal cultures was performed by seeding iPS cells at high density onto GelTrex-coated plates using predifferentiation medium supplemented with thiazovivine. The predifferentiation medium was replaced daily for the following 2 days with thiazovivine-free predifferentiation medium. Cells were then seeded onto poly(D-lysine) (Sigma-Aldrich)- and laminin (Trevigen)-coated plates using maturation medium supplemented with 1:100 GelTrex. Half of the medium was exchanged once per week until analysis.

## Nodose ganglion explant cultures

Wild-type C57BL/6 mice (3–5 weeks old) were killed by cervical dislocation, and nodose ganglia were isolated using an intracranial approach[75]. The top of the skull was removed, followed by extraction of the brain and brainstem to expose the base of the skull. Under stereomicroscopic visualization, a midline incision was made into the occipital bone plate, extending rostrally from the foramen magnum. The occipital bone plate was then detached from the temporal bone to expose the vagus nerve and its associated nodose ganglion. Following isolation, surrounding tissues and the ganglion capsule were carefully removed using fine scissors and forceps. The isolated ganglia were then plated in 96-well plates (Sarstedt) precoated with collagen I (Ibidi) and containing Neurobasal-A medium (Gibco), supplemented with 2% FBS (Gibco), 2% B27 supplement (Gibco), 1% penicillin/streptomycin (Gibco), 0.5 mM GlutaMAX (Gibco), 25 µM L-glutamate (Sigma), 50 ng ml$^{-1}$ nerve growth factor (Alomone Labs), 20 ng ml$^{-1}$ glial cell line-derived neurotrophic factor (PeproTech) and 20 ng ml$^{-1}$ brain-derived neurotrophic factor (PeproTech). Explant cultures were maintained at 37 °C and 5% $CO_2$ throughout the experiment. Medium changes were performed once per week. Twelve days after plating, explant cultures were examined under a microscope to evaluate attachment and extension of neurites. Only explants exhibiting neurite outgrowth were used for subsequent co-culture experiments.

## Monitoring of proliferation in cell culture

Proliferation was assessed using IncuCyte live-cell imaging. Cancer cells were stably transduced using lentiviral vectors carrying an EF1α-tdTomato-IRES-G418 transgene. For co-culture experiments, 30,000 fibroblasts or neurons were plated per well. For analysis of high-density monocultures, 3,000 non-fluorescent cells were plated per well. Conditioned medium was collected from neuronal cultures and filtered through 0.2-µm filters. Two thousand cancer cells were added to each well and transferred into the IncuCyte system 1 day after initiation of co-culture. Whole-well images (×4 objective) were captured every 6 hours for a total of 6 days. Bright-field and fluorescence channels were acquired (557 nm, 607 nm). The captured images were analysed using IncuCyte analysis software (Sartorius) to quantify total integrated intensity as a measure of cell proliferation. As a control, cells in monoculture were plated in separate wells on the same plate and were maintained under identical culture conditions. The intensity was normalized to the intensity of the first scan, and the AUC was calculated over 5 days of culture. The AUC was further normalized to the AUC of monocultures in the same plate. In Extended Data Fig. 10a, all conditions were compared to the co-cultures with cortical neurons using a two-sided Mann–Whitney test followed by FDR correction. In Fig. 5, the median normalized AUC values of SCLC cell lines were compared between co-cultures with cortical neurons and cultures in neuron-conditioned medium using a two-sided Wilcoxon signed-rank test. For visualization of nodose fibres, the ganglia were transduced with a peripheral nervous system-specific AAV encoding tdTomato (AAV-PHP.S-hSyn-tdTomato-P2A-APEX2-V5; VectorBuilder) and seeded into 96-well plates. Two thousand COR-L88 cells were added once neurites started to form, and cultures were monitored for 5 days at ×4 resolution over the whole well.

## RABV tracing

Neurons were plated at a density of 60,000–70,000 cells per coverslip (24-well plate) on poly(L-lysine) (0.1 mg ml$^{-1}$; Merck)-coated glass coverslips. After 4 h, the medium was replaced with Neurobasal serum-free medium (Gibco) containing B27 supplement (1%; Gibco) and GlutaMAX (0.5 mM; Merck). Neurons were then maintained at 37 °C and 5% $CO_2$ throughout the experiment and semi-feeding was performed once per week. SCLC cells were transduced with retroviruses encoding DsRed alone, DsRed and TVA, or DsRed, TVA and G and preconditioned in

Neurobasal medium for 48 h before seeding of 500 cells onto the neuronal layer. At neuronal division 12 (DIV12)–DIV13, EnvA-pseudotyped (ΔG) RABV encoding GFP was added to the medium. Analysis was performed after an additional 3 days in culture, at which time samples were fixed in paraformaldehyde (PFA; 4% in PBS; Sigma) and the number of GFP-only-positive presynaptic neurons was quantified and normalized to the number of double-positive starter SCLC cells. For quantification, a confocal Stellaris microscope (Leica Microsystems) equipped with a ×10 air objective was used to acquire two distinct ROIs of 3 × 3 tiles randomly chosen within the coverslip, and two coverslips for each condition were examined per biological replicate. One embryo preparation obtained from a pregnant mouse was considered to be a biological replicate. Within each ROI, GFP-positive neurons were manually counted using the plugin Cell Counter v3.0.0 for ImageJ v1.54h, and their numbers were normalized to those of DsRed and GFP double-positive starter cancer cells. A minimum of four biological replicates per condition were used for quantification. The $q$ value was calculated by comparing each condition to the full experimental system (DsRed, TVA, G) using a two-sided Mann–Whitney test followed by FDR correction. For time-lapse imaging, COR-L88 cancer cells were transduced with a DsRed retrovirus encoding G and TVA, followed by addition of RABV-GFP 2–3 days later. After 24 h, cancer cells were washed thoroughly with PBS, trypsinized and plated onto DIV12 neuronal cultures. The resulting co-cultures were placed in an IncuCyte live-cell analysis system (Sartorius) for the subsequent 72 h. For monosynaptic tracing experiments of grafted cancer cells in vivo, a suspension of retrovirally transduced mouse SCLC cells derived from the RP model and freshly added EnvA-pseudotyped (ΔG) RABV-GFP were infused into the hippocampus of mice as previously described[76]. After mice were killed, GFP-positive neurons and cancer starter cells double positive for DsRed and GFP were quantified to obtain a connectivity ratio per starter cell. Neurons were classified according to their morphology and location within the layers of the hippocampus, dentate gyrus and subiculum. Occasional glial cells exhibiting morphological hallmarks of astrocytes or oligodendrocytes were excluded from the analysis. The $P$ value was calculated using a two-sided Mann–Whitney test.

## EdU chase assays

To compare the proliferation rates of monocultures and co-cultures with immunofluorescence staining, cultures of COR-L88 cells were treated with 20 µM EdU and incubated for 2 h before fixation with 4% PFA prewarmed at room temperature for 10 min. Cells were washed three times with 1× PBS followed by EdU staining using the Click-iT EdU imaging kit (Invitrogen). $P$ values were calculated with a paired two-sided $t$ test.

## Transplantation experiments

*Thy1*-GFP-M mice[77] were anaesthetized by intraperitoneal injection of a ketamine/xylazine mixture (100 mg kg$^{-1}$ ketamine and 10 mg kg$^{-1}$ xylazine), injected subcutaneously with carprofen (5 mg kg$^{-1}$) and fixed in a stereotactic frame provided with a heating pad. A portion of the skull covering the somatosensory cortex (from bregma: caudal, −2.0; lateral, 1.5) was thinned with a dental drill, avoiding disturbing the underlying vasculature, and a small craniotomy sufficient to allow penetration of a glass capillary was performed. A finely pulled glass capillary containing a suspension of mouse SCLC cells derived from the RP model in sterile PBS was then inserted through the dura to reach the hippocampus, and an estimated total of about 30,000–50,000 cells (corresponding to a total injected volume of 0.8–1.0 µl) were slowly infused using a manual syringe (Narishige) in multiple vertical steps spaced by 50 µm (−1.9 to −1.3 from bregma) over a total duration of 10–20 min. After capillary removal, the scalp was sutured and mice were placed on a warm heating pad until fully recovered. The physical condition of the animals was monitored daily before they were killed 10–12 days after surgery.

## Fluorescence immunostaining of brain slices and co-cultures

Immunostaining of fixed brain slices and cultures (Figs. 2 and 4 and Extended Data Figs. 6a,e, 7a and 9) was performed using conventional procedures described previously[72]. The following primary antibodies were used: chicken anti-GFP (1:500; Aves Labs, GFP-1020), rabbit anti-RFP (1:500; Rockland, 600401379), chicken anti-MAP2 (1:500; Abcam, ab5392), mouse anti-VGluT1 (1:500; Synaptic Systems, 135311). rabbit anti-Homer (1:500; Synaptic Systems, 160003) and mouse anti-BSN (1:500; Synaptic Systems, 141111). The following secondary antibodies were used (raised in donkey): Alexa Fluor 488-conjugated secondary antibody anti-chicken (1:1,000; Jackson Immuno-Research, 703-545-155), Alexa Fluor 546-conjugated secondary antibody anti-rabbit (1:1,000; ThermoFisher Scientific, A10040), Alexa Fluor 647-conjugated secondary antibody anti-rabbit (1:500; Jackson ImmunoResearch, 711-605-152), Alexa Fluor 647-conjugated secondary antibody anti-mouse (1:1,000; Jackson ImmunoResearch, 715-605-150) and DyLight™ 405-conjugated secondary antibody anti-rabbit (1:100; Jackson ImmunoResearch, 711-475-152). Images were acquired using an SP8 confocal microscope (Leica) equipped with a ×20 (NA 0.75), ×40 (NA 1.3), ×63 (NA 1.4) or ×100 (NA 1.3) oil-immersion objective and further processed with Fiji v2.14.0.

## Imaging of co-cultures with cortical neurons

Co-cultures of mouse cortical neurons and human SCLC cells (Fig. 3 and Extended Data Fig. 6b–d) were fixed with 4% PFA and quenched with 50 mM glycine for 10 min and were immunostained as described in ref. 78. In brief, samples were permeabilized and blocked with 0.2% Triton X-100, 2.5% bovine serum albumin (BSA), 2.5% normal goat serum (NGS) and 2.5% donkey serum in PBS for 30 min and then washed with 2.5% NGS in PBS. Samples were incubated with primary antibodies (chicken anti-MAP2 (1:1,000; Novus Biologicals, NB300-213), mouse anti-mNeonGreen (1:500; ChromoTek, 32F6), rabbit anti-HOMER1 (1:500; Synaptic Systems, 160003), mouse anti-BSN (1:500; Enzo, ADI-VAM-PS003-F, SAP7F407), mouse anti-SMI-312 (1:1,000; HISS Diagnostics, SMI-312R), guinea pig anti-VGluT1 (1:500; Synaptic Systems, 135304), rabbit anti-VGluT1 (1:500; Synaptic Systems, 135308) and mouse anti-VGluT1 (1:500; Synaptic Systems, 135011)) for 1.5 h at room temperature, washed with 2.5% NGS in PBS and incubated with secondary antibodies (Alexa Fluor 405-conjugated anti-chicken (1:500; Abcam, ab175674), Alexa Fluor 568-conjugated anti-mouse (1:1,000; Life Technologies, A-11004), STAR 635P-conjugated anti-rabbit (1:1,000; Abberior, 1002-500UG) and Alexa Fluor 750-conjugated anti-guinea pig (1:500; Abcam, ab175758)) for 45 min. Samples were washed five times with 0.2% Triton X-100, 2.5% BSA, 2.5% NGS and 2.5% donkey serum in PBS with gentle shaking and two times with PBS. They were then washed with double-distilled water, before mounting with Prolong Glass for non-expanded specimens. A Stellaris 8 PP STED Falcon microscope (Leica Microsystems) was used for confocal, 3D STED, and 2D and 3D ONE microscopy imaging[46]. Confocal overview images were acquired using the navigator function spiral mode scan, and tiles were stitched with 12% overlap. An HC PL apo ×100/1.4 NA oil STED W objective was used for all imaging modalities. The white-light laser was used as the main excitation source, tuned to the best excitation wavelength for each fluorophore, at a pulse frequency of 80 MHz. Blue-shifted dyes were excited using a separate 405-nm DMOD laser. 3D STED images were acquired using a theoretical pixel size set between 20 and 37 nm. Three STED depletion beams, at 775, 660 and 592 nm, were used at a repetition rate of 80 MHz with more than 1.5 W of output power together with a 50-nm $xy$ vortex donut and an 130-nm $z$ donut. Near-infrared and/or far-red emissions were detected using a Power HyD R SP detector, red-shifted emissions were detected using Power HyD S SP Core Unit detectors, green-shifted emissions were detected using Power HyD X SP detectors and blue-shifted emissions were detected using HyD S SP detectors, in the presence of the respective notch filter set STED 3X.

The detectors were set to either counting intensity or counting τSTED mode. 3D reconstructions were done through the 3D viewer in LAS-X. ONE microscopy images were acquired using a 12-kHz tandem scanner with and without dynamic signal enhancement (between 5 and 11, weighing of 0.4). The theoretical acquired pixel size was set to 92 nm, which yielded a final computed pixel size of 0.92 nm after computation with 32-bit image depth. Two thousand frames per channel were acquired for 2D ONE images. Three hundred to 500 frames per channel were acquired for 3D ONE images. The resultant images were processed with TRAC4, radiality magnification of 25 or TRA mode, as described in refs. 46,79. A TCS SP5 STED microscope (Leica Microsystems) was used for 2D ONE microscopy, using 488-, 561- and 633-nm laser lines and an HCX Plan apo STED ×100/1.4 NA oil-immersion objective. Images were acquired using an 8-kHz resonant scanner in unidirectional line scan mode, and emission was detected using HyD and PMT detectors. The theoretical pixel size was set to 98 nm and the image bit depth was set to 8 bits, with a line format of 128 × 128 and a frame count ranging between 1,000 and 2,000 frames.

## Expansion microscopy

Samples were expanded following the x10ht protocol as described in ref. 80. In brief, specimens were anchored overnight at 4 °C with 0.3 mg ml$^{-1}$ Acryloyl-X (SE; ThermoFisher Scientific, A-20770) in PBS (pH 7.4). Gel monomer solution was added onto the samples, which were later homogenized by application of disruption buffer and autoclaving for 60 min at 110 °C. The samples were then expanded by adding double-distilled water to 22 ×22 cm$^2$ square culture dishes.

## Imaging of mouse nodose ganglion neuron and human SCLC co-cultures

Co-cultures of mouse nodose ganglia and human COR-L88 cells (Extended Data Fig. 6f–i) were fixed with 4% PFA and quenched for 30 min with 100 mM ammonium chloride. They were then permeabilized and blocked with 0.2% Triton X-100, 2.5% BSA, 2.5% NGS and 2.5% donkey serum in PBS for 30 min, before washing with 2.5% NGS in PBS. Samples were incubated with primary antibodies (chicken anti-MAP2 (1:1,000; Novus Biologicals, NB300-213), mouse anti-mNeonGreen (1:500; ChromoTek, 32F6), rabbit anti-HOMER1 (1:500; Synaptic Systems, 160003), alpaca anti-VGluT1 nanobody (1:500; Nanotag, N1602-AF568-L, conjugated to AZDye 568), mouse anti-SMI-312 (1:1,000; HISS Diagnostics, SMI-312R) and mouse anti-SMI311 (1:1,000; Biozol, BLD-837801)) for 1.5 h at room temperature, washed with 2.5% NGS in PBS and incubated with secondary antibodies (Alexa Fluor 405-conjugated anti-chicken (1:500; Abcam, ab175674), STAR 635P-conjugated anti-rabbit (1:1,000; Abberior, ST635P, 1002-500UG) and Alexa Fluor 750-conjugated anti-mouse (1:1,000; ThermoFisher, A21037)) for 45 min. The samples were washed five times with 0.1% Triton X-100, 2.5% BSA, 2.5% NGS and 2.5% donkey serum in PBS with gentle shaking and two times with PBS. They were then washed with double-distilled water, before mounting with Prolong Glass. A Stellaris 8 PP STED Falcon microscope was used for confocal and 3D τSTED Xtend imaging.

## Imaging of mouse autochthonous SCLC tumours in tissue sections

For Extended Data Fig. 7b–d, the lungs of tumour-bearing RPC mice induced with CGRP-driven Cre were snap frozen in liquid nitrogen using OCT medium and sectioned at a thickness of 10 μm with a Leica CM3050 S cryotome. Slices were fixed in 4% PFA and quenched for 30 min with 50 mM glycine. They were then washed three times with PBS + iT-Fx image signal enhancer for 20 min. Samples were permeabilized and blocked with 0.3% Triton X-100, 2.5% BSA, 2.5% NGS and 2.5% donkey serum in PBS for 45 min and washed twice in 2.5% NGS in PBS for 5 min each. Specimens were stained with rabbit anti-HOMER1 (1:500; Synaptic Systems, 160003), alpaca anti-VGluT1 nanobody

(1:500; Nanotag, N1602-AF568-L, conjugated to AZDye 568), mouse anti-SMI-312 (1:1,000; HISS Diagnostics, SMI-312R), mouse anti-SMI311 (1:1,000; Biozol, BLD-837801) and anti-GFP nanobody Alexa Fluor 488 (1:500; Nanotag, N0301) for 3 h and 45 min in 2.5% NGS in PBS. Samples were washed three times with 0.1% Triton X-100, 2.5% BSA, 2.5% NGS and 2.5% donkey serum in PBS for 10 min and then stained with secondary antibodies for 1.5 h using STAR 635P-conjugated anti-rabbit (1:1,000; Abberior, ST635P, 1002-500UG), Alexa Fluor 750-conjugated anti-mouse (1:1,000; ThermoFisher, A21037). The samples were then washed five times with 0.1% Triton X-100, 2.5% BSA, 2.5% NGS and 2.5% donkey serum in PBS for 10 min each and twice with PBS for 15 min each. To label the tissues with pan-NHS-ester labelling, the specimens were washed with sodium bicarbonate buffer, stained with NHS-ester Pacific Blue (1:5,000, BroadPharm, 215868-33-0) in sodium bicarbonate buffer for 15 min, washed four times with PBS and once with double-distilled water, and mounted using Aqua-Poly Mount, before imaging with the microscope indicated in the previous section.

### Automated analysis of HOMER1–VGluT1 co-localization

For the analysis in Extended Data Figs. 6h,i and 7c,d, 3D stacks were reduced to 2D summed images. HOMER1 images were subjected to an automatic threshold, equal to the mean and the standard deviation of the fluorescence signal, which found the spots above background. Signals corresponding to background noise were removed using an automated erosion procedure. The centres of mass of the remaining signals (true HOMER1 spots) were determined, and vertical and horizontal line scans were generated through the centres of mass. The vertical and horizontal line scans were averaged for every spot. The correlation of the scans was determined using the Pearson correlation coefficient, obtained using MATLAB (MathWorks, version 2023b).

### Fluorescence immunostaining of lung cryostat sections

RP and RPC mice were killed 2, 4 or 8 months after tumour induction. Lungs were fixed by intratracheal instillation with 4% PFA in 0.1 M phosphate buffer and processed to collect cryostat sections[14]. Immunostaining was performed as previously described for mouse lungs[14]. The following primary antibodies were used: goat anti-CGRP (1:1,000; Abcam, ab36001), rabbit anti-GAP43 (1:2,000; Novus Biologicals, NB300-143), chicken anti-GFP (1:500; Abcam, 13970), rabbit anti-PGP9.5 (1:2,000; Abcam, ab108986), rabbit anti-P2X3 (1:1,000; Chemicon, AB5895), rat anti-SP (1:200; Biogenesis, 8450-0505), guinea pig anti-SYP (1:4,000; Synaptic Systems, 101002) and rabbit anti-VGluT1 (1:250; Synaptic Systems, 135303). The following secondary antibodies were used (raised in donkey): Alexa Fluor 647-conjugated anti-chicken (1:400; Jackson ImmunoResearch, 703-605-155), Cy3-conjugated anti-rabbit (1:2,000; Jackson ImmunoResearch, 711-167-003), Cy3-conjugated anti-goat (1:400; Jackson ImmunoResearch, 705-165-147), Cy3-conjugated anti-guinea pig (1:400; Jackson ImmunoResearch, 706-165-148), FITC-conjugated anti-rabbit (1:500; Jackson ImmunoResearch, 711-095-152) and FITC-conjugated anti-goat (1:500; Jackson ImmunoResearch, 705-095-147). To enhance staining intensity, biotinylated secondary antibodies (1:500; Jackson ImmunoResearch, 711-065-152 and 1:200; Jackson ImmunoResearch, 712-065-150) were combined with FITC (1:1,000; Jackson Immuno-Research, 016-010-084)- or Cy3 (1:6,000; Jackson ImmunoResearch, 016-160-084)-conjugated streptavidin. Confocal images were acquired using a microlens-enhanced dual-spinning-disk confocal microscope (UltraVIEW VOX, PerkinElmer) equipped with 488-nm, 561-nm and 640-nm diode lasers for excitation of FITC, Cy3 and Alexa Fluor 647, respectively.

### Immunohistochemistry of human samples

Patients consented to the use of their tissue specimens, and approval was obtained by the ethics committee of the University of Cologne (Biomasota 13-091, 2016). Formalin-fixed and paraffin-embedded tissue sections of human SCLC tumours were deparaffinized and immunohistochemically stained according to standard protocols using an automated immunostainer and a horseradish peroxidase-based detection system with diaminobenzidine as the chromogen. Primary mouse monoclonal antibodies were directed against SYP (1:100; Leica Biosystems, PA0299), neurofilament, 200-kDa subunit (NF-H) (1:500; Sigma, N0142) and neurofilament, 70-kDa subunit (NF-L) (1:500; Agilent, M0762). Immunostained sections were counterstained with haemalum.

### Transmission electron microscopy

Anaesthetized mice were transcardially perfused with a fixative solution containing 4% formaldehyde and 2.5% glutaraldehyde in 0.1 M cacodylate buffer. The lungs were isolated and cut into 1-mm-thick sagittal sections, and the examined area was dissected according to the location of the tumour mass. Epon embedding was performed and ultrathin sections were prepared using standard procedures[81]. Electron micrographs were taken with a JEM-2100 Plus transmission electron microscope (JEOL) equipped with a OneView 4K 16-bit camera (Gatan) and DigitalMicrograph v3.32.2403.0 software (Gatan). For analysis, electron micrographs were acquired with a digital zoom of ×5,000 or ×6,000.

### CLEM in vivo and in vitro

Mice were perfusion fixed with electron microscopy-grade 4% formaldehyde in PBS. For brain tissue, 100-μm sections (obtained with a Leica vibratome) were stained for nuclei with DAPI (ThermoFisher; 3 μM) and placed into imaging dishes with a glass bottom (Ibidi) filled with PBS. Co-cultures were directly grown in glass-bottom dishes (MatTek, P356-1.5-14-C) coated with a carbon finder pattern using a mask (Leica, 16770162) and an ACE 200 carbon coater (Leica). Cells were fixed for 10 min with electron microscopy-grade 4% formaldehyde in PBS. z stacks of the ROI showing reporter-positive tumour cells (DsRed or tdTomato) were acquired using an SP8 confocal microscope (Leica). After confocal and bright-field imaging, samples were prepared for transmission electron microscopy using standard protocols. In brief, post-fixation was applied using 1% osmium tetroxide (Science Services) and 1.5% potassium hexacyanoferrate (Merck) for 30 min at 4 °C. After three 5-min washes with double-distilled water, samples were dehydrated using an ascending ethanol series (50%, 70%, 90%, 100%) with 10 min at each step. Infiltration was carried out with a mixture of 50% epon in ethanol for 1 h, 70% epon in ethanol for 2 h and 100% epon overnight (Merck). After incubation with fresh epon for 4 h, vibratome sections were mounted onto empty polymerized epon blocks and covered with Aclar foil to provide a flat surface. For co-cultures, TAAB capsules (Agar Scientific) filled with epon were placed upside down onto the glass bottom. Samples were cured for 48 h at 60 °C. Aclar foil was removed by peeling it off, and the glass bottom was removed by alternating between putting the dish in boiling water and liquid nitrogen. Samples were trimmed to the ROI, which was previously acquired by confocal microscopy, using a diamond 90° trimming tool (Diatome). For orientation, stereotypic shapes of the hippocampus including the granule cell and molecular layers as well as the vasculature were used and matched to measurements obtained from confocal images of the same region. For cell culture, the carbon pattern was used to find the back ROI. Serial sections (300 nm) were cut using an UC6 ultramicrotome (Leica) and collected onto pioloform (Plano)-coated slot grids. Post-staining was performed with 1.5% uranyl acetate (Agar Scientific) for 15 min and Reynolds lead citrate (Roth) solution for 3 min. Electron micrographs were acquired using a JEM-2100 Plus transmission electron microscope (JEOL) operating at 200 kV and equipped with a OneView 4K camera (Gatan). Tomograms of ROIs were acquired using SerialEM v3.7.11 and reconstructed using IMOD v4.11.7 (ref. 82). Registration of images obtained by light (confocal) and electron microscopy was done using nuclei (with nucleoli) as fiducials with the plugin EC-CLEM v1.1.0.0 from ICY v2.5.2.0 software[83]. 3D reconstruction of identified synaptic

contacts was performed using Imaris v10.2.0 (Oxford Instruments). 3D-rendered volumes were generated from masks created through manual segmentation of pre- and postsynaptic compartments and synaptic vesicles using Microscopy Image Browser (MIB) software v2.84 (ref. 84).

### Electrophysiology of COR-L88 cells and allograft slices

Acutely isolated brains were sectioned into coronal slices (300-µm thick) by using a vibrating microtome (HM-650 V, ThermoFisher Scientific) filled with ice-cold carbogenated (95% $O_2$ and 5% $CO_2$) aCSF cutting solution (125.0 mM NaCl, 2.5 mM KCl, 1.25 mM sodium phosphate buffer, 25.0 mM $NaHCO_3$, 25.0 mM D-glucose, 1.0 mM $CaCl_2$ and 6.0 mM $MgCl_2$, adjusted to pH 7.4 and 310 to 330 mOsm). The obtained brain slices were transferred to a chamber containing aCSF recording solution (125.0 mM NaCl, 2.5 mM KCl, 1.25 mM sodium phosphate buffer, 25.0 mM $NaHCO_3$, 25.0 mM D-glucose, 4.0 mM $CaCl_2$ and 3.5 mM $MgCl_2$, adjusted to pH 7.4 and 310 to 320 mOsm). Slices were stored for at least 30 min to allow recovery before performing recording. All recordings were performed using a microscope stage equipped with a fixed recording chamber and a ×20 water-immersion objective (Scientifica). For ex vivo experiments, recordings were performed in aCSF recording solution. For in vitro experiments, SCLC cells (day 3–6 in mono- or co-culture) were used, and recordings were performed in extracellular solution (124.0 mM NaCl, 10.0 mM D-glucose, 10.0 mM HEPES-KOH (pH 7.3), 3 mM KCl, 2 mM $CaCl_2$ and 1 mM $MgCl_2$, adjusted to pH 7.4). Patch pipettes with a tip resistance of 5 to 10 MΩ were made from borosilicate glass capillary tubing (GB150-10, 0.86 mm × 1.5 mm × 100 mm; Science Products) with a horizontal pipette puller (P-1000, Sutter Instruments). The patch pipette was filled with internal solution (4.0 mM KCl, 2.0 mM NaCl, 0.2 mM EGTA, 135.0 mM potassium gluconate, 10.0 mM HEPES, 4.0 mM ATP(Mg), 0.5 mM guanosine triphosphate (GTP)(Na) and 10.0 mM phosphocreatine, adjusted to pH 7.25 and 290 mOsm (sucrose)). Recordings were performed with an ELC-03XS patch-clamp amplifier (npi electronic) controlled with Signal software (v6.0; Cambridge Electronic). Experiments were recorded with a sampling rate of 12.5 kHz. The signal was filtered with two short-pass Bessel filters that had cut-off frequencies of 1.3 and 10 kHz. Capacitance of the membrane and pipette was compensated by using the compensation circuit of the amplifier. All experiments were performed under visual control using an Orca-Flash 4.0 camera (Hamamatsu) controlled with Hokawo software (v2.8; Hamamatsu). SCLC cells were identified by expression of the cytosolic fluorescent protein DsRed or tdTomato. Cells were clamped at a holding potential of −30 mV after rupturing the membrane, and spontaneous activity was recorded for 5 min in whole-cell voltage-clamp mode. Synaptic inputs were isolated by adding the following blockers to the recording solution: CNQX (Sigma, C127; 10 µM), D-AP5 (Sigma, A8054; 20 µM), bicuculline (bicuculline-methiodide; Sigma, 14343; 100 µM), riluzole (Sigma; 10 µM) and TTX (1 µM). sPSCs were identified and measured with Igor Pro (v32 7.01; WaveMetrics) using a semiautomatic identification script.

### Electrophysiology of H524 cells

For in vitro experiments, SCLC cells (on day 3 in mono- or co-culture; see also above) were used, and recordings were performed under constant superfusion of oxygenated aCSF (130 mM NaCl, 1.25 mM $NaH_2PO_4$, 10 mM $NaHCO_3$, 10 mM D-glucose, 3.5 mM KCl, 2 mM $CaCl_2$ and 1 mM $MgCl_2$, adjusted to pH 7.4).

Patch pipettes with a tip resistance of 6–8 MΩ were made from borosilicate glass 3.3 capillary tubing with filament (outer diameter, 1.5 mm; inner diameter, 1.2 mm; length, 100 mm; Hilgenberg). The patch pipette was filled with a caesium-based internal solution (125 mM $CH_3CsO_3S$, 16 mM $KHCO_3$, 2.0 mM NaCl, 1 mM EGTA, 4.0 mM ATP(Mg), 0.3 mM GTP(Na) and 10 mM QX-314-Cl, adjusted to an osmolarity of 295 mOsm).

Voltage-clamp recordings were performed with an Axon Multi-Clamp 700B amplifier (Molecular Devices) and digitized by an Axon Digidata 1550B (Molecular Devices). Electrophysiological recordings were acquired using Clampex (v10.7.0.3; Molecular Devices). Experiments were recorded with a sampling rate of 20 kHz for spontaneous event recordings or 50 kHz for optogenetic experiments. The signal was filtered at 3 kHz. Pipette capacitance was compensated using the compensation circuit of the amplifier.

All experiments were performed under visual control using a Teledyne Moment camera (Teledyne Technologies) controlled with Micro-Manager software (v2.0). SCLC cells were identified by expression of the cytosolic fluorescent protein tdTomato.

Pharmacological receptor blockade was performed by bath application of D-AP5 (Sigma, A8054; 20 µM) and/or Gbz (MCE, HY-103533; 12.5 µM). The event frequency of spontaneous synaptic events was analysed using Clampfit v11.2.2.17 (Molecular Devices).

For optogenetic experiments, neuronal cultures were first transduced at DIV2 with a mixture of CMV-driven Cre-expressing AAV (Addgene, 105537-AAV9) and double-*loxP*-flanked ChR2–eYFP AAV (Addgene, 20298-AAV1). For optogenetic stimulation of co-cultured neurons expressing ChR2–eYFP, the microscope was equipped with a SOLIS-470C LED (470-nm peak; Thorlabs), triggered for 5 ms at approximately 50% peak power every 120 s. An average trace of three recordings was calculated and subsequently analysed. Synaptic event peak amplitude was analysed using Clampfit v11.2.2.17 (Molecular Devices). In cases where synaptic events were abolished, the peak current in the corresponding temporal region of the previous synaptic event was used.

### Preclinical SCLC mouse model

For preclinical experiments, we used the RP genetically engineered mouse model for SCLC, in which tumour formation is driven by Cre-inducible conditional *Rb1* and *Trp53* knockout, as previously described[25]. Tumours were induced and monitored with MRI as described above. Following tumour detection (minimum tumour size of 5 mm³), mice were randomly assigned to the treatment cohorts, with sample size determined by power analysis. Compound solutions were prepared and injected as follows: etoposide (Hexal) was administered on days 1, 2 and 3 of a 14-day cycle, intraperitoneally, at a concentration of 8 mg kg⁻¹. Cisplatin (Accord) was administered on day 1 of a 14-day cycle, intraperitoneally, at a concentration of 4 mg kg⁻¹. Riluzole (15 mg kg⁻¹) was dissolved in 10% DMSO, 40% PEG-300, 5% Tween-80 and 45% PBS and administered 5 days per week. DCPG (60 mg kg⁻¹) was dissolved in PBS and administered intraperitoneally for 5 days per week. Best response was calculated as the lowest percentage change measured from the last MRI scan before treatment, including only mice that had evaluable tumours at the first follow-up. The burden per mouse was calculated as the total sum of the volumes of individual tumours. Growth curves of the total burden of individual mice were linearly interpolated between scans. The median value of the interpolated curves was plotted for each day, as long as at least seven mice were alive in the treatment cohort. The interpolated value at five times the initial burden was used to calculate the time to fivefold burden, excluding mice that did not reach a fivefold burden. Data from preclinical experiments were analysed with blinding.

### Reporting summary

Further information on research design is available in the Nature Portfolio Reporting Summary linked to this article.

## Data availability

Reference genomes were downloaded from the Genomic Data Commons (TCGA GRCh38.d1.vd1, https://api.gdc.cancer.gov/data/254f697d-310d-4d7d-a27b-27fbf767a834), from Ensembl (https://www.ensembl.org, GRCm38.102 and GRCm39.110) and from GTEx (https://www.gtex-portal.org, Homo_sapiens_assembly38_noALT_noHLA_noDecoy_ERCC.

fasta). Gene annotations were downloaded from GENCODE (vM23 and v22, https://www.gencodegenes.org/). Orthology mapping was downloaded from the HGNC database (https://www.genenames.org/; downloaded 6 January 2020). Mutation data were downloaded from the supplementary tables of the referenced publications or from the CCLE website (Cell_lines_annotations_20181226.txt and CCLE_DepMap_18q3_maf_20180718.txt, https://portals.broadinstitute.org/ccle/). TCGA expression data were downloaded from the Genomic Data Commons Data Portal (v27, https://portal.gdc.cancer.gov). GTEx expression data were downloaded from the GTEx database (v8, https://gtexportal.org). GO data were downloaded from the GO website (v2020-09-10, http://geneontology.org). ChIP–seq data were downloaded from the CISTROME database (v2, http://cistrome.org/db; accessed 27 November 2019). scRNA-seq data, as well as the corresponding metadata, were downloaded from Synapse (Synapse:syn21560406, https://www.synapse.org/) and CZ CellXGene (https://datasets.cellxgene.cziscience.com/7a30310a-2239-4d84-b99e-a12456c2fe19.h5ad). PhyloP conservation tracks across 470 mammalian genomes were downloaded from UCSC (hg38.470way.phyloP, https://genome.ucsc.edu/). UniProt Knowledgebase annotations were downloaded from the UniProt website (v2022_5, https://www.uniprot.org). Raw sequencing data from the *piggyBac* screen and mouse snRNA-seq are available through the Sequence Read Archive (SRA; https://www.ncbi.nlm.nih.gov/sra) under accessions PRJNA1275653 and PRJNA1276342, respectively. A scanpy data object of the snRNA-seq dataset is available at Zenodo (https://doi.org/10.5281/zenodo.15647008)[85]. The full analysed data from our whole-genome analyses are available in the supplementary information tables. Source data are provided with this paper.

## Code availability

Python scripts generated in this study are available from GitHub (https://github.com/beleggia-lab/neuron-to-SCLC-synapses) and Zenodo (https://doi.org/10.5281/zenodo.15667860)[86].

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

**Acknowledgements** We are indebted to our patients, who provided primary material. We thank B. Göbel, J. Knufer, A. Florin, M. Müller, U. Rommerscheidt-Fuß, G. Piper and J. Klimek from the University Hospital Cologne for their outstanding technical support; M. Krasnow from Stanford University for scientific input and suggestions; the CECAD In Vivo Facility and the CECAD Imaging Facility in Cologne for use of the electron and light microscopes and the Cologne Center for Genomics for sequencing; the IT Center University of Cologne (ITCC) for use of the CHEOPS and RAMSES computing clusters. Stem cell work was performed at the iPS cell laboratory of the Center for Molecular Medicine Cologne (CMMC). This work was funded through the German Research Foundation (DFG) (SFB1399–grant no. 413326622 to H.C.R., J. George, R.K.T., M.L.S., S.v.K., M.P., M.F., H.G., R.B., F.B., C.F., L.W., H.W., J. Brägelmann and R.H.-H; SFB1218–grant no. 269925409 to M. Bergami and E.M.; SFB1430–grant no. 424228829 to H.C.R.; SFB1403–grant no. 414786233 to S.v.K. and H.W.; SFB1310–grant no. 325931972 to S.v.K. and J. Brägelmann; SFB1530–grant no. 455784452 to H.C.R., H.W., R.D.J. and S.v.K.; CECAD EXC 2030–grant no. 390661388 to M. Bergami and E.M.; SFB1451–grant no. 431549029 to M. Bergami; Co 260/6-1 to A.A.H. and K.-K.C.; SFB1286/A03 and RTG 2824 to S.O.R.; HA 8562/4-1 to R.H.-H.; FOR5504 HA 8562/5-1 to R.H.-H.; SFB1678 INST 216/1317-1 to R.H.-H.; SO1155/5-1 and GRK2338 (P13) to M.L.S.; grant nos. 496650118 JA 2439/5-1 and 496874193 JA2439/4-1 to R.D.J.; SFB1588–grant no. 493872418 to M.F.; grant no. 497777992 to J. George and M.P.; and GRK 3110/1 to J. George, M.P. and S.v.K.), the Deutsche Krebshilfe (1117240 to H.C.R.; 70113041 to H.C.R. and F.B.; 70116707 to F.B.; and the Mildred Scheel Nachwuchszentrum grant 70113307 to F.B. and J. Brägelmann), the German-Israeli Foundation for Research and Development (I-65-412.20-2016 to H.C.R.), the Else Kröner-Fresenius Stiftung (EKFS-2014-A06 to H.C.R. and 2016_Kolleg.19 to H.C.R. and R.D.J.), the Boehringer Ingelheim Stiftung (exploration grant to M. Bergami and E.M.), the European Research Council (ERC-StG-2015, grant no. 677844 to M. Bergami and grant agreement no. 835102 to S.O.R.), the Alzheimer Forschung Initiative (24019R) with kind support of the Stiftung Alzheimer Initiative (SAI) to M. Bergami, the German Ministry of Education and Research (BMBF e:Med Consortium InCa, grant nos. 01ZX1901 and 01ZX2201A to R.K.T., J. George, H.C.R., S.v.K., M.L.S. and M.P.), the Chan Zuckerberg Foundation (iNano) to S.O.R., the Alexander von Humboldt Foundation (postdoctoral award to H.W.), Cancer Research UK (A27323 to H.W.), the Wellcome Trust (214342/Z/18/Z to H.W.), the Medical Research Council (MR/S00811X/1 to H.W.), the Fritz Thyssen Foundation (10.22.1.010MN to R.H.-H.), the CMMC (JRG X to R.H.-H.), the Bruno-Helene-Joester Foundation (J. George and M.P.) and the Jean Uhrmacher Foundation (J. George). Additional funding was received from the programme 'Netzwerke 2021', an initiative of the Ministry of Culture and Science of the state of North Rhine-Westphalia for the CANTAR project to H.W., H.C.R., J. George, R.K.T., M.P., S.v.K., J. Brägelmann, R.D.J., M.F. and R.H.-H. We are grateful for support by the translational cancer programme of the German Cancer Aid TACTIC, project no. 70115201. The results shown here are in part based on data generated by the TCGA Research Network (https://www.cancer.gov/tcga). The data used for the analyses described in this article were obtained in part from the GTEx portal on 28 August 2020.

**Author contributions** M. Bergami, E.M., H.C.R. and F.B. designed and coordinated the overall study, with scientific advice from R.K.T., M.A. and S.O.R. H.C.R. is the lead corresponding author. R.R. and H.C.R. designed the *piggyBac* screen. A. Schmitt, J.W., R.M., C.M.B. and F.B. performed the *piggyBac* screen. R.R., H.C.R. and F.B. analysed *piggyBac* screen data. A. Schmitt, G.R. and R.B. performed immunohistochemical analysis. M.P., J. George, R.K.T. and F.B. designed the analysis of human genetic data. M.P., J. George and F.B. analysed human genetic data. M.F., R.K.T. and F.B. designed the analysis of human expression data. M.P., J. George and F.B. analysed human expression data. J.S. and F.B. designed the *Reln* knockout experiment. M.T., N.M.E. and G.S.G.B. designed and cloned CRISPR guides for the *Reln* experiment. G.G.H. performed the *Reln* knockout experiments. G.G.H., J.S. and F.B. analysed data from the *Reln* knockout experiment. M.S., J. Brägelmann, J. George and F.B. analysed human scRNA-seq data. R.H.-H., H.C.R. and F.B. designed snRNA-seq experiments. V.S., P.N., I.K. and P.H. performed snRNA-seq experiments. Y.L., P.N. and F.B. analysed snRNA-seq data. D.A., I.B., A.H.S., S.O.R., H.C.R. and F.B. designed immunofluorescence experiments in mouse lung tumours. A.H.S., I.P., I.B. and S.O.R. performed immunofluorescence analysis of mouse lung tumours. A.H.S., D.A., I.B. and S.O.R. analysed immunofluorescence data from mouse lung tumours. M. Bergami and E.M. designed immunofluorescence experiments with brain-engrafted cancer cells. F.O., K.N., M. Bergami and E.M. performed immunofluorescence

experiments with brain-engrafted cancer cells. A.H.S., S.O.R., M. Bergami, E.M., H.C.R. and F.B. designed immunofluorescence experiments with cultured cells. F.O., K.N., A.H.S., G.A.W. and E.M. performed immunofluorescence experiments of cultured cells. F.O., A.H.S., K.M.M., S.O.R. and E.M. analysed the immunofluorescence data from experiments with cultured cells. A. Schauss, M. Bergami, E.M. and F.B. designed electron microscopy experiments with cultured cells, brain-engrafted cells and mouse lungs. K.N., A. Chihab, J. Benz, L.P.-R., J.A.S.-C. and F.G. performed electron microscopy experiments of cultured cells, brain-engrafted cells and mouse lungs. V.S., M.T., K.N., A. Chihab and M. Bergami analysed electron microscopy data. H.F., M.A., M. Bergami and F.B. designed electrophysiology experiments. K.N., M. Minère, M. Müller, M.A. and M. Bergami performed electrophysiological measurements. K.N., M.A. and M. Bergami analysed electrophysiology data. H.M.J.-K. and M. Bergami designed rabies tracing experiments. A.A.H. and K.-K.C. generated rabies virus. M.J. generated retroviruses. F.O., K.N., A. Chihab and M. Bergami performed rabies tracing experiments in vitro and in vivo. K.N., F.O. and A. Chihab analysed rabies tracing data. V.S., A. Schmitt, K.N., M.T., M. Boecker, A.H.S., I.P., I.B., M. Bergami and F.B. prepared ex vivo samples. M.T., C.F., T.H., M.L.S., M. Bergami, E.M., H.C.R. and F.B. designed neuronal and nodose ganglion co-culture experiments. V.S., M.T., G.A.W., A.P., M.J., J.I. and E.M. established neuronal and nodose ganglion co-cultures. H.Z. established induced human neuronal cultures. V.S., F.O., M.T., A.H.S., G.A.W., M. Boecker, C.A.F., A.L., M.J., H.Z., J.I. and A.H. prepared co-cultures and co-culture samples. F.O., M.T., R.D.J., S.v.K., A.A., M. Bergami, E.M., H.C.R. and F.B. designed cell viability experiments. V.S., F.O., M.T., M.A.H., P.N., M.K.A.C., E.M. and F.B. analysed data from cell viability experiments. H.W., T.P., H.G., M.L.S., H.C.R. and F.B. designed preclinical experiments. V.S., A. Schmitt, M.T., M. Boecker, M.I.C., O.I., H.L.T., L.W. and F.B. performed preclinical experiments. A. Schmitt, M.T., M. Boecker, M.I.C., O.I., A. Chitsaz, T.P. and F.B. analysed preclinical and MRI data. A. Schmitt, F.O., K.N., I.P., J. Brägelmann, A.H.S., S.O.R., G.R., G.G.H., I.B., M.A., M. Bergami, E.M. and F.B. generated the figures. M. Bergami, E.M., H.C.R. and F.B. wrote the manuscript. All authors revised the manuscript.

**Funding** Open access funding provided by Universität zu Köln.

**Competing interests** H.C.R. received consulting and lecture fees from Abbvie, AstraZeneca, Vertex and Merck. H.C.R. received research funding from AstraZeneca and Gilead Pharmaceuticals. H.C.R. is a co-founder of CDL Therapeutics. R.K.T. is a founder of PearlRiver Bio (now part of Centessa), a shareholder of Centessa, a founder and shareholder of Epiphanes and a consultant to PearlRiver Bio and Epiphanes. R.K.T. has received research support from Roche. R.K.T. and J.S. are co-founders and shareholders of DISCO Pharmaceuticals. M.L.S. is a co-founder and was an advisor of PearlRiver Bio (now part of Centessa). M.L.S. received research funding from PearlRiver Bio (now part of Centessa). S.O.R. is a shareholder of NanoTag Biotechnologies. H.W. was a co-founder of Apogenix. J. George is a consultant to DISCO Pharmaceuticals and received honoraria from MSD and Boehringer Ingelheim. J. Brägelmann has received research funding from Bayer and travel grants from Merck and Bicycle Therapeutics. The other authors declare no competing interests.

**Additional information**
**Correspondence and requests for materials** should be addressed to Max Anstötz, Silvio O. Rizzoli, Matteo Bergami, Elisa Motori, Hans Christian Reinhardt or Filippo Beleggia.

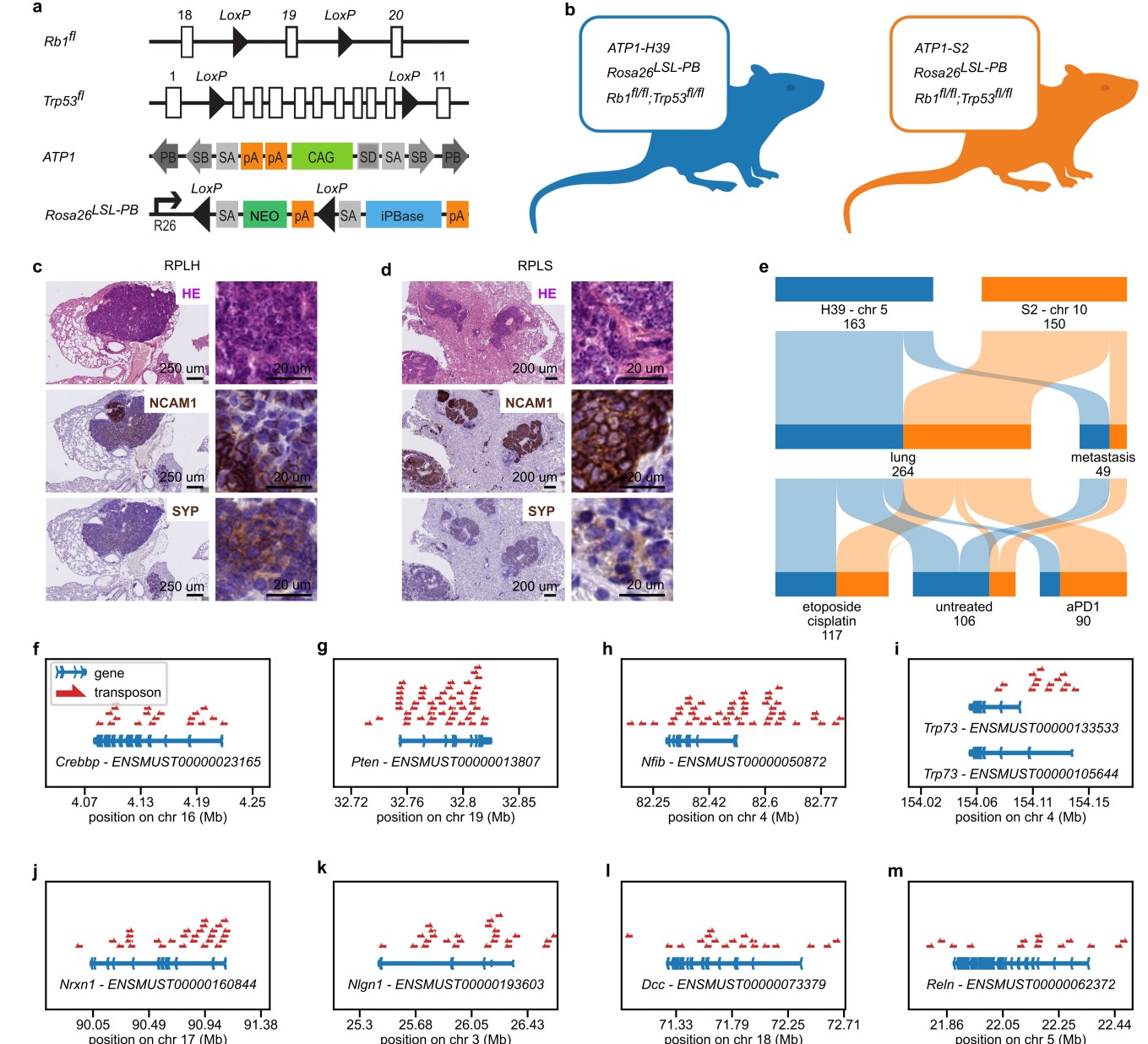

**Extended Data Fig. 1 | *piggyBac* insertional mutagenesis screen. a**) Alleles included in the mouse model **b**) Mouse lines included in the screen carry the *Rb1^{fl/fl}* and *Tp53^{fl/fl}* alleles with the addition of the conditional allele to express the *piggyBac* transposase (*Rosa26^{LSL-PB}*). The RPLH line (blue) additionally carries the donor allele *ATP1-H39*, with 80 copies of the *ATP1* transposon on chromosome 5. The RPLS line (orange) additionally carries the donor allele *ATP1-S2*, with 20 copies of the *ATP1* transposon on chromosome 10. **c, d**) Tumors derived from RPLH and RPLS mice display typical SCLC morphology, including scant cytoplasm, salt and pepper chromatin and positivity for NCAM1 and SYP. **e**) Tumors harvested from 31 RPLH (blue) and 27 RPLS (orange) mice include lung and metastatic samples and derive from untreated mice, from mice treated with etoposide and cisplatin and from mice treated with anti-PD1 antibody RMP1-14. PB, *piggyBac* inverted terminal repeat (ITR); SB, *Sleeping Beauty* ITR; SA, splicing acceptor; pA, polyadenylation signal; CAG, CMV enhancer and chicken beta-actin promoter; SD, splicing donor; NEO, neomycin resistance; iPBase, *piggyBac* transposase. **f-m**) Transposon insertions (red arrows) identified in selected genes (horizontal blue lines). The orientation of the exons (vertical obtuse blue angles) point to the direction of transcription. **f**) Insertions in *Crebbp* **g**) Insertions in *Pten*. **h**) Insertions in *Nfib*. **i**) Insertions in *Trp73*. **j**) Insertions in *Nrxn1*. **k**) Insertions in *Nlgn1*. **l**) Insertions in *Dcc*. **m**) Insertions in *Reln*.

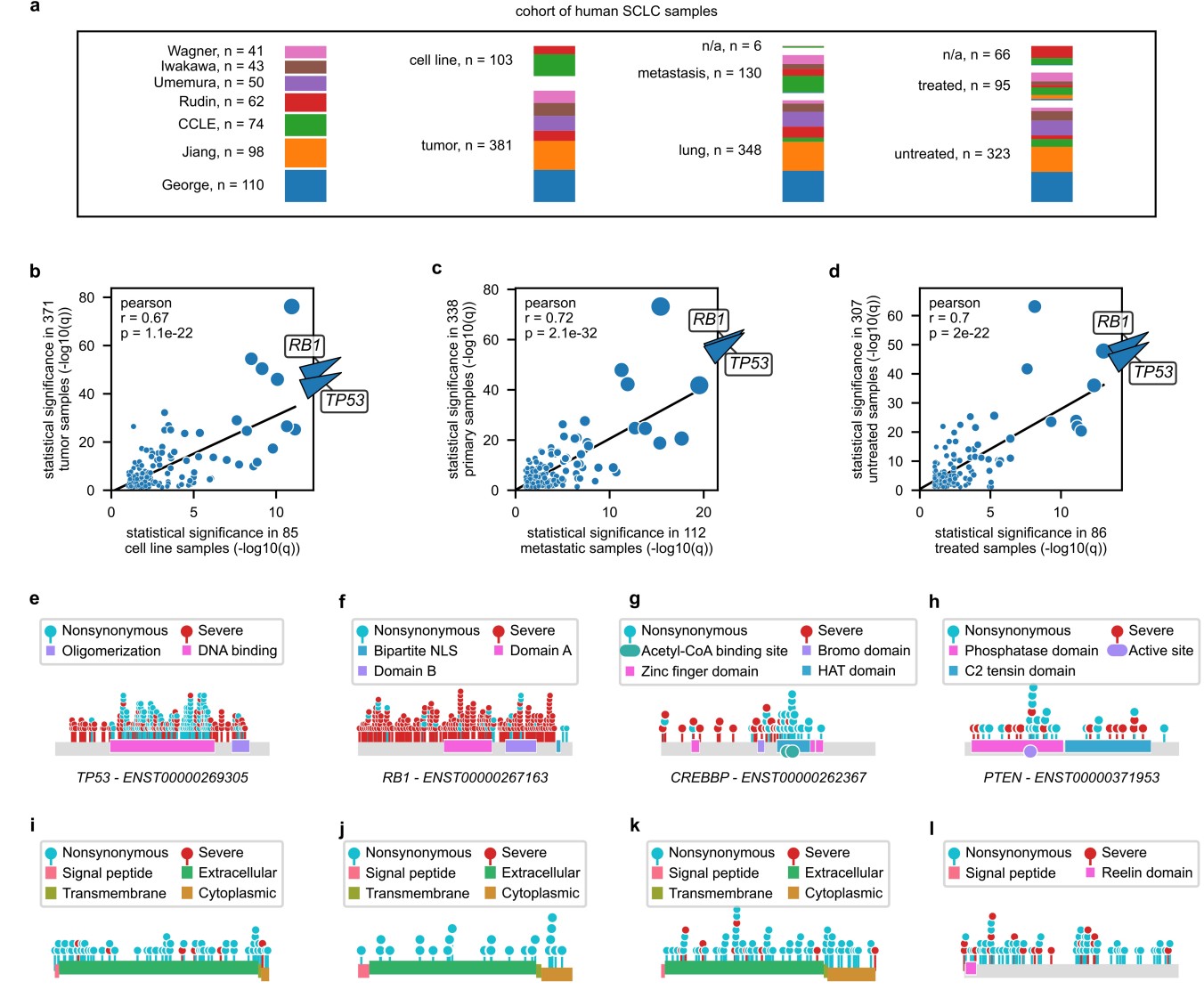

**Extended Data Fig. 2 | Overview of genetic data from human SCLC patients.**
**a**) Origin and characteristics of human samples from different studies.
**b**) Similar genes are identified in tumor and cell line samples. **c**) Similar genes are identified in primary and metastatic samples. **d**) Similar genes are identified in treated and untreated samples. **e-l**) Selected genes are shown with the corresponding proteins annotated with UniProt Knowledgebase annotations. Mutations identified in SCLC samples are shown as a lollipop chart above the protein. Severe mutations (stop, frameshift, start-loss, and canonical splice-site

mutations) are shown in red. Nonsynonymous mutations (amino-acid substitutions, non-frameshift indels) are shown in light blue. **e**) Mutations in *TP53* are either severe or clustered in the DNA-binding domain. **f**) Mutations in *RB1* are almost exclusively severe. **g**) Mutations in *CREBBP* are severe or clustered in the HAT domain. **h**) Mutations in *PTEN* are severe or clustered on the active site. **i**) Mutations in *NRXN1* are mainly nonsynonymous. **j**) Mutations in *NLGN1* are exclusively nonsynonymous. **k**) Mutations in *DCC* are mainly nonsynonymous. **l**) Mutations in *RELN* are nonsynonymous or severe.

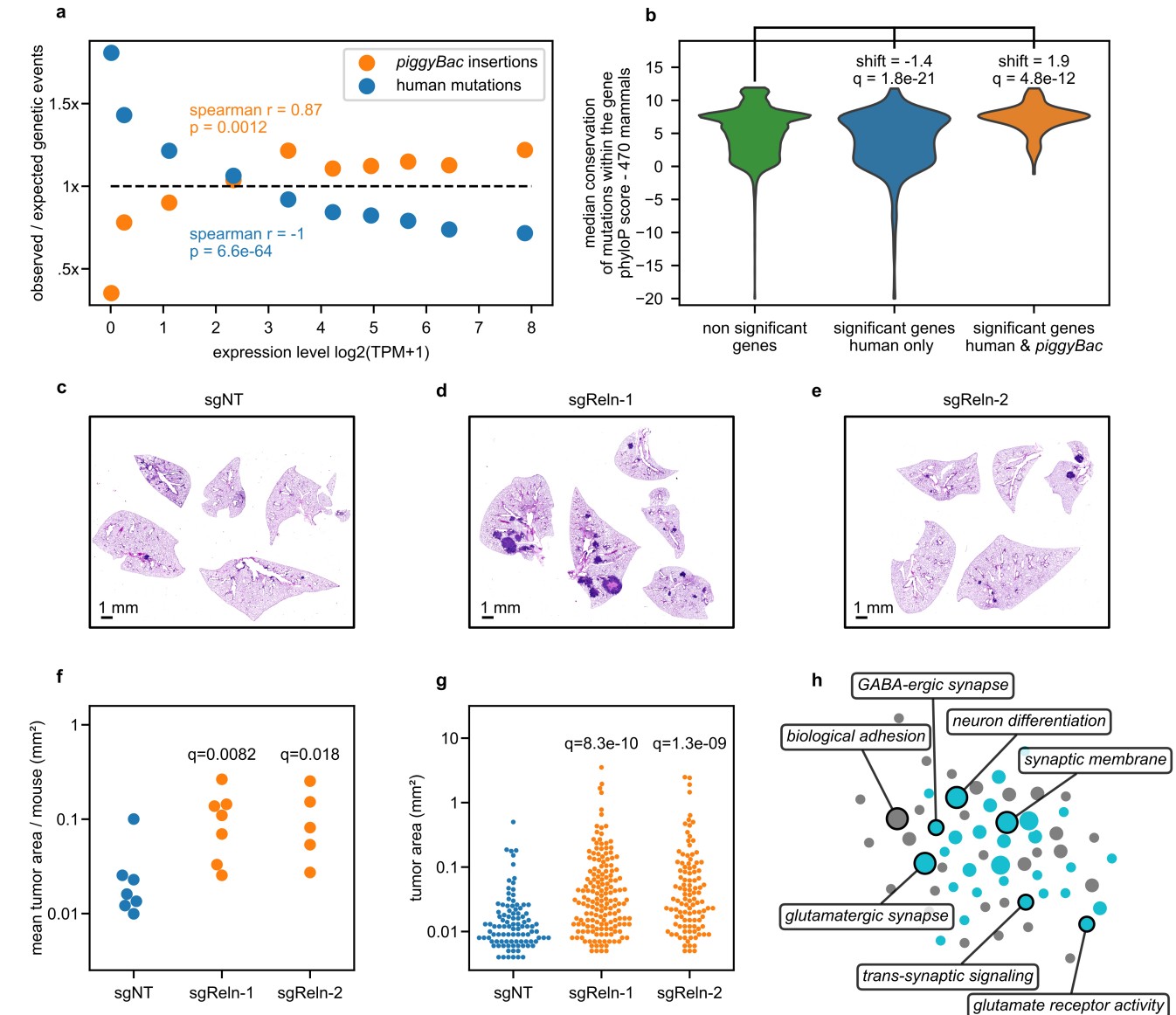

**Extended Data Fig. 3 | Cross-validation of genetic datasets. a**) The background rate of genetic events in the *piggyBac* and human mutation datasets have opposite correlations to expression levels. Genes are binned into equal-sized bins based on their expression level. On the x axis, the bins are plotted on the mean expression of their genes. On the y axis, the bins are plotted on the ratio of total observed/total expected events for the bin. p-value: two sided Spearman correlation test. **b**) Mean conservation of mutated nucleotides for genes not identified in the human mutation datasets, for genes identified only in the human mutations dataset and for genes identified in both the human mutations and *piggyBac* datasets. q-values: two-sided Mann-Whitney test with FDR correction, both compared to non-significant genes **c-e**) Hematoxylin-eosin stains of

individual RPR2TC mice induced with lentiviral vectors carrying a non-targeting sgRNA or sgRNAs targeting *Reln*. Representative of 7, 5 and 5 mice, respectively. **f**) The mean area of tumors identified in mice induced with sgRNAs targeting *Reln* is significantly larger than the area of tumors induced with the non-targeting sgRNA. N = 7 mice for non-targeting sgRNA and sgReln-1, n = 5 mice for sgReln-2. q-values: Mann-Whitney test with FDR correction, both compared to sgNT controls. **g**) The size of individual tumors from the mice in **f** is significantly greater in mice induced with sgRNAs targeting *Reln*. q-values as in **f. h**) Force-directed graph of gene ontology analysis, showing gene sets enriched in both the *piggyBac* dataset and the analysis of human genetic data. Most gene sets are related to synaptic and neuronal functions (light blue).

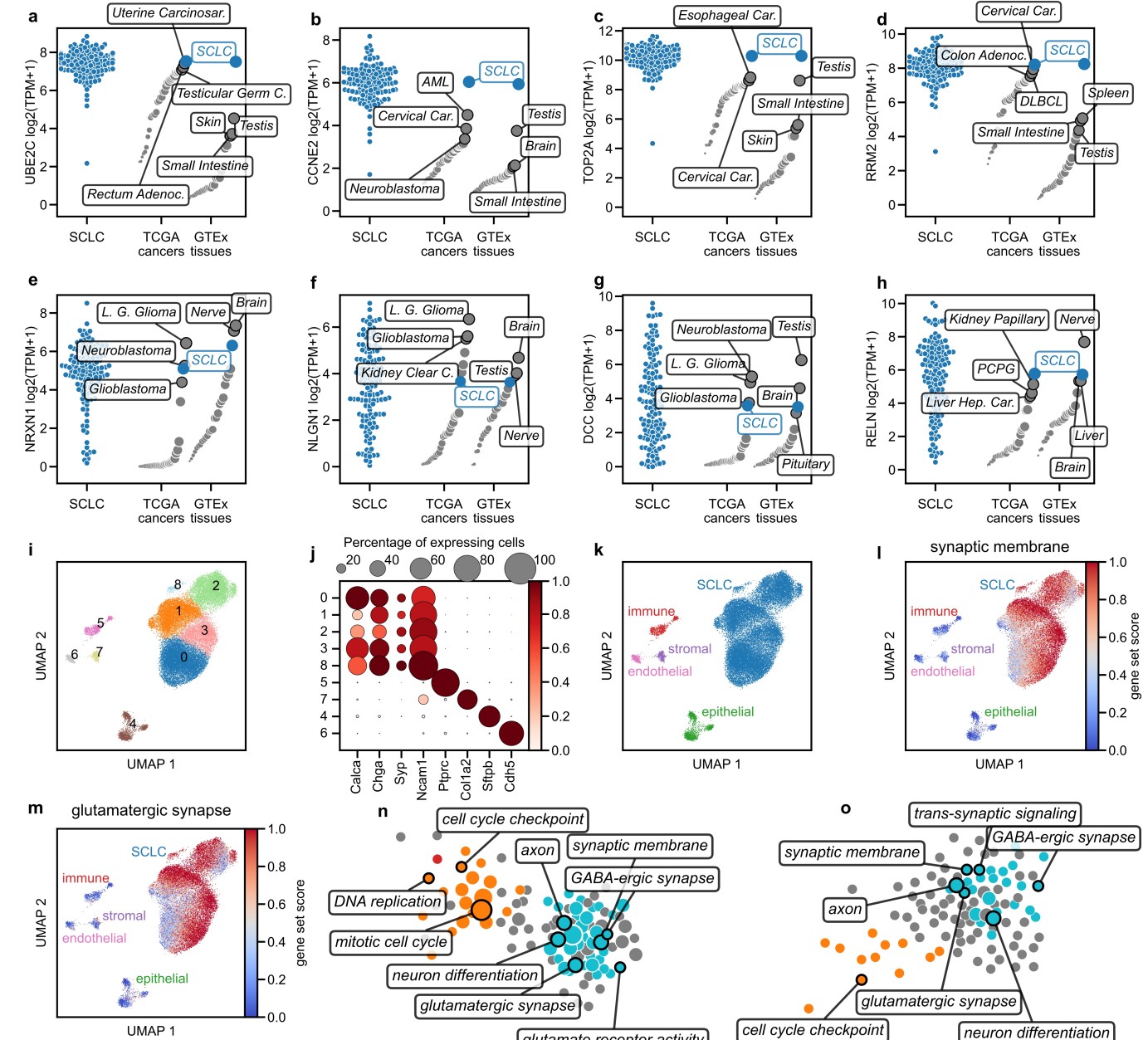

**Extended Data Fig. 4 | Expression of synaptic gene sets in SCLC. a-h)** Selected genes highly expressed in SCLC. The expression levels of individual SCLC samples are shown on the left of each panel. The median expression levels in cancer types included in TCGA and Neuroblastoma as positive control are depicted in the middle. The median expression levels of healthy tissues are on the right. **a, b, c, d)** The expression levels of *TOP2A*, *CCNE2*, *RRM2* and *UBE2C*, representative of genes involved in cell-proliferation, are higher in SCLC than in any other cancer or healthy tissue. **e, f, g, h)** The expression levels of *NRXN1, NLGN1, DCC and RELN*, representative of synaptic and neuronal genes, are higher in SCLC than in most other cancers and tissues. **i)** Leiden clustering of snRNA-seq data from six murine tumors derived from *Rb1^fl/fl*;*Trp53^fl/fl* mice. **j)** The leiden clusters from panel **i** show markers of SCLC cells (*Calca, Chga, Syp,*

*Ncam1*) or of one of four broad cell types expected in the lung (*Ptprc* for immune cells, *Col1a2* for stromal cells, *Sftpb* for epithelial cells and *Cdh5* for endothelial cells). **k)** Visualization of cell types based on markers identified in panel **j. l)** Genes within the *Synaptic Membrane* GO term are enriched in the cancer cells. **m)** Genes within the *Glutamatergic Synapse* GO term are enriched in the cancer cells. **n)** Comparison of murine SCLC cells to other lung cell types revealed an enrichment in neuronal and cell proliferation GO terms, with striking resemblance to the analysis of bulk human RNA-seq data (Fig. 1c). **o)** Comparison of SCLC cells to other cell types in published human lung scRNA-seq data revealed an enrichment in neuronal and cell proliferation GO terms, with striking resemblance to the analysis of bulk human RNA-seq data (Fig. 1c).

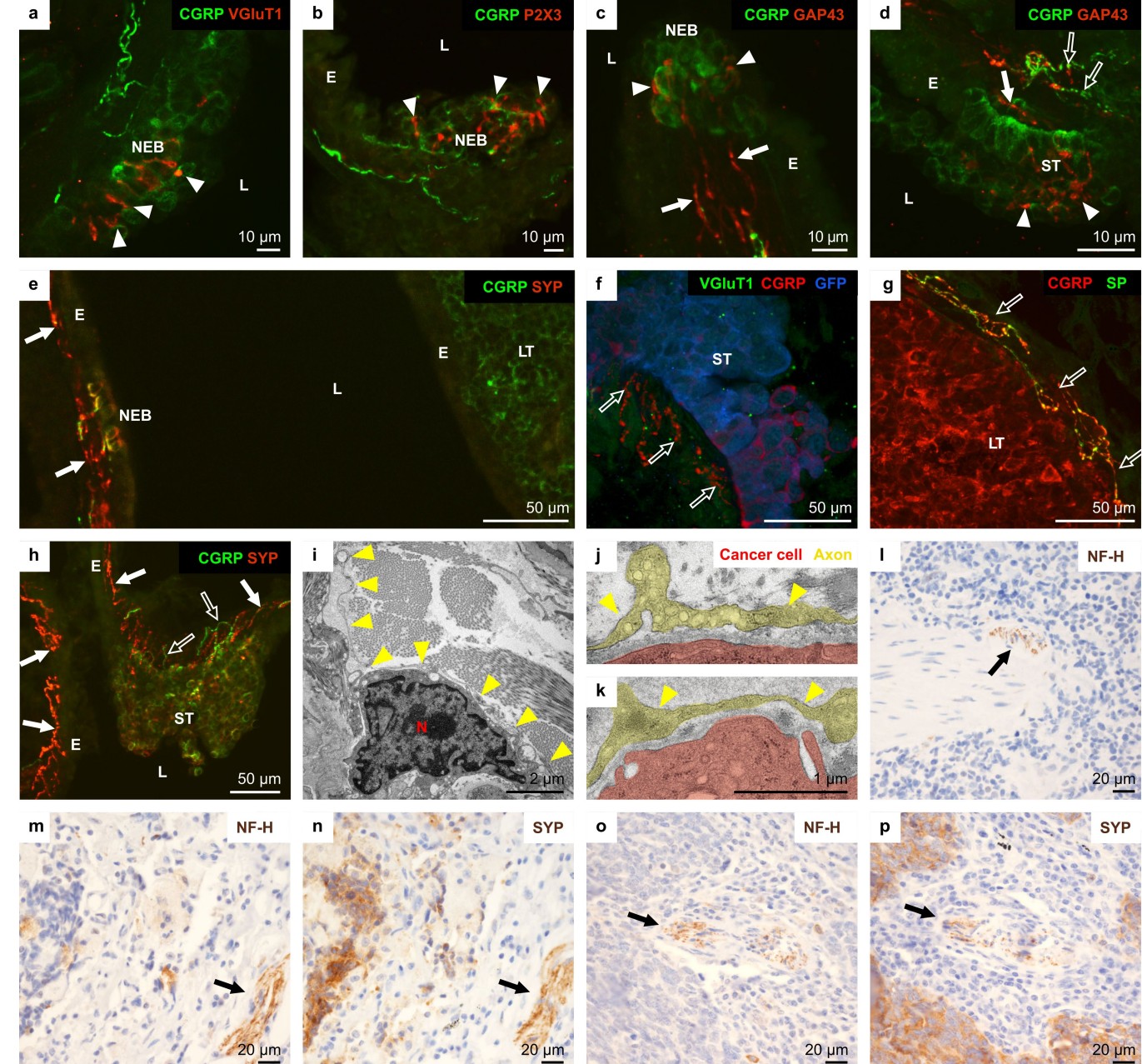

**Extended Data Fig. 5 | Nerve fibers in the SCLC microenvironment.**
**a-h)** Confocal images of lung cryostat sections of RP mice. L: lumen of the
airways. E: airway epithelium. **a)** Intraepithelial VGluT1+ nerve terminals
(arrowheads) branch between the CGRP+ PNECs. **b)** Intraepithelial P2X3+ nerve
terminals (arrowheads) protruding between the CGRP+ (green) neuroendocrine
cells of a NEB. **c)** GAP43+ nerve fibers (arrows) branch and protrude (arrowheads)
between the CGRP+ PNECs. **d)** A GAP43+ nerve fiber (arrow) branches
(arrowheads) between the CGRP + SCLC cells (green). CGRP+ nerve fibers (open
arrows) are seen close to the base of the tumor. **e)** Subepithelial SYP+ and CGRP+
nerve terminals (arrows) innervate a NEB. Remarkable is that the subepithelial
area adjacent to a large tumor appears devoid of nerve fibers. **f)** Small tumor
(ST) from an RPC mouse, with no visible innervation from VGluT1+ fibers.
Varicose CGRP+ fibers are visible below the tumor (open arrowheads). **g)** Large

tumor (LT) surrounded by varicose CGRP+ and substance P+ (SP) nerve fibers.
**h)** SYP+ (arrows) and CGRP+ (open arrows) nerve fibers can be seen in the
epithelium at the base of a small tumor (ST). **i)** Electron micrographs showing a
cancer cell surrounded by long axon-like fibers near the periphery of a tumor in
the lung of an RP mouse. **j, k)** Magnifications showing the presence of enlarged
structures along identified fibers (yellow pseudocolor) containing multiple
vesicles and mitochondria (M) near the cancer cell (red pseudocolor). **l-p)** DAB
staining of biopsies from three SCLC patients. All sections are counterstained
with hemalum. **l)** NF-H-positive nerve fibers near an intratumoral vessel in the
biopsy from the first patient. **m, n)** NF-H-positive nerve fibers at the borders of
a SYP-positive tumor in a biopsy from the second patient. **o, p)** NF-H-positive
nerve fibers at the borders of a SYP-positive tumor in a biopsy from a third
patient.

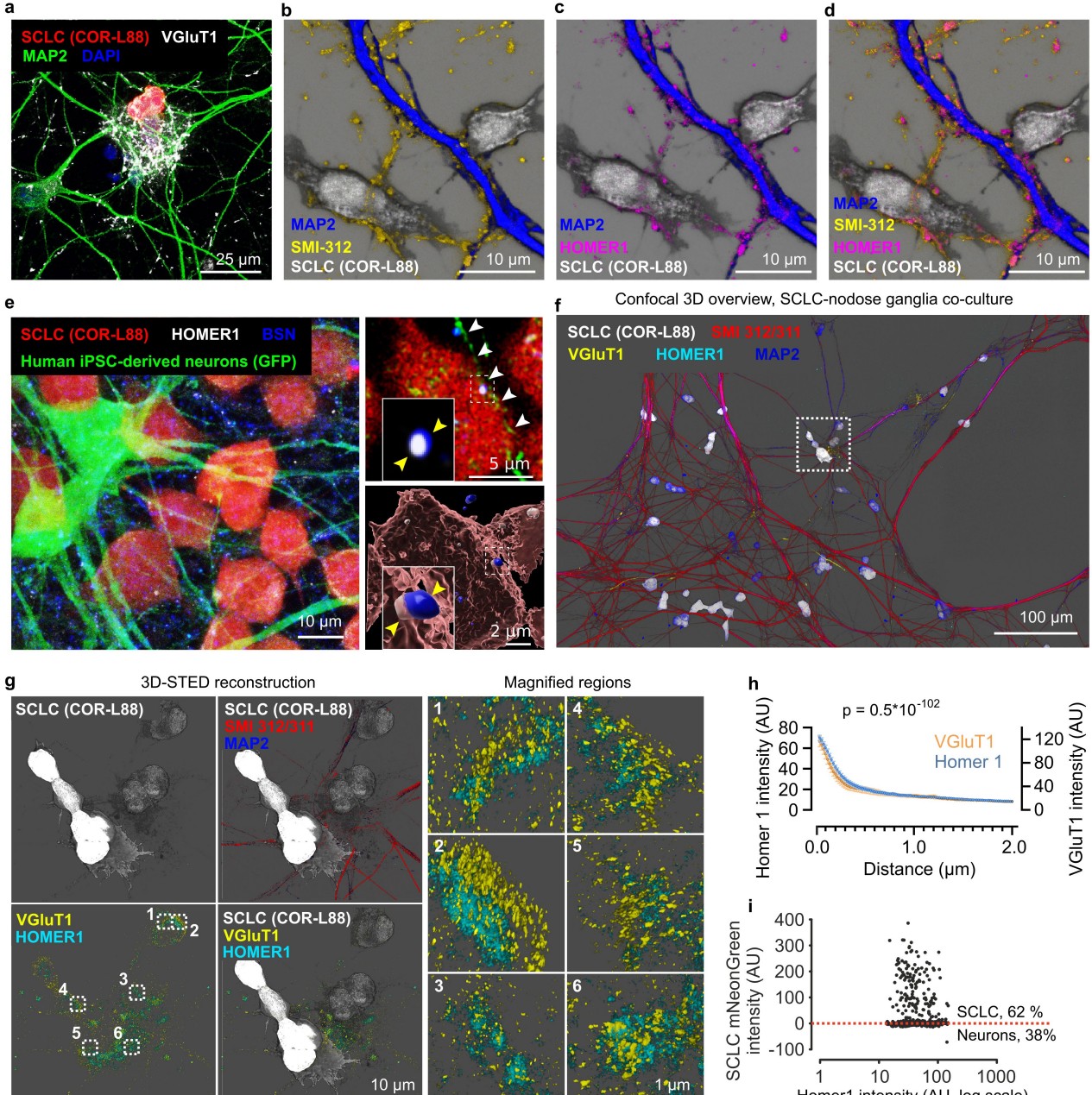

**Extended Data Fig. 6 | Cancer-to-neuron contacts in vitro. a)** Co-culture of murine cortical neurons (immunolabeled against MAP2) and SCLC cells (COR-L88, expressing DsRed) showing the appearance of dense VGLUT1-positive puncta onto SCLC cells contacted by neuronal terminals **b-d)** Different views of a 3D-reconstruction of 3D-STED for co-cultures immunolabeled against axonal marker SMI-312, mNeonGreen to mark SCLC cells, dendritic marker MAP2, and postsynaptic marker HOMER1, showing that that the contacts on cancer cells are predominantly axonal. Representative of 3 experiments. **e)** Co-culture of human iPSC-derived cortical neurons and SCLC cells (COR-L88, expressing tdTomato), immunostained for the pre- and post-synaptic markers BSN and HOMER1. Right panels show a single confocal stack (top) and 3D reconstruction (bottom) of an SCLC cell contacted by a GFP-positive axonal fiber (white arrowheads) exhibiting BSN and HOMER1 co-localizing puncta (yellow arrowheads) located outside and inside the SCLC cell surface. **f)** Confocal overview of SCLC cells

(mNeonGreen + , shown in white) co-cultured with murine nodose ganglia. **g)** A detailed view of the boxed region from panel **f**, followed by individual magnified regions, which indicate the arrangements of VGluT1 (presynaptic, neuronal) and HOMER1 (postsynaptic, within SCLC cell) molecules. **h)** Line scans were drawn automatically across HOMER1 spots, starting in their intensity maxima, and moving towards the periphery. The signal drops, as expected; a similar drop is seen in the VGluT1 signal, confirming their close apposition (N = 4 independent experiments, n = 782 line scans. Colocalization tested using a two-sided Pearson correlation test. Error bars: standard error of the mean). **i)** Correlative intensity scatter plot of SCLC mNeonGreen signal vs HOMER1 signal (N = 4 independent experiments, N = 782) indicates that a substantial proportion of the HOMER1-marked spots are formed on SCLC cells, and therefore show a measurable mNeonGreen signal (62.02%).

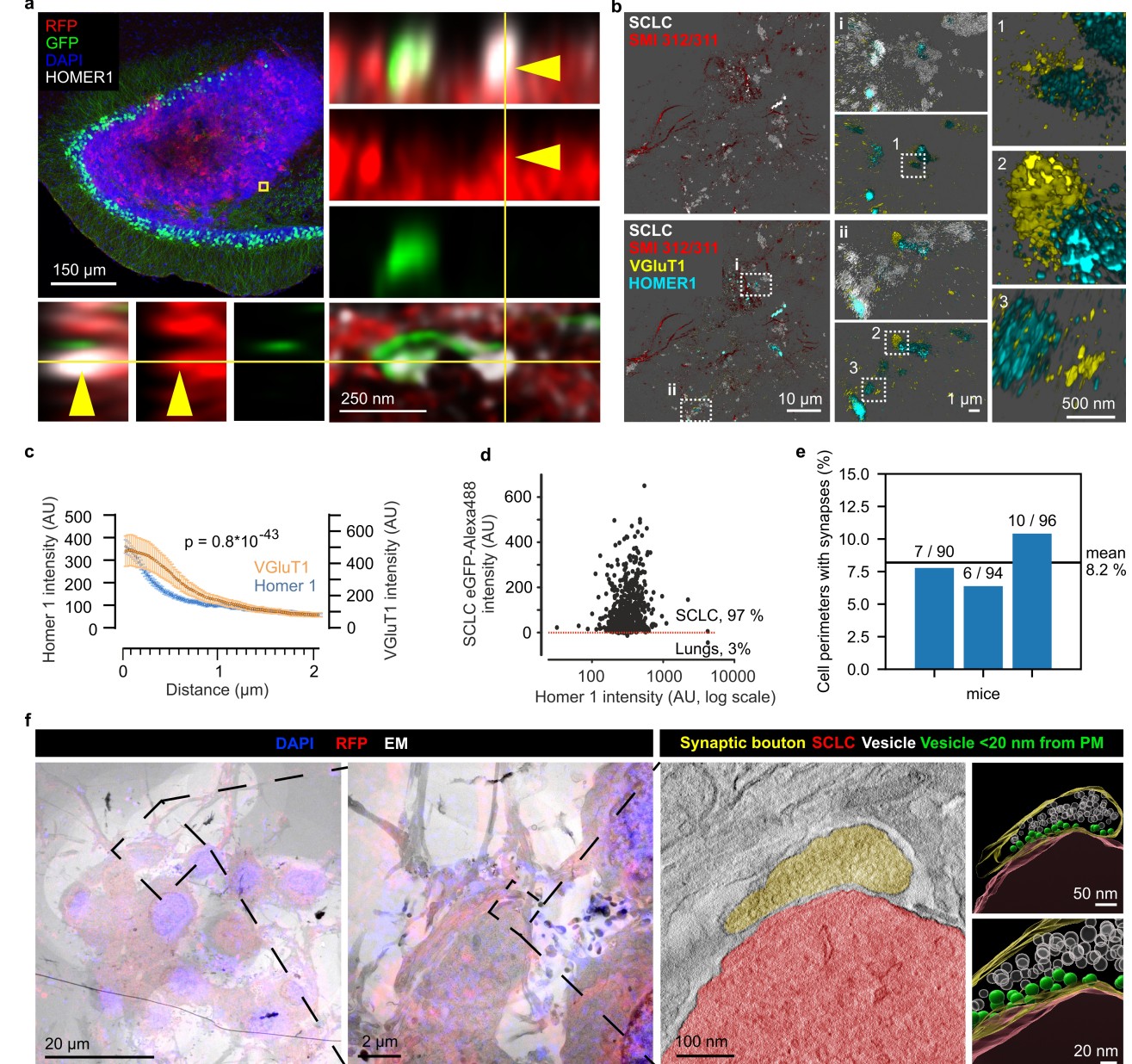

**Extended Data Fig. 7 | Cancer-to-neuron contacts in vivo. a**) Confocal imaging of grafted DsRed-expressing SCLC cells in the hippocampus of a *Thy1*-GFP mouse (top-left), depicting GFP-positive fibers contacting SCLC cells in the tumor periphery (lower-right). On the right and below are orthogonal views on a point of contact between a putative axonal bouton and a DsRed/HOMER1 double-positive punctum in the SCLC cell (arrowheads). **b**) 3D-STED image of a lung section immunolabeled against the presynaptic marker VGluT1, the postsynaptic marker HOMER1, and an axonal marker (SMI312/SMI311 epitopes). The right panels show magnifications of putative synapses on cancer cells in the marked regions. **c**) Automatic line scans from the intensity maxima of HOMER1 spots towards the periphery. The signal drops and a similar drop is seen in the VGluT1 signal (N = 3 independent experiments, n = 609 line scans, of which 213 represented putative synapses. Two-sided Pearson correlation test. Error bars: standard error of the mean). **d**) Correlative intensity scatter plot of SCLC mNeonGreen signal vs HOMER1 signal (N = 3 independent experiments, n = 609 measurements) indicates that most HOMER1 spots in these regions are within SCLC cells. **e**) Quantification of synapses contacting tdTomato-positive cancer cells in brain allografts. For each mouse (n = 3), 90-96 perimeters in 12-14 consecutive ultrathin sections were examined, for a total of 280 cell perimeters. **f**) CLEM of COR-L88 SCLC cells (expressing DsRed) co-cultured with cortical neurons. The left panels depict the registered overlay between fluorescent and EM images. The third panel shows the electron tomogram of a synapse, with a presynaptic bouton (yellow pseudocolor) filled with vesicles, contacting the cancer cell (red pseudocolor). The right panels show a 3D reconstruction of the tomogram (250 nm thick), depicting cancer cell (red), axonal bouton (yellow) with vesicles (white), and vesicles located within 20 nm from the plasma membrane (PM) (green).

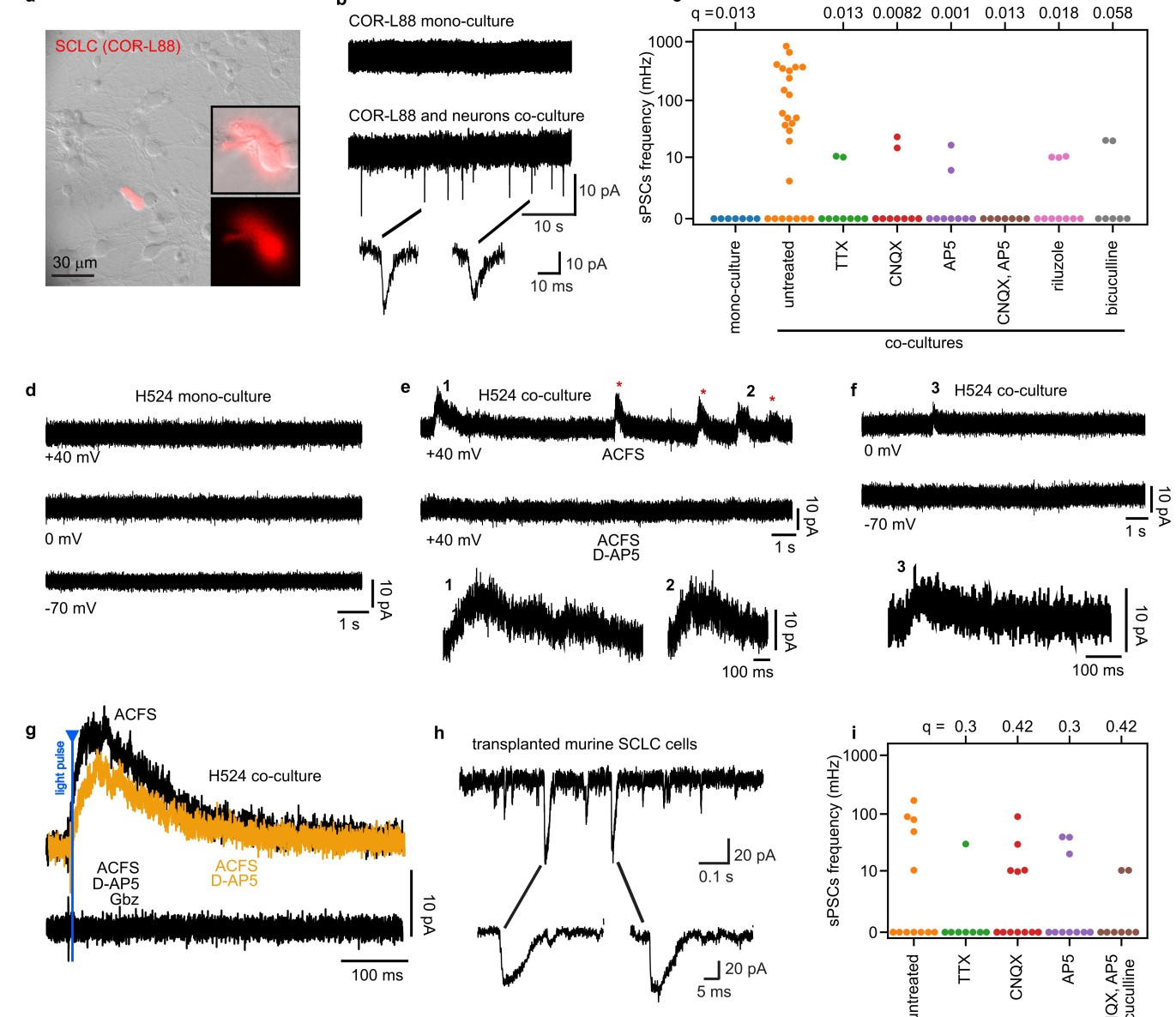

**Extended Data Fig. 8 | Electrophysiology of SCLC cells. a**) Example of a patched DsRed-expressing SCLC cell (COR-L88) under whole-cell configuration in cortical neuron-SCLC co-cultures. **b**) Whole-cell, voltage-clamp traces of sPSCs in SCLC cells (COR-L88) in the presence or absence of neurons. **c**) Quantification of sPSC frequency in co-culture in the presence or absence of the indicated blockers (TTX, CNQX, D-AP5, Riluzole and Bicuculline) (n = 7-30 cells per condition). All conditions are compared to untreated co-cultures. q values: two-sided Mann-Whitney with FDR correction **d-g**) Whole-cell voltage-clamp traces of H524 cells **d**) Traces recorded at three different voltages (−70 mV, 0 mV, and +40 mV) in mono-culture. **e**) Traces recorded at +40 mV in co-culture with cortical neurons. The synaptic events (red stars and numbers) can be completely abolished by the application of the NMDA receptor blocker D-AP5 and display a long decay time lasting several hundred milliseconds.

**f**) Traces recorded at −70 mV and 0 mV in co-culture with cortical neurons. Note the occurrence of synaptic events at 0 mV, indicating a GABA-A-mediated chloride inward current. **g**) Traces recorded at +40 mV in co-culture with Channelrhodopsin 2-eYFP expressing (ChR2-eYFP, green) cortical neurons after a short blue light pulse (5 ms). Note the partial decrease in event amplitude during NMDA receptor blockade with D-AP5 (orange), followed by complete abolishment after additional GABA-A receptor blockade with Gbz (lower trace). **h**) Whole-cell, voltage-clamp recording in an acute hippocampal slice of grafted DsRed-expressing murine SCLC cells. **i**) Quantification of sPSCs in grafted cancer cells in acute slices in the absence or presence of the indicated blockers (TTX, CNQX, D-AP5 and Bicuculline). All conditions are compared to untreated slices. q values: two-sided Mann-Whitney with FDR correction, n = 8-17 cells per condition.

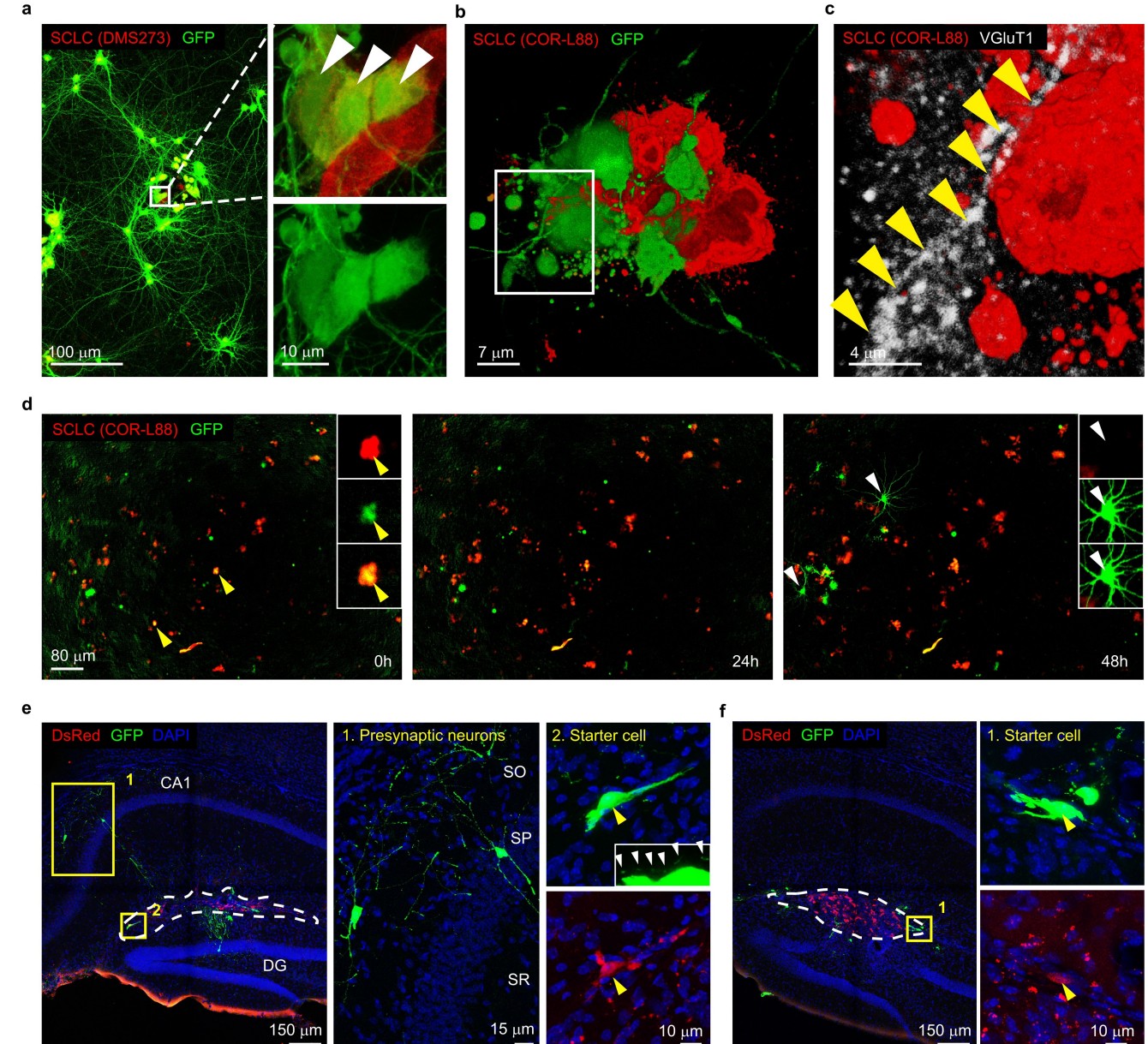

**Extended Data Fig. 9 | RABV-tracing of SCLC cells to presynaptic neurons.**
**a**) RABV-GFP-based tracing of neurons monosynaptically connected to
DMS273 SCLC cells expressing DsRed. Right panels show enlarged views of the
boxed area containing double-positive starter SCLC cells (arrowheads). **b**) 3D
reconstruction of double-positive starter cells in a cluster of DsRed-expressing
SCLC cells (COR-L88) following RABV-GFP-based tracing. **c**) Magnification of
the panel boxed in **b**, showing the profuse expression of VGluT1-positive puncta
in GFP-positive neuronal fibers (yellow arrowheads) contacting starter SCLC
cells. **d**) Time-lapse of RABV-traced neurons in neuron-SCLC co-cultures over
48 h. Selected frames at the indicated time points show the initial presence of
starter cancer cells (double-positive for the retrovirally-encoded DsRed and
the RABV-encoded GFP, yellow arrowheads), which proliferate over time, and

the emergence of GFP+ neurons at 48 h (white arrowheads). **e**) Example of
RABV-GFP-based tracing of morphologically identified inhibitory GABAergic
neurons located in the stratum oriens (SO) and pyramidale (SP) of CA1,
following transplantation of G-TVA-expressing murine SCLC cells (dashed
area). Right panels show zooms of the boxed areas depicting identified GFP+
neurons (1) and starter SCLC cell (2), contacted by varicosities of a passing axon
(arrowheads). SR, stratum radiatum. **f**) Example of RABV-GFP-based tracing
following transplantation of TVA-only-expressing murine SCLC cells (dashed
area), showing the virtual absence of GFP-positive presynaptic neurons. Right
panels show zooms of the boxed area depicting an identified DsRed/GFP
double-positive SCLC cell (arrowhead).

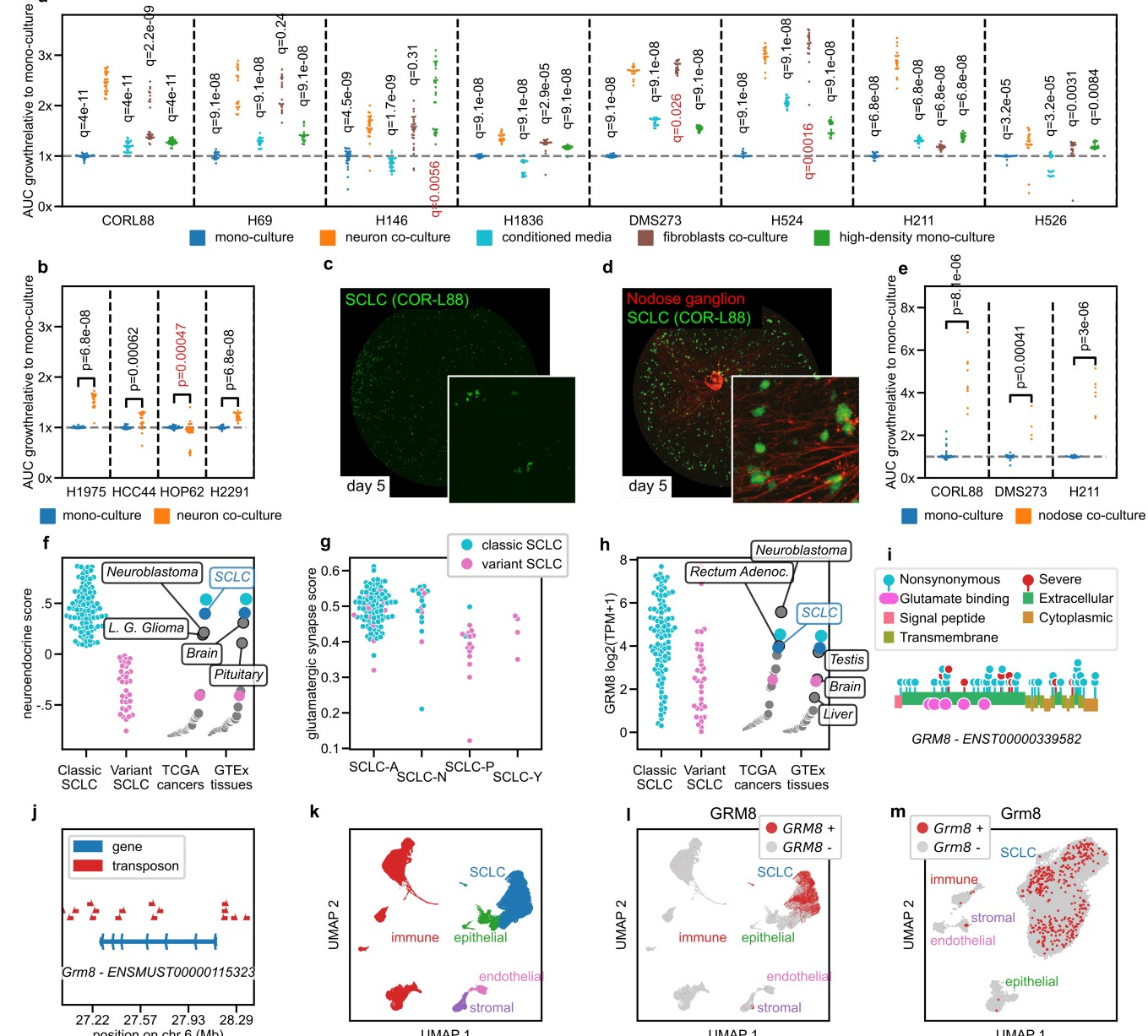

**Extended Data Fig. 10 | neuron-promoted SCLC proliferation and Grm8.**
**a**) Growth of SCLC cell lines monitored via live cell imaging under different conditions. Each dot represents an individual well. All conditions are compared to the growth in co-culture with cortical neurons. q-value: two-sided Mann-Whitney test with FDR correction. n ≥ 20 wells / condition, n ≥ 4 neuron batches. Red q-values indicate faster growth than neuronal co-cultures.
**b**) Growth of NSCLC cell lines monitored via live cell imaging in mono-culture or co-culture. Each dot represents an individual well. p-values: two-sided Mann-Whitney test. n ≥ 20 wells / condition, n ≥ 4 neuron batches. **c, d**) Individual wells containing COR-L88 SCLC cells in mono-culture (**c**) or in co-culture with nodose ganglia (**d**). **e**) Quantification of the growth of SCLC cell lines via live cell imaging with and without nodose ganglia. Each dot represents an individual well. p-value: two-sided Mann-Whitney test as in **b**. n = 4-29 wells/condition,

n ≥ 4 individual ganglia. **f**) SCLC samples are separated into classic and variant subtypes based on the expression of neuroendocrine features. **g**) SCLC samples of the SCLC-A and SCLC-N subtypes express higher level of genes included in the GO term *Glutamatergic Synapse*. **h**) The expression of *GRM8* is higher in SCLC than in most other cancers and tissues and is especially high in classic SCLC with strong neuroendocrine features. **i**) GRM8 protein with annotations from the UniProt Knowledgebase. Mutations in SCLC samples are shown as a lollipop chart. **j**) Transposon insertions identified in *Grm8*. **k**) UMAP plot of published human SCLC and normal lung scRNA-seq. The cells are grouped into differentiation groups. **l**) *GRM8* is specifically expressed in SCLC cells from panel **k**. **m**) UMAP plot of snRNA-seq samples from murine RP tumors, characterized in Extended Data Fig. 4. *Grm8* is specifically expressed in SCLC cells.

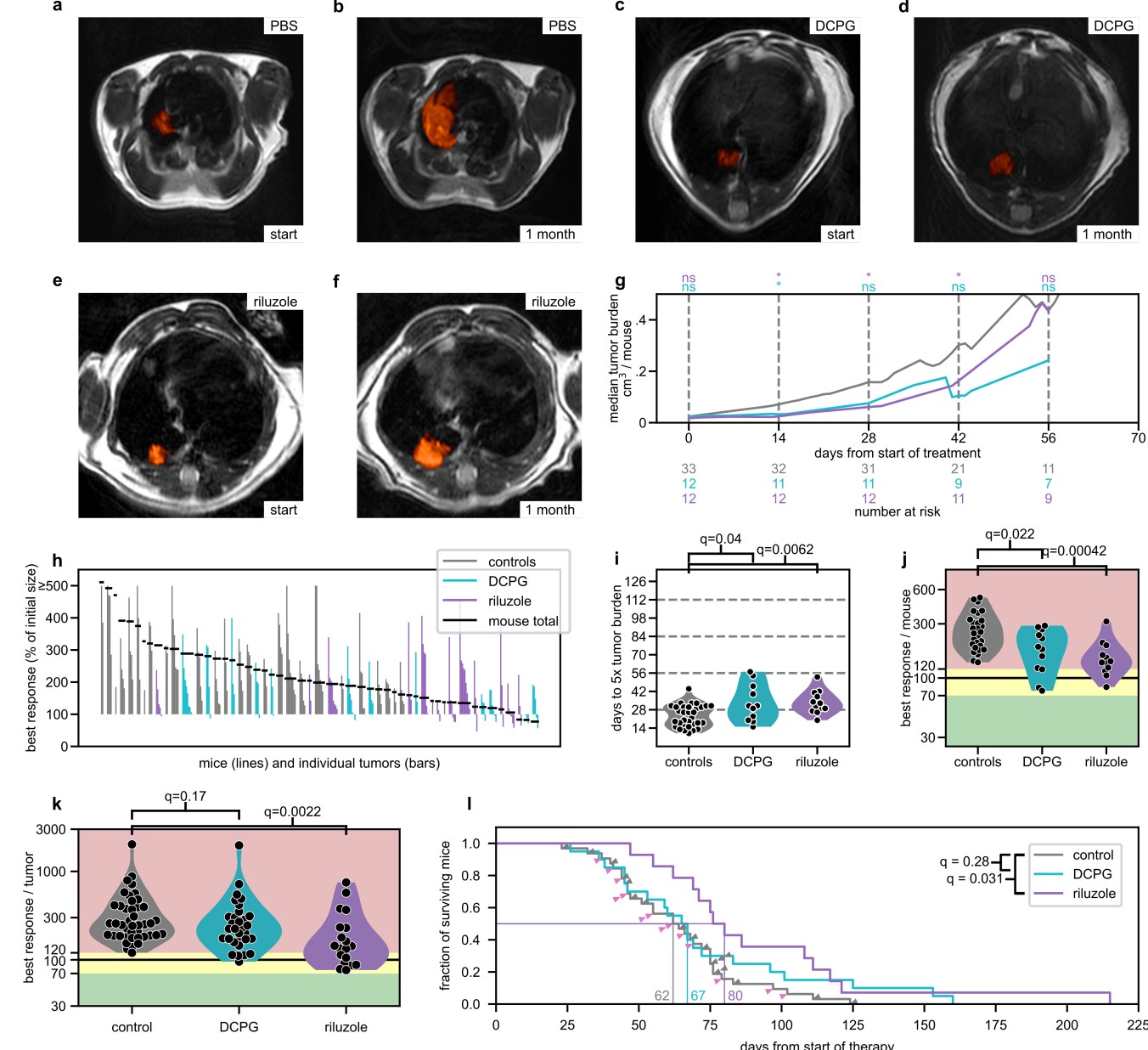

**Extended Data Fig. 11 | Response of SCLC tumor-bearing mice under anti-glutamatergic treatment. a-f**) Representative MRI scans of tumor-bearing *Rb1fl/fl;Trp53fl/fl* mice under different treatments. Tumors are pseudocolored in red. **a, b**) Mouse treated with PBS, showing a large increase in tumor size after one month. **c, d**) Mouse treated with DCPG, showing a minor increase in tumor size after one month. **e, f**) Mouse treated with riluzole, showing a minor increase in tumor size after one month. **g**) Median tumor burden for RP mice treated with riluzole (n = 12), DCPG (n = 12) or vehicle controls (n = 33). q-values: two-sided Mann-Whitney test with FDR correction. **h**) Waterfall chart, showing the best response of individual tumors, grouped by mouse. The mice are sorted based on the total best response. **i**) Time required for tumors to reach a size five fold greater than the size at inclusion for mice treated with riluzole (n = 12), DCPG (n = 11) or the relative controls (n = 32). q-values: two-sided Mann-Whitney test with FDR correction. **j**) Best response achieved throughout treatment, calculated based on the total tumor burden for each mouse for mice treated with DCPG (n = 12), riluzole (n = 12) or the relative controls (n = 28). q-values: two-sided Mann-Whitney test with FDR correction. **k**) Best response of individual tumors from RP mice induced with CGRP-Cre. The mice were treated with DCPG (31 tumors from 18 mice), riluzole (19 tumors from 13 mice), or the relative controls (23 tumors from 17 mice for PBS plus 20 tumors from 12 mice for riluzole vehicle). **l**) Survival of RP mice induced with CGRP-Cre. Riluzole treatment (n = 14) results in significantly longer survival compared to control mice (n = 18 for PBS plus n = 14 for riluzole vehicle). The benefit provided by DCPG (n = 20) is not statistically significant. q-values: two-sided Mann-Whitney test with FDR correction.

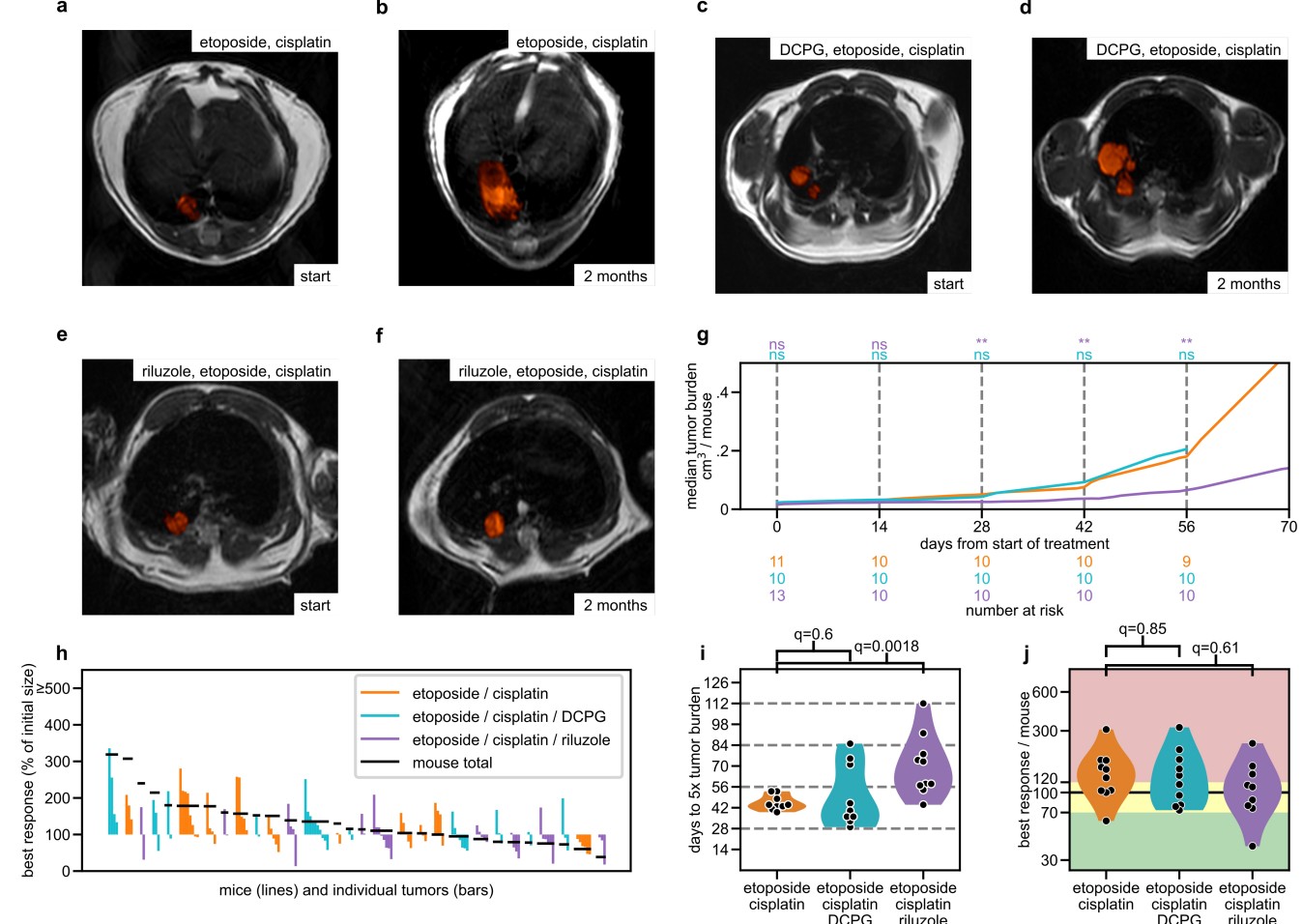

**Extended Data Fig. 12 | Combination treatment with chemotherapy and anti-glutamatergc drugs. a-f**) Representative MRI scans of tumor-bearing *Rb1fl/fl;TrpS3fl/fl* mice under different treatments. Tumors are pseudocolored in red. **a, b**) Mouse treated with etoposide and cisplatin (EC), showing the increase in tumor size after two months. **c, d**) Mouse treated with EC and DCPG (ECD), showing the increase in tumor size after two months. **e, f**) Mouse treated with EC and riluzole (ECR), showing a stable disease after two months. **g**) Median tumor burden for RP mice treated with EC (n = 11), ECR (n = 13) or ECD (n = 10). q-values: two-sided Mann-Whitney test with FDR correction.

**h**) Waterfall chart, showing the best response of individual tumors, grouped by mouse. The mice are sorted based on the total best response. **i**) Time required for tumors to reach a size five fold greater than the size at inclusion for mice treated with EC (n = 9), ECD (n = 9) or ECR (n = 10). q-values: two-sided Mann-Whitney test with FDR correction. **j**) Best response achieved throughout treatment, calculated based on the total tumor burden for each mouse for mice treated with EC (n = 10), ECR (n = 9) and ECD (n = 10). q-values: two-sided Mann-Whitney test with FDR correction.

Filippo Beleggia
Christian Reinhardt
Elisa Motori
Matteo Bergami
Silvio Rizzoli

# Reporting Summary

## Statistics

For all statistical analyses, confirm that the following items are present in the figure legend, table legend, main text, or Methods section.

| n/a | Confirmed | |
|---|---|---|
| ☐ | ☒ | The exact sample size (*n*) for each experimental group/condition, given as a discrete number and unit of measurement |
| ☐ | ☒ | A statement on whether measurements were taken from distinct samples or whether the same sample was measured repeatedly |
| ☐ | ☒ | The statistical test(s) used AND whether they are one- or two-sided<br>*Only common tests should be described solely by name; describe more complex techniques in the Methods section.* |
| ☒ | ☐ | A description of all covariates tested |
| ☐ | ☒ | A description of any assumptions or corrections, such as tests of normality and adjustment for multiple comparisons |
| ☐ | ☒ | A full description of the statistical parameters including central tendency (e.g. means) or other basic estimates (e.g. regression coefficient) AND variation (e.g. standard deviation) or associated estimates of uncertainty (e.g. confidence intervals) |
| ☐ | ☒ | For null hypothesis testing, the test statistic (e.g. *F*, *t*, *r*) with confidence intervals, effect sizes, degrees of freedom and *P* value noted<br>*Give P values as exact values whenever suitable.* |
| ☒ | ☐ | For Bayesian analysis, information on the choice of priors and Markov chain Monte Carlo settings |
| ☒ | ☐ | For hierarchical and complex designs, identification of the appropriate level for tests and full reporting of outcomes |
| ☐ | ☒ | Estimates of effect sizes (e.g. Cohen's *d*, Pearson's *r*), indicating how they were calculated |

*Our web collection on statistics for biologists contains articles on many of the points above.*

## Software and code

Policy information about availability of computer code

| Data collection | Electron micrographs were acquired with DigitalMicrograph (Gatan). Electrophysiology data were acquired using Signal (version 6.0, Cambridge Electronic, Cambridge, UK), Hokawo (version 2.8, Hamamatsu, Geldern, Germany), Igor Pro (version 32 7.01, WaveMetrics, Lake Oswego, OR, USA), Clampex (version 10.7.0.3, Molecular Devices, LLC). Imaging during electrophysiological recording was acquired with Micro-Manager (version 2.0.0, Open Source, UCSF). Electron micrographs were taken with DigitalMicrograph v3.32.2403.0 (Gatan). Tomograms were acquired using SerialEM v3.7.11 |
|---|---|

| Data analysis | Genomic and expression data were processed using BWA version 0.7.15, samtools version 1.3.1, liftOver v385, GATK version 4.1.3.0, Annovar version 2018Apr16, STAR versions 2.4.2a, 2.5.3a and V_2.7.10b , HTSeq version 0.6.1p1, RNA-SeQC version 1.1.9. scRNAseq data was processed with the PARSE pipeline version 1.1.1. MRI images were analyzed with Horos version 3.0, with the package Export Rois version 2.0. Data was analyzed using Python version 3.8, 3.9 or 3.10 with the packages pandas version 1.1.4, numpy version 1.20.2, scipy version 1.6.3, statsmodels version 0.12.2, datashader version 0.12.1, matplotlib version 3.4.2, seaborn version 0.11.0, lifelines version 0.25.6, scanpy version 1.9.3, cellbender version 0.3.0, doubletdetection version 4.2. Analysis of electrophysiological data was performed with Clampfit (version 11.2.2.17, Molecular Devices, LLC). Pearson correlation analysis was performed with Matlab (The Mathworks, Inc., version 2023b). Tomograms were reconstructed with IMOD v4.11.7. The 3D reconstruction tomograms was performed with Imaris v10.2.0 (Oxford Instruments). Segmentation of EM structures was performed with Microscopy Image Browser (MIB, version 2.84). Registration of CLEM images was performed with the plugin EC-CLEM v1.1.0.0 from the software ICY v2.5.2.0. Image analysis was performed with ImageJ v1.54h, the Cell Counter plugin v3.0.0 and Fiji v2.14.0. Python scripts generated in this study are available from github (https://github.com/beleggia-lab/neuron-to-SCLC-synapses) and Zenodo (https://doi.org/10.5281/zenodo.15667860). |
|---|---|

For manuscripts utilizing custom algorithms or software that are central to the research but not yet described in published literature, software must be made available to editors and reviewers. We strongly encourage code deposition in a community repository (e.g. GitHub). See the Nature Portfolio guidelines for submitting code & software for further information.

## Data

Policy information about availability of data

All manuscripts must include a data availability statement. This statement should provide the following information, where applicable:
- Accession codes, unique identifiers, or web links for publicly available datasets
- A description of any restrictions on data availability
- For clinical datasets or third party data, please ensure that the statement adheres to our policy

Reference genomes were downloaded from GDC (TCGA GRCh38.d1.vd1, https://api.gdc.cancer.gov/data/254f697d-310d-4d7d-a27b-27fbf767a834 ), from Ensembl (https://www.ensembl.org, GRCm38.102 and GRCm39.110 ) and from GTEx (https://www.gtexportal.org, Homo_sapiens_assembly38_noALT_noHLA_noDecoy_ERCC.fasta). Gene annotations were downloaded from gencode (vM23 and v22, https://www.gencodegenes.org/). Orthology mapping was downloaded from the HGNC database (https://www.genenames.org/, downloaded January 6th 2020). Mutation data were downloaded from the supplementary tables of the referenced publications or from the CCLE website (Cell_lines_annotations_20181226.txt and CCLE_DepMap_18q3_maf_20180718.txt, https://portals.broadinstitute.org/ccle/). TCGA expression data were downloaded from the Genomic Data Commons Data Portal (v27, https://portal.gdc.cancer.gov). GTEx expression data were downloaded from the GTEx database (v8, https://gtexportal.org). Gene Ontology (GO) data were downloaded from the GO website (v2020-09-10, http://geneontology.org). ChIP-seq data were downloaded from the CISTROME database (v2, http://cistrome.org/db, accessed November 27th 2019). ScRNA-seq data, as well as the corresponding metadata were downloaded from Synapse (Synapse:syn21560406, https://www.synapse.org/) and CZ cellxgene (https://datasets.cellxgene.cziscience.com/7a30310a-2239-4d84-b99e-a12456c2fe19.h5ad). PhyloP conservation tracks across 470 mammalian genomes were downloaded from UCSC (hg38.470way.phyloP, https://genome.ucsc.edu/). Uniprot Knowledgebase annotations were downloaded from the Uniprot website (v2022_5, https://www.uniprot.org). Raw sequencing data from the piggyBac screen and murine snRNAseq are available through the Sequence Read Archive (SRA, https://www.ncbi.nlm.nih.gov/sra) under accessions PRJNA1275653 and PRJNA1276342, respectively. A scanpy data object of the snRNAseq dataset is available at Zenodo (https://doi.org/10.5281/zenodo.15647008). The full analyzed data from our whole-genome analyses are available in the supplementary information tables and the source data for all figures is provided in the online version of this manuscript.

## Research involving human participants, their data, or biological material

Policy information about studies with human participants or human data. See also policy information about sex, gender (identity/presentation), and sexual orientation and race, ethnicity and racism.

| Reporting on sex and gender | Human samples were not stratified by sex or gender |
|---|---|
| Reporting on race, ethnicity, or other socially relevant groupings | Human samples were not grouped based on race, ethnicity or other socially relevant grouping |
| Population characteristics | No covariate analysis was performed |
| Recruitment | Patients were recruited as part of the Biomasota study  (13-091, 2016) |
| Ethics oversight | Patients consented to the use of their tissue specimens and approval was obtained by the Ethics Committee of the University of Cologne |

Note that full information on the approval of the study protocol must also be provided in the manuscript.

# Field-specific reporting

Please select the one below that is the best fit for your research. If you are not sure, read the appropriate sections before making your selection.

☒ Life sciences          ☐ Behavioural & social sciences          ☐ Ecological, evolutionary & environmental sciences

For a reference copy of the document with all sections, see nature.com/documents/nr-reporting-summary-flat.pdf

# Life sciences study design

All studies must disclose on these points even when the disclosure is negative.

| | |
|---|---|
| Sample size | Sample sizes for animal studies was selected based on power analysis. Sample sizes for other experiments were chosen based on previous experience with cancer cell lines and neuronal cultures (e.g. PMID: 30612738, 25661179), in order to capture the technical and biological variability of the different experimental settings. All the sample sizes are stated in the figure legends, main text or methods sections. |
| Data exclusions | No data were excluded from analysis. |
| Replication | All experimental findings were reproducible across at least two replicates. |
| Randomization | The allocation of samples to experimental groups was randomized. The neuronal batches could not each be tested with every cell line but the allocation of cell lines to neuronal batches was random. |
| Blinding | The evaluation of termination criteria for survival analysis of the mice was not blinded as the scientists also performed treatments which could not be blinded due to the different dosing schedules. All other data collection and data analysis was performed blindly or with automated pipelines. |

# Reporting for specific materials, systems and methods

We require information from authors about some types of materials, experimental systems and methods used in many studies. Here, indicate whether each material, system or method listed is relevant to your study. If you are not sure if a list item applies to your research, read the appropriate section before selecting a response.

## Materials & experimental systems

| n/a | Involved in the study |
|---|---|
| ☐ | ☒ Antibodies |
| ☐ | ☒ Eukaryotic cell lines |
| ☒ | ☐ Palaeontology and archaeology |
| ☐ | ☒ Animals and other organisms |
| ☒ | ☐ Clinical data |
| ☒ | ☐ Dual use research of concern |
| ☒ | ☐ Plants |

## Methods

| n/a | Involved in the study |
|---|---|
| ☒ | ☐ ChIP-seq |
| ☒ | ☐ Flow cytometry |
| ☒ | ☐ MRI-based neuroimaging |

## Antibodies

| | |
|---|---|
| Antibodies used | Chicken anti-GFP 1:500 (Alves Labs, Cat# GFP-1020)<br>Rabbit polyclonal anti-RFP 1:500 (Rockland, Cat#600401379)<br>Chicken anti-MAP2 1:500 (Abcam, Cat# ab5392)<br>Mouse monoclonal anti-vGluT1 1:500 (Synaptic Systems, Cat# 135 311)<br>Rabbit polyclonal anti-Homer1 1:500 (Synaptic Systems, Cat# 160 003)<br>Mouse monoclonal anti-Bassoon 1:500 (Synaptic Systems, Cat# 141 111)<br>goat anti-CGRP 1:1000 (Abcam, #ab36001)<br>rabbit GAP43 1:2000 (Novus Biologicals, #NB300-143)<br>chicken anti-GFP 1:500 (Abcam, #13970)<br>rabbit PGP9.5 1:2000 (Abcam, #ab108986)<br>rabbit anti-P2X3 1:1000 (Chemicon, #AB5895)<br>rat anti-SP 1:200 (Biogenesis, 8450-0505)<br>guinea-pig anti-SYP 1:4000 (Synaptic Systems, #101002)<br>rabbit anti-VGluT1 1:250 (Synaptic Systems, #135303)<br>mouse anti-Bassoon 1:500 (Enzo, ADI-VAM-PS003-F, cat# SAP7F407)<br>AlexaFluor 488 Donkey anti-Chicken 1:1000 (Jackson Immuno Research Labs, Cat# 703-545-155)<br>AlexaFluor 546 Donkey anti-Rabbit 1:1000 (Thermo Fisher Scientific, Cat# A10040)<br>AlexaFluor 647 Donkey anti-Rabbit ICC-IF 1:1000; IHC-IF 1:500 (Jackson Immuno Research Labs, Cat# 711-605-152)<br>AlexaFluor 647 Donkey anti-Mouse 1:1000 (Jackson Immuno Research Labs, Cat# 715-605-150)<br>Biotinylated donkey anti-rabbit 1:500 (Jackson Immuno Research Labs, Cat#711-065-152)<br>Biotinylated donkey anti-rat 1:200 (Jackson Immuno Research Labs, Cat#712-065-150)<br>Alexa Fluor 647 donkey anti-chicken 1:400 (Jackson Immuno Research Labs, Cat#703-605-155)<br>Cy™3-conjugated Fab Fragment Donkey Anti-Rabbit 1:2000 (Jackson Immuno Research Labs, Cat#711-167-003)<br>(FITC)-conjugated donkey anti-rabbit 1:500 (Jackson Immuno Research Labs, Cat#711-095-152)<br>(FITC)-conjugated donkey anti-goat 1:500 (Jackson Immuno Research Labs, Cat#705-095-147)<br>Cy™3-conjugated donkey anti-goat 1:400 (Jackson Immuno Research Labs, Cat#705-165-147)<br>Cy™3-conjugated donkey anti-guinea-pig 1:400 (Jackson Immuno Research Labs, Cat#706-165-148)<br>Cy™3-conjugated streptavidin 1:6000 (Jackson Immuno Research Labs, Cat#016-160-084)<br>(FITC)-conjugated streptavidin 1:1000 (Jackson Immuno Research Labs, Cat#016-010-084)<br>chicken anti-MAP2 1:1000 (Novus biologicals, Cat# NB300-213) |

mouse mNeonGreen 1:500 (ChromoTek, Cat# 32F6)
mouse anti-SMI-312 1:1000 (HISS diagnostics, Cat# SMI-312R)
guinea pig VGluT1 1:500 (SySy, Cat# 135 304)
rabbit VGluT1 1:500 (SySy Cat# 135 308)
mouse anti-synaptophysin 1:100 (Leica Biosystems, Wetzlar, Germany, #PA0299)
mouse anti-neurofilament 200 kDa subunit 1:500 (NF-H) (Sigma, Saint Louis, MO, #N0142)
mouse anti-neurofilament 70 kDa subunit 1:500 (NF-L, Agilent, Santa Clara, CA, #M0762)
Mouse anti-SMI311 1:1000 (BIOZOL, Hamburg, cat#BLD-837801)
anti-GFP nanobody AF 488 1:500, (Nanotag, Göttingen, Cat# N0301)
Alpaca anti-VGluT1 nanobody 1:500 (Nanotag, Göttingen, cat# N1602-AF568-L)
goat anti-chicken Alexa 405 1:500 (Abcam Cambridge UK, cat# ab175674)
goat anti-rabbit STAR635P 1:1000 (Abberior, Göttingen Germany, ST635P Cat# 1002-500UG)
anti-mouse Alexa 750 1:1000 (ThermoFisher, Waltham USA, cat# A21037)
Alexa Fluor 568-conjugated polyclonal goat ant-rabbit 1:1000 (Life Techn. Carlsbad USA Cat #A-11036)
Alexa Fluor 750-conjugated polyclonal goat anti-guinea pig IgG H&L 1:500 (Abcam, #ab175758)
Alexa Fluor 568-conjugated polyclonal goat anti-mouse 1:1000 (Invitrogen #a-11004)

Validation

Chicken polyclonal anti-GFP, validated in previous refs, reported in manufacturer´s page https://www.antibodiesinc.com/products/anti-green-fluorescent-protein-antibody-gfp
Rabbit Polyclonal anti-RFP, validated in previous refs, reported in manufacturer´s page https://www.rockland.com/categories/primary-antibodies/rfp-antibody-pre-adsorbed-600-401-379/
Chicken polyclonal anti-MAP2, validated in previous refs, reported in manufacturer´s page https://www.abcam.com/en-us/products/primary-antibodies/map2-antibody-ab5392
Mouse polyclonal anti-vGlut1, validated in previous refs and in KO samples, reported in manufacturer´s page https://sysy.com/product/135311
Rabbit polyclonal anti-Homer1, validated in previous refs, reported in manufacturer´s page https://www.sysy.com/product/160003
Mouse polyclonal anti-Bassoon, validated in previous refs and in KO samples, reported in manufacturer´s page https://sysy.com/product/141111
CGRP antibody (Go Pc; Abcam, #ab36001) was previously validated on mouse lungs (10.1186/s12931-018-0915-8)
GAP43 antibody (Rb Pc; Novus Biologicals, #NB300-143) was validated by the manufacturer on mouse samples (https://www.novusbio.com/products/gap-43-antibody_nb300-143)
GFP antibody (Ch Pc; Abcam, #13970) was validated by the manufacturer on mouse samples (https://www.abcam.com/en-us/products/primary-antibodies/gfp-antibody-ab13970)
PGP9.5 antibody (Rb Pc; Abcam, #ab108986) was validated by the manufacturer on mouse samples (https://www.abcam.com/en-us/products/primary-antibodies/pgp95-antibody-epr4118-neuronal-marker-ab108986)
P2X3 antibody (Rb Pc; Chemicon, #AB5895) and SP antibody (Ra Mc; Biogenesis, 8450-0505) were previously validated on mouse lungs (10.1007/s00418-008-0495-7)
P2X3 antibody (Rb Pc; Chemicon, #AB5895) and SP antibody (Ra Mc; Biogenesis, 8450-0505) were previously validated on mouse lungs (10.1007/s00418-008-0495-7)
SYP antibody (GP Pc; Synaptic Systems, #101002) was validated by the manufacturer on mouse samples (https://sysy.com/product/101002)
VGLUT1 antibody (Rb Pc; Synaptic Systems, #135303) was validated by the manufacturer on mouse samples (https://sysy.com/product/135303)
Bassoon Mouse, 1:500, Enzo (New York, USA), ADI-VAM-PS003-F, cat# SAP7F407, validated 73 times, most recent: Yamamoto et. al., 2022 - Cell Biol.
AlexaFluor 488 anti-Chicken, validated in previous refs, reported in manufacturer´s page  https://www.jacksonimmuno.com/catalog/products/703-545-155
Alexa 546 anti-Rabbit, validated in previous refs, reported in manufacturer´s page   https://www.thermofisher.com/antibody/product/Donkey-anti-Rabbit-IgG-H-L-Highly-Cross-Adsorbed-Secondary-Antibody-Polyclonal/A10040
Alexa 647 anti-Rabbit, validated in previous refs, reported in manufacturer´s page  https://www.jacksonimmuno.com/catalog/products/711-605-152
Alexa 647 anti-Mouse, validated in previous refs, reported in manufacturer´s page https://www.jacksonimmuno.com/catalog/products/715-605-150
Biotinylated donkey anti-rabbit IgG (1:500) validated in previous refs reported in manufacturer's page (https://www.jacksonimmuno.com/catalog/products/711-065-152)
Biotinylated donkey anti-rat IgG (1:200) validated in previous refs reported in manufacturer's page (https://www.jacksonimmuno.com/catalog/products/712-065-150)
Alexa Fluor® 647 donkey anti-chicken IgY (IgG) (1:400) validated in previous refs reported in manufacturer's page (https://www.jacksonimmuno.com/catalog/products/703-605-155)
Cy™3-conjugated Fab Fragment Donkey Anti-Rabbit IgG (1:2000) validated in previous refs reported in manufacturer's page (https://www.jacksonimmuno.com/catalog/products/711-167-003)
Fluorescein (FITC)-conjugated donkey anti-rabbit IgG (1:500) validated in previous refs reported in manufacturer's page (https://www.jacksonimmuno.com/catalog/products/711-095-152)
Fluorescein (FITC)-conjugated donkey anti-goat IgG (1:500) validated in previous refs reported in manufacturer's page (https://www.jacksonimmuno.com/catalog/products/705-095-147)
Cy™3-conjugated donkey anti-goat IgG (1:400) validated in previous refs reported in manufacturer's page (https://www.jacksonimmuno.com/catalog/products/705-165-147)
Cy™3-conjugated donkey anti-guinea-pig IgG (1:400) validated in previous refs reported in manufacturer's page (https://www.jacksonimmuno.com/catalog/products/706-165-148)
Cy™3-conjugated streptavidin (1:6000) validated in previous refs reported in manufacturer's page (https://www.jacksonimmuno.com/catalog/products/016-160-084)
Fluorescein (FITC)-conjugated streptavidin (1:1000) validated in previous refs reported in manufacturer's page (https://www.jacksonimmuno.com/catalog/products/016-010-084)
chicken anti-MAP2, validated in previous refs, reported in manufacturer's page https://www.novusbio.com/products/map2-antibody_nb300-213
mouse mNeonGreen validated in previous refs, reported in manufacturer's page https://www.ptglab.com/products/mNeonGreen-

antibody-32F6

SMI312 Mouse, 1:1000, HISS Diagnostics/Covance (Freiburg, Germany), cat# SMI-312R, validated in 100 citations, most recent: Abedin MJ, et al. 2023. Front Bioeng Biotechnol.

guinea pig VGluT1, SySy, Cat# 135 304, validated in manufacturer's page https://www.sysy.com/product/135304

rabbit VGluT1, SySy Cat# 135 308, KO validated, reported in manufacturer's page: https://www.sysy.com/product/135308

mouse anti-synaptophysin, Leica Biosystems, #PA0299 validated, reported in manufacturer's page https://shop.leicabiosystems.com/ihc-ish/ihc-primary-antibodies/pid-synaptophysin

mouse anti-neurofilament 200 kDa subunit NF-H, Sigma, #N0142, validated in 425 citations, most recent: Ke et al., 2025 CNS Neuroscience & Therapeutics

mouse anti-neurofilament 70 kDa subunit, NF-L Agilent, #M0762, validated in manufacturer's page https://www.labome.com/product/Dako/M0762.html

SMI311 Mouse, 1:1000, BIOZOL (Hamburg, Germany), cat# BLD-837801, validated in 17 citations, most recent: Yang J, et al. 2023. Brain Sci.

anti-GFP nanobody AF 488, Nanotag, Cat# N0301, validated in 23 citations, most recent: Shaib A et al, (2024) Nat. Biotec.

vGlut1 nbAZDye568 Alpaca, 1:500, Nanotag (Göttingen, Germany), cat# N1602-AF568-L, verified in 8 citations, most recent: Mougios et al., 2024 Nat. Com.

goat anti-chicken Alexa 405, Abcam Cambridge UK, cat# ab175674, validated, reported in manufacturer's page https://www.abcam.com/en-us/products/secondary-antibodies/goat-chicken-igy-h-l-alexa-fluor-405-ab175674

goat anti-rabbit STAR635P, Abberior, ST635P Cat# 1002-500UG validated in manufacturer's page https://abberior.shop/abberior-STAR-635P-goat-anti-rabbit-IgG-500-ll-1-mg-ml

anti-mouse Alexa 750, ThermoFisher, Waltham USA, cat# A21037) validated, reported in manufacturer's page: https://www.thermofisher.com/antibody/product/Goat-anti-Mouse-IgG-H-L-Cross-Adsorbed-Secondary-Antibody-Polyclonal/A-21037

Alexa Fluor 568-conjugated polyclonal goat ant-rabbit IgG H&L, cited in 2220 publications, with 55 published images. Applications used for ICC-IF, and IHC.

Alexa Fluor 750-conjugated polyclonal goat anti-guinea pig IgG H&L, from Abcam antibodies, cited in one publication (PMID: 33789950, DOI: 10.1681/ASN.2020101459). Applications include WB, ICC/IF, ELISA, IHC-P, Flow Cyt, IHC-Fr.

Alexa Fluor 568-conjugated polyclonal goat anti-mouse antibody, supplied by Invitrogen Antibodies, cited in 2603 publications, with 209 published images. Applications used include ICC-IF, IHC, IHC-IF, and IF.

# Eukaryotic cell lines

Policy information about cell lines and Sex and Gender in Research

| | |
|---|---|
| Cell line source(s) | Murine cell lines (AVR424.3 and RP1462) were isolated from murine tumors in the RP line, human cell lines (COR-L88, H1836, H69, H146, DMS273, H524, H211, H526, H1975, HCC44, HOP62, H2291, HEK293T) were gifts from professor Roman Thomas |
| Authentication | Cell lines were authenticated through genotyping (murine lines) and STR profiling (human lines). |
| Mycoplasma contamination | All cell lines tested negative for mycoplasma contamination |
| Commonly misidentified lines (See ICLAC register) | No commonly misidentified lines were used |

# Animals and other research organisms

Policy information about studies involving animals; ARRIVE guidelines recommended for reporting animal research, and Sex and Gender in Research

| | |
|---|---|
| Laboratory animals | Animal experiments were performed with adult male and female mice derived from the following lines: Rb1-flox, Trp53-flox, Rosa26-LSL-PB, ATP1-S2, ATP1-H39, Thy1-GFP-M, Rosa26-Cas9-GFP, Rbl2-flox,Rosa26-LSL-tdTomato, H11-LSL-Cas9 and wild type C57BL/6. Wild type C57BL/6 mice were also used for neuronal preparation (adult mothers and embryos E13.5-16.5) and nodose ganglia preparations (3-5 weeks old mice). |
| Wild animals | This study did not involve wild animals |
| Reporting on sex | This study included animals of both sexes, but was not powered to detect sex-specific effects. |
| Field-collected samples | This study did not involve field-collected samples |
| Ethics oversight | The animals experiments performed in Cologne, Germany were approved by the Landesamt für Natur, Umwelt und Verbraucherschutz Nordrhein-Westfalen. For animal experiments performed at Stanford University, mice were maintained according to practices approved by the NIH, the Stanford Institutional Animal Care and Use Committee (IACUC), and the Association for Assessment and Accreditation of Laboratory Animal Care (AAALAC). The study protocol was approved by the Stanford Administrative Panel on Laboratory Animal Care (APLAC) (protocol 13565). |

Note that full information on the approval of the study protocol must also be provided in the manuscript.

## Plants

**Seed stocks**

*Report on the source of all seed stocks or other plant material used. If applicable, state the seed stock centre and catalogue number. If plant specimens were collected from the field, describe the collection location, date and sampling procedures.*

**Novel plant genotypes**

*Describe the methods by which all novel plant genotypes were produced. This includes those generated by transgenic approaches, gene editing, chemical/radiation-based mutagenesis and hybridization. For transgenic lines, describe the transformation method, the number of independent lines analyzed and the generation upon which experiments were performed. For gene-edited lines, describe the editor used, the endogenous sequence targeted for editing, the targeting guide RNA sequence (if applicable) and how the editor was applied.*

**Authentication**

*Describe any authentication procedures for each seed stock used or novel genotype generated. Describe any experiments used to assess the effect of a mutation and, where applicable, how potential secondary effects (e.g. second site T-DNA insertions, mosiacism, off-target gene editing) were examined.*

