## [Peer Review File · Nature]

Functional synapses between neurons and small cell lung cancer

Corresponding Author: Dr Filippo Beleggia

Version 0:

Reviewer comments:

Referee #1

(Remarks to the Author)

The manuscript of Schmitt et al adds a relevant new angle to the emerging field of cancer neuroscience: the potential existence of neuron-cancer synapses outside the brain, and outside gliomas. A specific strength are the broad genetic screening methodologies and gene network analysis approaches applied here. The therapeutic outlook with 2 clinical applicable agents interfering with glutamatergic signaling that have antitumor effects in their SCLC mouse model is very interesting, too. The manuscript is well written.

The weakness is the lack of a true proof for functional / bona fide synapses (see below), which is certainly critical; and also a lack of data showing which neurotransmitter receptor class (most important) and which downstream mechanisms (of secondary importance) exactly convey the protumor effects. The manuscript is meandering between data in support of AMPA receptors and/or NMDA receptors and/or metabotropic glutamate receptors. A clear line would be important, and distinct investigations of all potential pathways in the different experimental settings. Experiments to distinguish true synaptic from paracrine effects would be very important, too (e.g. by genetic interference with synapse-relevant genes in SCLC cells). Further details are given below.

First and foremost, the authors need to clearly demonstrate neuron-cancer cell synapses. First, this needs to be done by proper electron microscopy: in Ext Data Fig. 8i-l, they show good-quality EM images of mouse tumors, demonstrating that they are capable of doing these kind of experiments. Here, no synapse is visible, particularly no synapse between putative neuronal and putative cancer cell structures. The authors need to search for those and demonstrate them. They also need to better prove / demonstrate the nature of the presynaptic structure (clear neuronal structure) and the postsynaptic one (clear cancer cell structure), e.g. by immunogold stainings if necessary. Second, their functionality needs to be better demonstrated by proper electrophysiological experiments, in vitro and ideally also on acute slices ex vivo. The electrophysiological characterizations provided here are quite limited and need to be improved. First, why not inhibiting with CNQX OR AP5 to understand whether AMPAR OR NMDARs are involved here? Second, co-culture of SCLC cells with cortical neurons might create artificial glutamatergic synaptic contacts. It would therefore be particularly important to recapitulate these findings with PNS neurons that are found in the primary tumor microenvironment in SCLC. Third, it needs to be shown that inhibition of neuronal activity (eg by TTX) is also completely inhibiting the sPSCs. Fourth, a stimulation paradigm with EPSC analysis for the most important conditions would be important, too. Last but not least, investigations on tissue and not simple in vitro co-cultures would greatly increase the validity of the findings for the disease. The authors perform this for acute brain slices, but only with a very limited approach, and with a somewhat artificial model (direct implantation of cancer cells in the brain, no true brain metastasis model). They should perform this in their primary tumors, too.

In line, the images of Fig. 3d-f show no clear synaptic structures. The authors should apply superresolution fluorescent microscopy (or EM, see above) to make their point of synaptic contacts between CNS neurons and cancer cells. The interpretation of lack of nerve structures in the tumor mass and a neurotoxic effects is questionable – it is very well possible that simple expansive growth of a rapidly proliferation tumor mass is causing this phenomenon.

The growth benefit from co-culture with neurons (entire Fig. 4) is of course supportive but not really new, and not informative for the mechanism that is in place here.

Regarding the differences between the large groups (n= relevant for each group) of untreated, chemo-treated and checkpoint inhibitor treated mice: the authors argue they did not detect significant differences between the groups (please clarify better by statistical comparisons etc.) in the first place and therefore pooled these groups. However, it would be very interesting and important for the study to understand whether the distinct neuronal features / genes identified are different between those groups. This should be analyzed. The background is that neuronal features of tumors can vary with treatment; in fact, there is published data showing just that (increase of those features with treatment / resistance / relapse).

Moreover, the authors analyzed bulk tumor material, not single cell data that allows to clearly differentiate between cancer cells and nonmalignant (particularly neural) cells. It would greatly increase the value of the genomic and particularly the gene expression data if single cell data of tumor cells vs non-tumor cells would be analyzed, too, in man and mouse.

To get a better idea of the uniqueness of the neuronal phenotypes / synaptic functions detected in the genomic screen, it would be important to see the results and statistical testings in more granularity (better than shown in Fig. 1b, at least additional to Fig. 1b), not just the raw data as shown in Ext Data Tables 3,4.

Co-culture of human SCLC lines were performed only with murine cortical neurons, not peripheral nervous system cell types or human neurons. This needs to be extended.

The retrograde monosynaptic tracing experiments are a nice way to study the neuron-cancer cell connections/networks, although not finally proving the existence of bona fide synapses between the cell types. However, I am missing an important experimental validation: please provide dynamic (time-lapse) data of cancer cells infected first, and neurons following. Finally, the number of independent experiments, ROIs analyzed, with number of cells etc. is not really clear here. This also applies to many other figures from Fig. 2 on and should be corrected. All figure legends should contain exact information about the n=.

The lack of specificity of the CGRP staining makes it difficult to draw any meaningful conclusions from e.g. Ext Data Fig. 7. Other stainings for PNS neuronal types that are well established should be employed that allow a better identification of PNS neuron classes, and to separate them from cancer cells and healthy PNECs. This is actually tried in Fig. 3a-c and Ext Data Fig. 8. However, all this data lacks proper quantifications, and also staining controls – I am not sure what I really see on many of those images.

The entire brain metastasis data is not very strongly developed and not very well connected to the primary tumor data.

Minor points:

Ext Data Fig. 1c: comparison with human histology would be informative, and also arrows depicting what we see and why this is typical for SCLC.

When I understand the work correctly, the two 2019 Nature publications reporting bona-fide synapses between neurons and postsynaptic cancer cells (glioblastoma, K27M mut midline glioma) show that brain tumors can receive synaptic input, mainly to so-called “tumor microtubes” and tumor cell networks that are driven by neuronal features (Osswald et al Nature 2015). This picture has been further developed now by also demonstrating neuron-cancer synapses on invasive (non-network-connected) tumor cells in glioblastoma that reveal distinct neuronal features on multiple levels (Venkataramani et al Cell 2022). Together these four publications make a strong case that tumor cells themselves can exert distinct and even multiple neuronal features, and that those features appear to be a strong determinant of the formation neuron-cancer synapses. The authors should discuss this in more depth and relate this to their scientific question and findings. They should also include a short discussion of the Hanahan et al Nature 2019 publication and make clear that those are not bona-fide synapses but rather cancer cells in perisynaptic positions being stimulated by glutamate from the synaptic cleft. They should consider to restructure the introduction and avoid a lengthy introduction into the genetics and subclasses of SCLC without providing clear guidance to the reader why this is of importance for the work. The authors might want to consider to focus more on neuron-cancer synapses in this respect. The introduction needs to be shortened anyway for this journal’s format.

Referee #2

(Remarks to the Author)

The study by Schmitt et al uses an innovative genome-wide piggyBac transposase system in the context of the Rb1/p53 (RP) model of small cell lung cancer (SCLC) to identify genes involved in synaptogenesis as potential regulators of SCLC. Importantly, the analysis is rigorous using two distinct transposon models, analyzing 303 tumors from 58 mice under various treatment conditions including metastases. The screen is clearly successful, evident by the identification of key genes known to be functionally relevant in SCLC biology, including Trp73, Pten, and Crebbp (Cui et al, Mol Can Res, 2014; Jia et al, Cancer Discovery 2018; George et al, Nature, 2015). The new information provided by this study is that synaptogenesis genes such as Nlgn1, Nrnx1, Reln, and Dcc exhibit recurrent transposon integrations suggesting a potential functional role in SCLC—although these genes are never functionally interrogated. However, based on these findings, the authors determine that SCLC cells form functional synaptic inputs using a co-culture system of murine cortical neurons and human SCLC cell lines. They demonstrate that the presence of neurons promotes SCLC growth with multiple cell lines. Interestingly, they suggest that in the RP mouse model, small SCLC tumors have evidence of nerve fibers connecting with

tumors, but large tumors exclude nerve fibers. The absence of nerve fibers in large tumors was confirmed in an independent allograft experiment as well. This may suggest that innervation of SCLC is particularly relevant to tumor initiation, but not necessarily progression. The authors also find that a large class of glutamate receptors have transposon insertions, and they focus on GRM8, an inhibitory metabotropic glutamate receptor, but do not genetically perturb GRM8 to test its function. They conclude showing that the pharmacological inhibitor riluzole has a modest impact on tumor growth in the RP model alone and in combination with standard-of-care chemotherapy.

The authors employ innovative approaches toward an interesting topic not well understood in SCLC, and some of the experiments are rigorous. However, a number of conclusions are premature without further studies, and overall lack functional interrogation or mechanistic insights into the phenomenon described:

Major Points:

1. The phrase in the Results section following Figure 1, "Therefore, our genetic screen.... suggests a novel role of synaptic genes in SCLC" is over-stated at this point in the text. It is possible that expressed genes may have more integrations than non-expressed genes, without a functional role for these genes, which should be demonstrated. Genetic perturbation of *Nlgn1*, *Nrxn1*, *Reln*, or *Dcc* and its consequences on SCLC growth or phenotype (in cell lines or a relevant tumor initiation setting such as the pre-SC system; Kim et al, *Genes Dev*, 2016) would support these conclusions. Otherwise, it seems possible that mutations in these genes are simply passenger effects in genes that are highly expressed due to cell fate, but not necessarily functionally relevant.
2. As previously shown by multiple groups, neuroendocrine and neuronal genes are more commonly associated with the ASCL1 and NEUROD1 subtypes of SCLC (Rudin et al, *Nat Rev Cancer*, 2019). It is not mentioned what subtypes of cell lines are used for this study such as COR-L88, DMS273, H146, H69, H524, etc. It would be valuable to know whether functional synapses are formed with all subtypes, or just neuroendocrine/neuronal subtypes of SCLC? Does co-culture with neurons promote growth of all SCLC subtypes, or just SCLC-A and SCLC-N? This question speaks to the applicability of these findings.
3. The authors show in multiple cell lines that the presence of neurons promotes proliferation of SCLC cell lines, but the mechanisms are still entirely unknown. Is physical connection with neurons required for the pro-growth effect, or is supernatant from neurons or neuron+SCLC co-cultures sufficient to promote SCLC growth? This clue toward mechanism may be important for interpreting pharmacological *in vivo* experiments where effects on small, innervated tumors may differ from large, nerve-excluded tumors. Likewise, is the pro-growth effect specific to neurons? Have the authors studied the growth rate of SCLC cells in fibroblast co-culture, upon blockade of neuronal contacts, and/or as a function of cell:cell density? Is the pro-growth effect specific to neuroendocrine-high SCLC, or do neurons also promote the growth of lung adenocarcinoma or non-neuroendocrine SCLC?
4. Related to Figure 6, pharmacological studies are used in an attempt to modulate GRM8 function, specifically with DCPG and riluzole. While DCPG is a selective GRM8 agonist, riluzole appears to be quite non-specific and has many consequences, and its function in this context is unknown. This is important because the *in vivo* studies suggest that riluzole slows tumor growth and extends survival more than DCPG, especially when used in combination with chemotherapy where DCPG doesn't have a significant impact on tumor growth (Fig. 6c). It would be much more compelling to perform genetic knockout or overexpression studies with GRM8, and to test the impact of riluzole on SCLC tumor cells in the presence and absence of GRM8. It would also be informative to determine the impact of the GRM8 agonist on synaptic connections and growth rate in the neuron/SCLC co-culture model.
5. The RP mouse model used employs Ad-CMV-Cre to initiate SCLC, which may hit some NE cells, but undoubtedly hits other non-NE cell types. This is consistent with the observation that some tumors develop in peripheral locations that should lack normal NE cells (as in Ext. Data Fig. 1c). Some studies demonstrate that cell of origin impact the metastatic phenotype of SCLC (Yang et al, *Cancer Discovery*, 2018), while others demonstrate that cell of origin impacts tumor latency, transcriptional profile, and histopathology (Sutherland, *Cancer Cell*, 2011; Olsen et al, *Genes Dev*, 2021; Chen et al, *Oncogene*, 2022). The use of CMV-Cre contributes to tumor heterogeneity and potentially to pharmacological responses in Figure 6. Performing the studies in Figure 6 using CGRP-Cre to restrict tumorigenesis to NE cells would significantly control for these effects.
6. As the survival impact of riluzole in Figure 6 is relatively modest, it would be helpful to see representative images of MRI data, and to analyze the average tumor burden per animal rather than just the combination of individual tumors in Fig. 6a/c. It would be helpful to convey the amount of tumor heterogeneity per animal, as differing cells of origin (as in Point #5) may be one source of heterogeneity.

Minor:

1. While Rudin et al is a thorough review, it may not be sufficient to cover all of the points in the opening paragraph.
2. Introduction: "SCLC derives from pulmonary neuroendocrine cells" is over-stated and needs references. It has been shown that other cells can serve as a cell of origin for SCLC and this is highly influenced by genetic changes (Sutherland et al, *Cancer Cell*, 2011; Olsen et al, *Genes Dev*, 2021; Chen et al, *Oncogene*, 2022). It is more accurate that SCLC highly resembles PNECs, and may often derive from them, but we also know that adenocarcinoma can transition to SCLC (Oser et al, *Lancet Oncology*, 2015), making it clear that a tumor does not have to originate in a PNEC to become PNEC-like.
3. A number of figure legends are missing statistical tests used including Fig 2c/f, Fig 3i, Fig 5, etc.
4. Figure 4e: typo in the figure "colture" instead of "culture"

5. Figure 4e: This form of analysis as a group would be more accurate to provide individual cell line comparisons, with each cell line by itself plus and minus neurons. Statistical method and p value or q value (figure vs legend) should be clarified.
6. Figure 5a: Can you apply the NE signature used in Zhang et al, TCCR, 2018 which is a 50 gene signature that is more robust than the current 5 gene NE signature?
7. Extended Data Figure 1a: The unrecombined Trp53 allele from Meuwissen et al, Cancer Cell, 2003 has loxP sites flanking exons 2-10, thus the last exon after the LoxP site should be Exon 11 instead of 10.
8. Extended Data Figure 1d: The authors did not comment on whether any Piggybac mutations were enriched in metastases compared to primary tumors. This may be particularly interesting, as some genes do have migration-related functions such as Reelin and Netrin signaling.
9. Discussion: "As SCLC is a cancer entity...high degree of inter- and intratumor heterogeneity and plasticity" is missing Ireland et al, Cancer Cell, 2020 as a key reference.

Referee #3

(Remarks to the Author)

Schmitt and colleagues examine genetic parameters contributing to the growth of small cell lung cancer using mouse and human models/data. The use of an elegant piggyBAC mutagenesis screen in a mouse model to confirm previously known genes with contributions to tumor growth and identify – surprisingly – neuronal axon guidance and synaptogenic signaling components. Using in vitro experiments, the authors demonstrate that mouse cortical neurons can form synapse-like contacts onto human SCLC cell lines.

The notion that neuron-tumor interactions contribute to tumor growth has recently attracted a lot of attention. The strongest evidence supporting such a role is derived from work on gliomas. Thus, glioma cells have been demonstrated to express critical synaptogenic molecules and glutamate release from neurons has been demonstrated to promote tumor cell proliferation and enhance invasion (Venkataramani et al. 2019; Venkatesh et al., 2019; Venkataramani et al., 2019, 2022) and there is data published in preprint form suggesting a related mechanism for SCLC tumors (Savchuk et al., biorxiv). However, almost all of this work relies on gain-of-function manipulations such as non-physiological optogenetic stimulation. The study by Schmitt and colleagues stands out as it uses both genetic and pharmacological loss-of-function approaches to investigate molecular and functional components that modify tumor progression.

The work is very interesting and seems overall well-controlled, well-documented and the manuscript is very well written. There are several points where the individual observations of the study are not well connected. Addressing these points would significantly improve the study:

- 1) A major question is regarding the logic connecting the loss-of-function screen and follow-up experiments. Identifying synapse-components in the piggyBAC screen is intriguing – but this would indicate that loss of synapse proteins enhances tumor progression. However, the remainder of the study suggests the opposite, i.e. that synapse formation and synaptic transmission onto SCLC cells promotes tumor growth. How can this be reconciled? Does this relate to differential action of glutamatergic versus GABAergic receptors? Many components of the NRXN-NLGN system are shared between glutamatergic and GABAergic synapses. Moreover, the authors also identify GABA-specific molecules (CLSTN2, GABRB1).
- 2) The authors hypothesize that GRM8 is a major factor and provide evidence that it might serve as a therapeutic target.
 - a. For coherence of the data presentation, it would be valuable to show the insertion data for GRM8 in Figure 1 and Extended data figure 2.
 - b. Are the insertions thought to result in a gain or loss of function?
 - c. Do the SCLC cell lines used express GRM8 and does GRM8 inhibition modify growth of these cells in co-culture?
- 3) Fig.2a: These experiments are not straightforward to interpret. The neuronal density applied in this study, light microscopy does not allow to unambiguously identify direct contacts and to exclude the presence of dendritic processes. It is visible from the images provided that there are many vGLUT1-positive structures forming on the dendrites, i.e. non-tumor cells in the field of view. Ultrastructural investigations would be ideal to resolve this. At the very least, the authors should include the MAP2 marker in their high magnification views to confirm that there are no MAP2-positive dendrites between neuron and tumor cell.
- 4) Fig.2b. Rabies tracing is not a suitable method to assess function of synapses as stated in the text and not even a good method to confirm synaptic structures. There is increasing evidence that rabies viruses can also spread through non-synaptic membrane contacts, and in in vitro preparations this is an even bigger issue. The authors need to modify the text accordingly and / or consider moving this information to the supplement to not unintentionally mis-lead an uninformed reader.
- 5) Fig.2d-f. The electrophysiological assessments are much more convincing than the anatomical and rabies data. Considering that several of the genes identified in the piggyBAC screen are GABAergic synapse components, the authors should perform additional recordings to separately isolate glutamatergic and GABAergic currents. This is important with regards to the interpretation of the data overall (see below).
- 6) The authors provide in vivo data on SCLC innervation using allograft experiments. The data on nerve fibers and endings in proximity of tumors in the lung are less compelling. I appreciate the inclusion of EM data – but at present these are not

very convincing and it remains unclear at what frequency such structures were observed. Any experiments strengthening this would greatly increase the impact of this work. This could be done by additional ultrastructural investigations or (if a suitable cre line is available) re-deploying the rabies approach which should have more validity in an in vivo setting than in vitro.

7) Fig.4. The cell proliferation effect observed in mono- versus co-culture is interesting. However, this analysis remains somewhat preliminary. There is no evidence that this effect is specific to SCLC cells (would another cell line also increase proliferation in co-culture?) or that it relates to synaptic transmission from neurons. One way to assess the latter possibility would be blocking sodium channel-dependent action potentials with TTX, or – in line with the subsequent experiments – apply GRM8 inhibitors to test whether these effects are related. I appreciate that in this reductionist system inhibitors may not provide the data that are expected based on the other observations in more intact system. In this case, it might be worth considering to remove this experiment from the manuscript. As is, it seems too preliminary for a main figure.

Version 2:

Reviewer comments:

Referee #1

(Remarks to the Author)

I would like to thank the authors for answering my questions and for addressing my concerns with a large series of additional, meaningful experiments which made the manuscript significantly stronger.

My main remaining concern would be that the existence and/or role of synaptic communication (or other form of communication) between cancer cells and neurons in the primary tumor, the site of relevance for the genetic screening, are still undefined. The ultrastructural and electrophysiological data (and rabies tracing data) have all been performed in cultured brain neurons or in the mouse brain. The authors now add co-culture experiments with nodose ganglia, which is laudable, but this data is restricted to co-culture experiments (Fig. 5j,k). Do the authors find indications for synaptic (or other) contacts between these peripheral neurons and cancer cells? Did they try to detect neuron-cancer synapses in SCLC tumors, and/or performed electrophysiological experiments in these conditions? They should at least discuss these points/limitations.

Other points: can you provide the exact statistics for Fig. 4c and f?

How do the authors know that the cell depicted in the EM image is a cancer cell? Can they e.g. provide clear evidence from correlative light/electron microscopy? Can they provide quantifications of spatial parameters and number of synapses detected (how many per cancer cell / region)?

Referee #2

(Remarks to the Author)

Reviewer #2:

Overall, the authors have been thoroughly responsive to prior critique and made substantial improvements to the manuscript including:

- an improved organization and flow;
- clarified and more precise language that improves the accuracy of the claims;
- Related to Figure 5 and Ext Data Fig 14, they provide a more comprehensive assessment of SCLC subtype in neuronal co-culture including comparison to high density conditions and fibroblasts, non-neuronal cancer types, and examining conditioned media vs direct contact. These data suggest direct contact with neurons facilitates SCLC growth more than cell density or fibroblasts or paracrine factors alone. The figure legends still lack key details, which should be added.
- Related to Figure 1 and Extended Data Fig 5, they provide new data validating a gene from the Sleeping Beauty genetic screen (Reelin), suggesting Reelin is a tumor suppressor in SCLC, but animal numbers are lacking from this analysis, so it's hard to assess rigor.
- There is an appropriate de-emphasis on Grm8. The Grm8 data is somewhat preliminary (lack of functional genetic tests), and given the lack of effects of the GRM8 agonist in vivo in two mouse models, it is appropriate to de-emphasize any claims on GRM8.
- Related to Figure 6, the in vivo drug studies are now more rigorous, and the effects of riluzole (inhibitor of glutamate release) are now tested in RP mice from both CMV and neuroendocrine cells of origin with large animal numbers. These results show a modest but consistent and significant impact on tumor burden and overall survival. While it is not clear what this drug is doing, the results are overall consistent with the notion that neuronal interactions support SCLC growth.

I find minor errors/omissions that can be easily corrected:

1. New Extended Data 5c-g is missing animal numbers analyzed, so there's a lack of rigor without those details.
2. Extended Data Figure 20a appears to denote tumor number but does not include animal number.
3. Figure 5 legend lacks key details such as time points (Fig 5a-d), replicates (Fig 5e), etc.
4. Figure Legend Fig 1d, two typos, should be "ChIP-seq"

5. The authors may want to note that GRM8 was identified as an ASCL1 and NEUROD1 ChIP-seq target gene in human SCLC cell lines from Borromeo et al, Cell Rep, 2016. Moreover, analysis of data from Olsen et al, Genes Dev, 2021, suggest that Grm8 expression is significantly lost upon Ascl1 deletion in RPM tumors, consistent with being an ASCL1 target gene. Analysis in NIH SCLCCellMiner also suggests a correlation with ASCL1 and GRM8 in human SCLC cell line datasets.

6. SCLC-P is usually much less neuroendocrine/neuronal but still responds quite well to neurons in culture (Fig 5f) in some cases, so cells may not necessarily be neuronal in phenotype to benefit from neuronal interactions.

Referee #3

(Remarks to the Author)

The authors have made a substantial effort to address the comments made in the review. In particular, the EM and light microscopy analysis is substantially improved and now provides good evidence for synapse-like structures between neurons and SCLC tumor cells. On the other hand, some of the newly added electrophysiology experiments are raising new questions about the validity of the working hypothesis.

Specific Points:

Extended Data Figure 10: the new data on AMPA- and NMDA-receptor expression is not adding much value. From the staining presented, it is unclear whether this is specific immune-reactivity or some unspecific labeling. Examining these receptors would make sense when probing their concentration at neuron-tumor cell contacts. However, this was not done here. I suggest extending or removing this data.

The new EM analysis is a valuable extension of the previous dataset. The authors refer to "docked or fusing vesicles" in the text. I strongly recommend removing such statements. The experiments were done with chemical fixation. Over the past decade, it has become clear that this methods produces substantial artifacts with respect to the positioning of vesicles in the presynaptic terminal. Only high-pressure freezing allows for evaluation of vesicle docking. This does not detract from the value added regarding synaptic ultrastructure. However, docking or fusion state cannot be evaluated.

The new electrophysiological analysis of the co-cultures is difficult to interpret. The bath application of pharmacological inhibitors can have (at least) two effects: (1) it inhibits ion channels at the surface of the tumor cells, (2) it inhibits ion channels in the cultured neurons. The reduction of spontaneous events by all blockers (targeting glutamatergic and GABAergic postsynaptic receptors) is puzzling and raises concern. The authors need to use recording conditions that separate sIPSCs and sEPSCs. If the majority of events are glutamatergic, then one would actually expect bicucullin to increase the frequency (as it would block GABAergic transmission in the culture and increase network activity). These experiments raise concern as to the validity of the experiment.

Figure 4d (optogenetic stimulation) seems anecdotal. A single trace is shown, there is no information on replication, success rate of evoked responses or fraction of responsive cells.

Figure 5a-d: The only modest reduction of cancer cell number and EdU+ cells after TTX addition is not strongly supportive of synaptic transmission as a major component of neuron-derived growth stimulation. TTX does not block mEPSCs. Would glutamate receptor blockers have a more substantial effect in this assay? At present, these experiments rather suggest a non-synaptic neuronal contribution to tumor cell proliferation.

Version 4:

Reviewer comments:

Referee #1

(Remarks to the Author)

The authors have responded very well and convincingly to my remaining points.

Referee #2

(Remarks to the Author)

The authors have satisfactorily addressed all of my concerns and significantly improved the manuscript during the revision process.

Referee #3

(Remarks to the Author)

I thank the authors for their thoughtful responses and the revisions made in the article which have substantially improved the quality of the work.

University Hospital of Cologne | Translational Genomics
Dr. Filippo Beleggia Weyertal 115-b, 50931 Cologne

To:

Barbara Marte
Senior Editor, *Nature*

Dr. nat. med.

Filippo Beleggia
Junior Group Leader
Resident physician

University Hospital of Cologne
Department of
Translational Genomics

Department I
of Internal Medicine

Center for Integrated Oncology

Mildred Scheel
School of Oncology

Weyertal 115-b
50931, Cologne

Tel.: +49 221 478 96703

Fax: +49 221 478 97719

filippo.beleggia@uk-koeln.de

Manuscript 2022-12-20232 – plan to address point 5 raised by reviewer 2.

Dear Barbara,

Thank you again for the meeting last week. Here is an overview of our plans to answer point 5 raised by reviewer 2.

The reviewer writes:

“5. The RP mouse model used employs Ad-CMV-Cre to initiate SCLC, which may hit some NE cells, but undoubtedly hits other non-NE cell types. This is consistent with the observation that some tumors develop in peripheral locations that should lack normal NE cells (as in Ext. Data Fig. 1c). Some studies demonstrate that cell of origin impact the metastatic phenotype of SCLC (Yang et al, Cancer Discovery, 2018), while others demonstrate that cell of origin impacts tumor latency, transcriptional profile, and histopathology (Sutherland, Cancer Cell, 2011; Olsen et al, Genes Dev, 2021; Chen et al, Oncogene, 2022). The use of CMV-Cre contributes to tumor heterogeneity and potentially to pharmacological responses in Figure 6. Performing the studies in Figure 6 using CGRP-Cre to restrict tumorigenesis to NE cells would significantly control for these effects.”

In our understanding, the reviewer is worried about a potential impact of tumor heterogeneity caused by the use of the CMV promoter and suggests using the CGRP promoter instead, to restrict tumorigenesis to neuroendocrine cells.

As the reviewer points out in minor comment 2, the SCLC phenotype can arise from cells other than the neuroendocrine cells. Therefore, we believe that the heterogeneity caused by the CMV-driven model is a positive feature that increases the relevance of the generated preclinical data.

Nonetheless, we understand how investigating the possible influence of the cell of origin would improve our manuscript and we have already initiated the requested experiments.

The experiments shown in Figure 6 from our initial submission include 6 different treatment cohorts, divided into two groups:

- 1- Single agent controls (treated with either PBS or the vehicle solution for riluzole)
- 2- Riluzole
- 3- DCPG

- 4- Chemotherapy
- 5- Chemotherapy *plus* riluzole
- 6- Chemotherapy *plus* DCPG

As requested, we initiated the breedings and intratracheal instillations to repeat these treatments on tumors generated using the CGRP-promoter-driven Cre.

We will finish the intratracheal-instillations with the CGRP-Cre virus for the first three cohorts by the end of the month (April 2023). We expect these mice to develop tumors by the end of December 2023 and to reach termination criteria by the end of March 2024. Overall, we believe that these 3 cohorts would be enough to investigate whether tumors derived specifically from neuroendocrine cells are more or less amenable to neuromodulatory treatment.

Cohorts 4 through 6 include chemotherapy. We will be able to start tumorigenesis for cohorts 4-6 by the end of June 2023 and the treatments by the end of February 2024. These mice will likely reach termination criteria by the end of June 2024.

It is of course possible that the CGRP model may respond differently to the therapies, since it is more specific for the neuroendocrine cell of origin. This may change the timelines by 1-2 months in either direction.

A faster way to investigate the influence of the cell of origin could be the focus on the very early stages of tumor development, directly after tumor induction with CGRP-driven Cre. We could perform genetic manipulation of *Grm8* or treatment with DCPG. These experiments could be completed by the end of 2023 and would provide very complementary data to the denervation experiments from the Venkatesh paper.

We would therefore like to first submit a revised paper with the short-term experiments on early tumors, while at the same time working on the longer-term repetition of the treatments in Figure 6. Should the Venkatesh/Monje team need more time for their resubmission, we can provide the data related to combination treatments with chemotherapy, as well.

We are looking forward to hearing your thoughts on this strategy and would like to thank you again for your time and consideration.

Yours sincerely,

Universitätsklinikum Köln (AöR)
Translationale Genomik
Dr. Filippo Beleggia
Weyertal 115-b, 50931 Köln
Filippo Beleggia

Referee expertise:

Referee #1: cancer neuroscience

Referee #2: SCLC

Referee #3: neuroscience, synaptic transmission etc

Referees' comments:

Referee #1 (Remarks to the Author):

The manuscript of Schmitt et al adds a relevant new angle to the emerging field of cancer
neuroscience: the potential existence of neuron-cancer synapses outside the brain, and
outside gliomas. A specific strength are the broad genetic screening methodologies and
gene network analysis approaches applied here. The therapeutic outlook with 2 clinical
applicable agents interfering with glutamatergic signaling that have antitumor effects in
their SCLC mouse model is very interesting, too. The manuscript is well written.

The weakness is the lack of a true proof for functional / bona fide synapses (see below),
which is certainly critical; and also a lack of data showing which neurotransmitter receptor
class (most important) and which downstream mechanisms (of secondary importance)
exactly convey the protumor effects. The manuscript is meandering between data in
support of AMPA receptors and/or NMDA receptors and/or metabotropic glutamate
receptors. A clear line would be important, and distinct investigations of all potential
pathways in the different experimental settings. Experiments to distinguish true synaptic
from paracrine effects would be very important, too (e.g. by genetic interference with
synapse-relevant genes in SCLC cells). Further details are given below.

We thank the reviewer for the positive reception of our manuscript and for the
thoughtful review of our work. We truly appreciate the helpful and constructive input by
the reviewer, which we feel helped to improve the quality of our manuscript and
especially several aspects related to the interactions between cancer cells and neurons.
Please find below specific answers to each point raised by the reviewing expert.

First and foremost, the authors need to clearly demonstrate neuron-cancer cell synapses.
First, this needs to be done by proper electron microscopy: in Ext Data Fig. 8i-l, they show
good-quality EM images of mouse tumors, demonstrating that they are capable of doing
these kind of experiments. Here, no synapse is visible, particularly no synapse between
putative neuronal and putative cancer cell structures. The authors need to search for those
and demonstrate them. They also need to better prove / demonstrate the nature of the
presynaptic structure (clear neuronal structure) and the postsynaptic one (clear cancer cell
structure), e.g. by immunogold stainings if necessary. Second, their functionality needs to be
better demonstrated by proper electrophysiological experiments, in vitro and ideally also on
acute slices ex vivo. The electrophysiological characterizations provided here are quite
limited and need to be improved. First, why not inhibiting with CNQX OR AP5 to understand

whether AMPAR OR NMDARs are involved here? Second, co-culture of SCLC cells with
cortical neurons might create artificial glutamatergic synaptic contacts. It would therefore
be particularly important to recapitulate these findings with PNS neurons that are found in
the primary tumor microenvironment in SCLC. Third, it needs to be shown that inhibition of
neuronal activity (eg by TTX) is also completely inhibiting the sPSCs. Fourth, a stimulation
paradigm with EPSC analysis for the most important conditions would be important, too.
Last but not least, investigations on tissue and not simple in vitro co-cultures would greatly
increase the validity of the findings for the disease. The authors perform this for acute brain
slices, but only with a very limited approach, and with a somewhat artificial model (direct
implantation of cancer cells in the brain, no true brain metastasis model). They should
perform this in their primary tumors, too.

1) We thank the reviewer for these important comments. In the revised manuscript (**Figure**
**3**), we have better characterized the structural and ultrastructural features of neuron-
cancer cell contacts, showing that they exhibit typical hallmarks of *bona fide* synapses.

In co-cultures of human SCLC cells and neurons, we visualized the synaptic contacts using
three types of super-resolution microscopy, with increasing resolution: STED microscopy
(40-50 nm), X10 expansion microscopy (20-25 nm resolution), and ONE microscopy (1-5
68 nm resolution). These tools enabled us to demonstrate the co-localization of pre- and
69 post-synaptic markers (Vglut1 and Homer1) on neurons and cancer cells, respectively.
Within these contacts, the distance between the pre-synaptic Vglut1 and the post-
synaptic Homer1 fell in the range of that observed in *bona fide* synapses between
neurons present in these co-cultures (**Figure 3g, h**). We also identified Homer1-positive
regions at points of contacts between neuronal axonal fibers and SCLC cells in brain
allografts (**Extended Data Figure 11a**). The revised text describing these new data reads
as follows:

“To investigate the nature of these contacts, we performed stimulated emission
depletion (STED) microscopy of SCLC-neuron co-cultures. Immunostainings for
glutamatergic vesicles (anti-VGLuT1) and the postsynaptic protein HOMER1 revealed
colocalizing formations at the contacts between neurons and cancer cells (**Fig. 3a, b**). To
better inspect the structure of these contacts, we next turned to 10-fold expansion
microscopy, which reaches ~25 nm resolution ¹. The 3D reconstruction of these samples
disclosed a spatial organization consistent with synaptic structures, with VGLuT1-positive
puncta outside of the cancer cells juxtaposed to HOMER1 immunoreactivity within the
cancer cells (**Fig. 3c**). To visualize synapses in even higher detail, with single-digit
nanometer resolution, we employed ONE microscopy ². 3D and 2D ONE images
presented the clear separation of the pre- and post-synaptic elements, and demonstrated
that their localization resembles classical synapses between CNS neurons (**Fig. 3d, e**).
Importantly, the distance between the VGLuT1- and the HOMER1-positive puncta was

comparable with that from *bona fide* synapses (neuron-to-neuron synapses) within the
same cultures (**Fig. 3f-h**).

We further confirmed these findings in SCLC brain allografts *in vivo*. Here,
immunostaining for HOMER1 showed immunoreactivity of putative postsynaptic
structures within cancer cells in close proximity to GFP-positive axonal boutons
(**Extended Data Fig. 11a**)."

As suggested by the reviewer, we also conducted additional EM experiments, *in vitro* and
*in vivo*.

*In vitro*, we performed electron tomography of co-cultures of neurons and cancer cells,
focusing on points of contacts between axonal boutons (filled with synaptic vesicles) and
cancer cell plasma membrane (**Figure 3k**). *In vivo*, we performed CLEM of murine brain
sections harboring transplanted SCLC cells, followed by tomography of identified
reporter-positive cancer cells (**Figure 3i, j** and **Extended Data 11b**). In both settings, we
identified points of contacts showing classic hallmarks of *bona fide* synapses, including
presence of a putative synaptic cleft, a presynaptic bouton and an active zone
characterized by vesicles that were either docked or in the process of fusing with the
plasma membrane. The revised text describing these new data reads as follows:

"To further characterize these putative synaptic contacts at the ultrastructural level,
we next deployed Correlative Light Electron Microscopy (CLEM) in brain allografts and
co-cultures. Inspection of DsRed-positive SCLC cells confirmed the presence of synaptic
boutons filled with vesicles contacting the plasma membrane of cancer cells (**Extended**
**Data Fig. 11b**). Importantly, additional ultrastructural hallmarks of stereotypical synapses
were identified, including the presence of a synaptic cleft and vesicles docked to or
undergoing fusion with the presynaptic membrane (**Fig. 3i-k, Extended Data Fig. 11b**)."

To better demonstrate the functionality of the synaptic contacts between neurons and
SCLC cells, we performed additional electrophysiological characterization, as suggested
by the reviewer, in co-cultures *in vitro* and in allografted SCLC in mouse brain. Due to
technical limitations, we were not able to record from cancer cells in lungs, on account
of this tissue being not ideally suited for slice preparation and patch-clamp recordings of
potentially identified small tumor cells. Our experiments included treating co-cultures
with the individual inhibitory compounds riluzole, CNQX, AP5, bicuculline and TTX, in
addition to the combination of CNQX and AP5 that we originally included. Each of the
added treatments significantly reduced the frequency of sPSCs (**Figure 4c**), suggesting
that in co-culture with cortical neurons, SCLC cells do not exhibit a specificity for a certain
type of input. Although we are aware of the potential limitations of the brain
transplantation model to corroborate our data, a similar trend (albeit with higher
variability) was observed in SCLC cells, when recorded in acute brain slices (**Figure 4f**).
Consistent with these findings, we found that transplanted SCLC cells are innervated by
both excitatory and inhibitory neurons in the hippocampus, using transsynaptic tracing

(Figure 4j, Extended Data Figure 13a, c). Lastly, we performed SCLC cell recordings
following optogenetic stimulation of co-cultures in which cortical neurons had been
previously transduced with an AAV expressing ChR2. These experiments demonstrate the
reversible induction of synaptic currents in SCLC cells upon neuronal stimulation (Figure
4d). The revised paragraphs describing these new data reads as follows:

“While whole-cell patch-clamp recordings of COR-L88 mono-cultures
revealed the absence of any spontaneous inputs, the same cells developed
spontaneous postsynaptic currents (sPSCs) when co-cultured with cortical
neurons (Fig. 4a, b). These currents were reduced when the co-cultures were
treated with the voltage-gated sodium channel blocker TTX, with the AMPA
receptor antagonist CNQX, with the NMDA receptor antagonist AP5, with the
glutamate release inhibitor 2-amino-6-trifluoromethoxy benzothiazole (riluzole)³
and, interestingly, also with the GABA receptor inhibitor bicuculline, suggesting
that cancer cells in co-culture are able to promiscuously form contacts with both
excitatory and inhibitory neurons (Fig. 4c). Importantly, optogenetic stimulation of
co-cultured neurons expressing channelrhodopsin-2 reversibly elicited fast inward
currents in recorded SCLC cells, further corroborating the existence of direct
synaptic contacts (Fig. 4d).

Consistent with these *in vitro* data, patch-clamp recordings in acute brain
slices prepared from brain allografts revealed detectable biphasic sPSCs in a
fraction of recorded SCLC cells (Fig. 4e). Treatment with TTX, CNQX, AP5 or a
combination of CNQX, AP5 and bicuculline reduced the occurrence of sPSCs,
although the difference did not reach statistical significance (Fig. 4f).
Nevertheless, these data indicate that SCLC cells engage in synaptic transmission
in brain tissue.”

“We next conducted transsynaptic tracing experiments in brain allografts *in*
*vivo*, by stereotactically co-injecting G-TVA-expressing- or TVA-only-expressing-
SCLC cells and EnvA-pseudotyped Δ G RABV-GFP in the mouse hippocampus
(Fig. 4j). In G-TVA injected animals, the DsRed/GFP double-positive SCLC starter
cells were typically surrounded by GFP-only-positive axonal fibers (Fig. 4j,
Extended Data Fig. 13a). Consistently, GFP-positive neurons were found in the
regions (hippocampus and subiculum) adjacent to the grafted G-TVA-expressing
SCLC cells, while control experiments with cancer cells expressing exclusively

TVA resulted in absent or minor neuronal labelling (**Fig. 4k, Extended Data Fig.**
**13b**). Classification of neurons according to their morphology and layer positioning
in the traced anatomical regions near the injected area disclosed both putative
excitatory and inhibitory neurons, further indicating that SCLC cells can be
innervated by distinct neuronal sub-types *in vivo* (**Extended Data Fig. 13c**).”

In line, the images of Fig. 3d-f show no clear synaptic structures. The authors should apply
superresolution fluorescent microscopy (or EM, see above) to make their point of synaptic
contacts between CNS neurons and cancer cells. The interpretation of lack of nerve
structures in the tumor mass and a neurotoxic effects is questionable – it is very well
possible that simple expansive growth of a rapidly proliferation tumor mass is causing this
phenomenon.

2) As suggested, we have performed super-resolution imaging analysis of neuron-SCLC cells
contact sites, in co-cultures, showing that the distribution of putative pre- and post-
synaptic markers (VGluT1 and Homer1) is fully consistent with that expected for *bona*
*fine* neuron-neuron synapses found in the same co-culture (**Figure 3a-h, Extended Data**
**Figure 10e**). As detailed above, we also included EM tomography of cancer cells identified
by CLEM in brain allografts (**Figure 3i, j, Extended Data 11b**), as well as co-cultures (**Figure**
**3k**) showing clear synaptic structures with neurons. We also understand how a possible
neurotoxic effect is speculative and we changed the relative paragraph accordingly,
which now reads:

“Using confocal microscopy analysis of brain sections, we found that by 10-12
190 days after transplantation the cancer cells located in the periphery of the tumor
were profusely contacted by GFP-positive boutons and axonal bundles (**Fig. 2g-**
**h**). Moreover, most of these GFP-positive boutons were strongly immunoreactive
for the excitatory presynaptic marker VGluT1 (**Fig. 2i**).”

The growth benefit from co-culture with neurons (entire Fig. 4) is of course supportive but
not really new, and not informative for the mechanism that is in place here.

3) We thank the reviewer for this comment. As this point was raised by all three reviewers,
we have now performed a large series of experiments to further characterize the growth
advantage provided to SCLC cells by neurons. As requested by reviewer 3, we can now
show that the growth advantage is mediated – at least in part – by neuronal activity, as
it is significantly decreased in the presence of TTX. (**Figure 5c, d**). Additionally, we
investigated whether direct contacts are required for the proliferation advantage by
comparing co-cultures with cultures using media conditioned by neurons. These

experiments revealed that the growth advantage provided by true co-cultures was
substantially more pronounced in each of the 8 SCLC cell lines, although we could
measure a small but significant growth advantage in 5 out of 8 SCLC lines cultured with
conditioned media (**Figure 5f, Extended Data Figure 14a-h**). To put these observations in
perspective and following the advice provided by reviewer 2, we also included two
positive controls: high-density seeding and fibroblast co-culture. Both of these conditions
provided a growth advantage, which was less pronounced than the one provided by the
neurons (**Extended Data Figure 14a-h**). Of note, we also recapitulated the proliferation
advantage in co-cultures involving peripheral neurons derived from the nodose ganglion,
which physiologically innervate the pulmonary neuroendocrine cells which constitute the
most likely cell of origin of SCLC (**Figure 5h-k, Extended Data Figure 15**). In addition, as
requested by reviewers 2 and 3, we also analyzed whether Non-Small Cell Lung Cancer
(NSCLC) cell lines also derive a growth advantage by being co-cultured with neurons. As
shown in **Figure 5g** and **Extended Data Figure 14i-l**, 3 out of 4 NSCLC lines derived a small,
but significant, growth advantage in co-culture, which was less pronounced than the one
observed in our SCLC-neuron co-cultures. Altogether, these new data lend strong support
to the notion that the physical contacts between neurons and SCLC cells, coupled with
neuronal activity, promotes proliferation of the cancer cells. The results section
describing these new findings now reads:

“To directly investigate the functional relevance of the contacts between SCLC
cells and neurons, we tested whether SCLC cells derive a growth advantage when
kept in co-culture with neurons, as compared to mono-cultures. To this end, we
compared the proliferative capacity of DsRed-expressing human COR-L88 cells
seeded at low density and maintained either alone (mono-culture) or in co-culture
with cortical neurons, followed by EdU labeling 2h prior to analysis. While only few
scattered EdU-positive SCLC cells were found in mono-cultures, SCLC cells in co-
cultures often appeared as larger proliferating clusters (**Fig. 5a**). Importantly, this
effect was significantly reduced when the co-cultures were treated with TTX,
suggesting that the co-culture-specific proliferative advantage of SCLC cells is at
least partially mediated by neuronal activity (**Fig. 5c, d**).

To evaluate whether neuronal co-culture stimulates proliferation in all well-
defined SCLC subtypes, we monitored the growth of eight distinct cell lines with
live cell imaging: COR-L88, H1836, H69 and H146 for the SCLC-A subtype,
DMS273 and H524 for the SCLC-N subtype and H211 and H526 for the SCLC-P
subtype. All tested lines derived a significant proliferation advantage when co-
cultured with cortical neurons (**Fig. 5e, f, Extended Data Fig. 14a-h**). Importantly,
some cell lines (COR-L88, H69, DMS273, H524, H211) also derived a minor
proliferation advantage when cultured in conditioned media derived from neuronal

cultures. However, for all cell lines, the physical presence of the neurons conferred
a significantly stronger proliferation advantage (**Fig. 5f, Extended Data Fig. 14a-**
**h**). The proliferative advantage appears to be specific for neurons, as it vastly
exceeded that observed in high-density mono-cultures (**Extended Data Fig. 14a-**
**h**) and it was even stronger in four of the cell lines than that conferred by co-
cultures with fibroblasts, which have been shown to strongly promote SCLC
growth ⁴ (**Extended Data Fig. 14a-h**). Moreover, the effect appeared to be largely
specific for SCLC, as four non-small cell lung cancer (NSCLC) lines (H1975,
HCC44, HOP62, H2291) derived only a minor proliferation advantage when co-
cultured with neurons (**Fig. 5g, Extended Data Fig. 14i-l**). Finally, increased
proliferation also occurred when co-culturing SCLC cells with murine nodose
ganglia (**Fig. 5h-k, Extended Data Fig. 15**), which physiologically innervate the
PNECs in the lung and are the most likely origin of the VGluT1-positive fibers
observed in tumors *in vivo* (**Fig. 2a, d-f**) ⁵. Collectively, these data indicate that all
well-defined SCLC-subtypes derive a growth benefit when co-cultured with
neurons. This advantage is at least partially dependent on neuronal activity and
physical proximity.”

Regarding the differences between the large groups (n= relevant for each group) of
untreated, chemo-treated and checkpoint inhibitor treated mice: the authors argue they did
not detect significant differences between the groups (please clarify better by statistical
comparisons etc.) in the first place and therefore pooled these groups. However, it would be
very interesting and important for the study to understand whether the distinct neuronal
features / genes identified are different between those groups. This should be analyzed. The
background is that neuronal features of tumors can vary with treatment; in fact, there is
published data showing just that (increase of those features with treatment / resistance /
relapse).

4) We thank the reviewer for this comment and we apologize for the lack of clarity regarding
the statistical test used. We have included the missing information in the main text and
further developed the methods section. The main text now reads:

“Initial analysis using a random permutation test did not reveal any gene with a
significantly different number of insertions between untreated,
cisplatin/etoposide-exposed or anti-PD1 antibody-treated tumors or between
primary and metastatic tumors (**Extended Data Tables 1-3**). Therefore, we
pooled all samples (303 tumors from 58 mice) for the overall statistical analysis.”

The methods section now reads:

“We used a permutation test to compare the distribution of the transposon
insertions in different sub-cohorts: untreated versus chemotherapy, untreated
versus immunotherapy and lung tumors versus metastatic tumors. For each
comparison, the union of samples included in the comparison was shuffled
1000000 times, while maintaining the same number of samples from each mouse
line in each sub-cohort (RPLH and RPLS). For each gene, we then counted the
number of iterations in which the absolute difference in the fraction of samples
carrying an insertion is greater than in the real configuration. We then calculated
the FDR-corrected q values for each gene.”

No gene showed a statistically significant difference in transposon insertions across
groups. We have now increased the repetitions of the permutation test to 1 000 000 and
included the q values of the permutation test, which never reaches statistical
significance, in **Extended Data Table 1,2**.

While we initially expected the treatments to change the pattern of transposon
insertions, we now rationalize the lack of differences with the long latency of our tumor
model. We consider it likely that the bulk of the signal is generated in the 6-9 months
between the adenovirus-mediated recombination and the start of treatment as opposed
to the 1-3 months of treatment. In addition, transpositions happening at this early stage
would be present in a much larger fraction of the cells and therefore further blind our
ability to detect later events.

Moreover, the authors analyzed bulk tumor material, not single cell data that allows to
clearly differentiate between cancer cells and nonmalignant (particularly neural) cells. It
would greatly increase the value of the genomic and particularly the gene expression data if
single cell data of tumor cells vs non-tumor cells would be analyzed, too, in man and mouse.

5) We thank the reviewer for this important suggestion. Accordingly, we have performed
scRNAseq analysis of 6 murine tumors and we can now clearly show that the gene sets
we identified in bulk data are indeed present within the cancer cells (**Extended Data
Figure 7a-f**) but not within epithelial, endothelial, stromal or infiltrating immune cells. In
addition, we further analyzed available human scRNAseq data ⁶ and again see that it is
the cancer cells who express high levels of the same gene sets, including synaptic and
glutamatergic gene sets. (**Extended Data Figure 7g-i**). The text describing these additional
analyses reads as follows:

“To confirm that the expression of the SCLC-specific gene sets is driven by
the cancer cells themselves, we performed single-nuclei RNA sequencing
(snRNA-seq) on six tumors collected from RP mice and re-analyzed available
human scRNA-seq data ⁶. As expected, the gene sets enriched in cancer cells

compared to healthy cells in both datasets were dominated by cell proliferation
gene sets and neuronal gene sets, such as *Synaptic Membrane* and
*Glutamatergic Synapse*, resulting in a near identical pattern to our analysis of bulk
expression data (**Fig. 1c, Extended Data Fig. 7**)."

To get a better idea of the uniqueness of the neuronal phenotypes / synaptic functions
detected in the genomic screen, it would be important to see the results and statistical
testings in more granularity (better than shown in Fig. 1b, at least additional to Fig. 1b), not
just the raw data as shown in Ext Data Tables 3,4.

6) We thank the reviewer for this suggestion. To address this, we have replaced the original
Figure 1b with a more granular visualization of the top scoring gene sets. This new panel
clearly illustrates the striking and independent enrichment of the synaptic gene sets in
both datasets (**Figure 1b**). While the initial Figure 1b lacked a detailed description of the
enriched gene sets, it did illustrate that the majority of the gene sets are neuronal. To
preserve this information, we have moved the initial Figure 1b to the supplement
(**Extended Data Figure 5h**).

Co-culture of human SCLC lines were performed only with murine cortical neurons, not
peripheral nervous system cell types or human neurons. This needs to be extended.

7) We thank the reviewer for raising this important point. As briefly touched upon in
response to point 3, we have now performed co-culture experiments with the relevant
peripheral neuronal population that physiological innervates the PNECs. The experiments
within **Figure 5h-k** and **Extended Data Figure 15** demonstrated that human SCLC cells
derived a significant advantage when cultured with nodose ganglia. In addition, we show
that iPSC-derived human cortical neurons engage in putative synaptic contacts with
human SCLC cells, as evidenced by the colocalization of the presynaptic marker Bassoon
and the postsynaptic marker Homer1 (**Extended Data Figure 10d**). The text describing
these experiments reads as follows:

"Finally, increased proliferation also occurred when co-culturing SCLC cells
with murine nodose ganglia (**Fig. 5h-k, Extended Data Fig. 15**), which
physiologically innervate the PNECs in the lung and are the most likely origin of
the VGLUT1-positive fibers observed in tumors *in vivo* (**Fig. 2a, d-f**)⁵."

"To further investigate the ability of cancer cells to form contacts with neurons,
we established co-culture experiments with murine cortical neurons. Human COR-
L88 cells, of the SCLC-A subtype, expressed both AMPA and NMDA glutamate
receptors (**Extended Data Fig. 10a, b**) and were profusely contacted by VGLUT1-
positive neuronal processes (**Extended Data Fig. 10c**). Importantly, similar
contacts were found in co-cultures with human iPSC-derived cortical neurons, and

365 were characterized by the expression of bassoon (BSN) in neuronal fibers and
366 HOMER1 in cancer cells (**Extended Data Fig. 10d**)”

The retrograde monosynaptic tracing experiments are a nice way to study the neuron-
cancer cell connections/networks, although not finally proving the existence of bona fide
synapses between the cell types. However, I am missing an important experimental
validations: please provide dynamic (time-lapse) data of cancer cells infected first, and
neurons following. Finally, the number of independent experiments, ROIs analyzed, with
number of cells etc. is not really clear here. This also applies to many other figures from Fig.
2 on and should be corrected. All figure legends should contain exact information about the
n=.

8) We thank the reviewer for raising this important point, which has similarly been raised
by reviewer 3. To further substantiate our rabies tracing experiments, we have now
performed additional control experiments. First, as suggested, we performed a time-
lapse experiment in co-cultures with cortical neurons, using SCLC cancer cells that had
been previously (in monoculture) transduced with the G-TVA retrovirus followed by
RABV-GFP. 24h after RABV-GFP transduction of SCLC cells, we washed the culture, plated
these cancer cells onto neurons and placed the resulting co-culture under the time-lapse
microscope. At this time, only sparse DsRed/GFP double-positive cancer cells were
visible, demonstrating the presence of starter cells. Over the next 48hrs, these sparse
cells had replicated forming small clusters. Only starting from 48hrs did the first GFP+
neurons appear in the surrounding of initially infected starter cancer cells (**Extended Data
Figure 12d**), this time point being consistent with the time needed for retrograde
transmission of RABV from the cancer cells to neurons, and then to start expressing GFP
in neurons themselves. To control for spurious RABV-GFP labeling potentially mediated
by any non-synaptic transfer mechanisms in our *in vitro* and *in vivo* experiments, we also
included a new control condition using a retrovirus encoding exclusively for DsRed and
TVA (but not the glycoprotein G, which is required for RABV assembly of new viral
particles and thus transsynaptic spread). Using this control retrovirus, we quantified the
extent of non-specific RABV labeling in our settings (which is minor), thus validating our
data obtained using the experimental G-TVA retrovirus (**Figure 4k, h**). The revised text
describing these new data reads as follows:

“We further validated the expected transsynaptic RABV-GFP spread using time-
lapse experiments of these co-cultures, showing that neurons acquire GFP
fluorescence only starting from 48hrs after the appearance of DsRed/GFP double-
positive SCLC starter cells (**Extended Data Fig. 12d**). Assessment of GFP-
positive neurons in co-cultures with SCLC cells lacking any prior retroviral
transduction, or expressing only TVA and/or DsRed (but not G) as negative
controls revealed a low and quantifiable, spurious labelling by RABV in our
settings (**Fig. 4h**). In contrast, co-expression of G in SCLC cells resulted in a net

increase of neuronal labelling of 10-fold or more, corroborating the reliability of this
transsynaptic approach (**Fig. 4h**).”

We have also better specified the requested information about quantification methods,
the number of samples used, as well as statistical tests in the methods section and figure
legends.

The lack of specificity of the CGRP staining makes it difficult to draw any meaningful
conclusions from e.g. Ext Data Fig. 7. Other stainings for PNS neuronal types that are well
established should be employed that allow a better identification of PNS neuron classes, and
to separate them from cancer cells and healthy PNECs. This is actually tried in Fig. 3a-c and
Ext Data Fig. 8. However, all this data lacks proper quantifications, and also staining controls
– I am not sure what I really see on many of those images.

9) We thank the reviewer for this comment and acknowledge the difficulty in the
interpretation of the images in the original Figure 3, Extended Data Figure 7 and Extended
Data Figure 8. As a result, we have removed Extended Figure 7 from the manuscript and
improved Figure 3 (now **Figure 2**) and **Extended Data Figure 8** as well as the text as
detailed below.

The difficulty in the anatomical examination of the innervation of PNECs and SCLC tumors
stems from the shared positivity of the cancer cells for many neuronal markers, such as
gene-product 9.5 (PGP9.5), synaptophysin (SYP), calcitonin gene-related peptide (CGRP).
Nonetheless these markers are necessary to visualize healthy and transformed
neuroendocrine cells^{5,7-9}. More specific markers of neuronal subpopulations only label
positive nerve fibers, such as VGLUT1 and P2X3, which mark different types of vagal
afferents^{5,9}. The varicose CGRP-positive fibers that we highlighted are never observed to
penetrate into the PNEC clusters (neuroepithelial bodies, NEBs). As suggested by the
reviewer, we now employed a second, more specific marker for these fibers (Substance
P, SP) to clearly illustrate this point (**Extended Data Figure 8i, j**). We also included an
additional example of a VGLUT1-negative tumor as negative control (**Extended Data**
**Figure 8k**). Finally, we performed stainings in an additional mouse model which carries a
*Rosa26^{LSL-Cas9-IRES-EGFP}* allele and allowed us to better distinguish the arborizations of the
nerve fibers between the transformed and non-transformed neuroendocrine cells at
different stages of cancer development. This innervation pattern closely resembles what
we observed in healthy PNECs, which we consider positive controls (**Extended Data**
**Figure 8a-c**). The revised text describing these findings reads as follows:

“Interestingly, VGluT1-, P2X3 and GAP43-positive nerve fibers were detectable in
both a subset of healthy PNECs, clustered into neuroepithelial bodies (NEBs) (**Fig.**
**2a; Extended Data Fig. 8a-c**), and in small SCLC tumors (**Fig. 2a-c; Extended**
**Data Fig. 8d**). In contrast, larger tumors were mostly devoid of intra-lesional nerve
fibers and, when present, GAP43-positive fibers were observed at the tumor
border (**Fig. 2c; Extended Data Fig. 8h**). CGRP-positive, SP-positive and SYP-

positive fibers were also profusely present near, but not within, the tumors (**Fig.**
**2a-b; Extended Data Fig. 8d-g, i-k**). Using mice that additionally carry a GFP-
marked allele (*Rb1^{fl/fl};Trp53^{fl/fl};Rosa26^{Cas9-GFP}*, RPC), we detected VGlut1-positive
fibers arborizing between the neuroendocrine cells starting from the very initial
stages of transformation up to small and medium size tumors (**Fig. 2d-f**).”

The entire brain metastasis data is not very strongly developed and not very well connected
to the primary tumor data.

10) We thank the reviewer for this comment, we agree and are fully aware that the brain
transplantations do not fully recapitulate the complexity of the metastatic process. While
an investigation of metastatic spread was not an aim of our study, we believe that the
allograft experiments add to our manuscript by showing that synaptic innervation of
cancer cells can be established *in vivo*. As detailed above, we have now further developed
this part by including (i) electron tomograms of synaptic contacts identified via CLEM, (ii)
further immunostainings of synaptic markers, (iii) electrophysiological characterization
and (iv) *in vivo* monosynaptic tracing experiments. Finally, we reorganized the figures to
reflect the main findings of our paper rather than the model used. **Figure 2** now shows
the identification of contacts between SCLC cells and neurons, **Figure 3** now shows
structural/ultrastructural information consistent with the establishment of *bona fide*
synapses and **Figure 4** now shows that the functionality of SCLC-neuron synaptic
contacts. Therefore, we believe that the allograft data is now much better connected to
the rest of our manuscript.

**Minor points:**

Ext Data Fig. 1c: comparison with human histology would be informative, and also arrows
depicting what we see and why this is typical for SCLC.

11) We thank the reviewer for this suggestion. We have now improved the IHC panels in
**Extended Data Figure 1** and mention of how the positivity for neuroendocrine markers
(NCAM1 and SYP) and the scant cytoplasm with salt-and-pepper chromatin closely
resemble the histology of human SCLC. The text describing the comparison reads as
follows:

“Of note, the animals included in these experiments developed tumors that
morphologically resembled human SCLC, composed of small cells with scant
cytoplasm, which express neuroendocrine markers, as previously described for
the RP model (**Extended Data Fig. 1c, d**)^{8,10}.”

When I understand the work correctly, the two 2019 Nature publications reporting bona-
fide synapses between neurons and postsynaptic cancer cells (glioblastoma, K27M mut
midline glioma) show that brain tumors can receive synaptic input, mainly to so-called
“tumor microtubes” and tumor cell networks that are driven by neuronal features (Osswald
et al Nature 2015). This picture has been further developed now by also demonstrating
neuron-cancer synapses on invasive (non-network-connected) tumor cells in glioblastoma
that reveal distinct neuronal features on multiple levels (Venkataramani et al Cell 2022).
Together these four publications make a strong case that tumor cells themselves can exert
distinct and even multiple neuronal features, and that those features appear to be a strong
determinant of the formation neuron-cancer synapses. The authors should discuss this in
more depth and relate this to their scientific question and findings. They should also include
a short discussion of the Hanahan et al Nature 2019 publication and make clear that those
are not bona-fide synapses but rather cancer cells in perisynaptic positions being stimulated
by glutamate from the synaptic cleft. They should consider to restructure the introduction
and avoid a lengthy introduction into the genetics and subclasses of SCLC without providing
clear guidance to the reader why this is of importance for the work. The authors might want
to consider to focus more on neuron-cancer synapses in this respect. The introduction
needs to be shortened anyway for this journal’s format.

12) We thank the reviewer for these interesting suggestions. We have included the points
that neuronal features within the cancer cells are a likely determinant of synapse
formation. This interpretation fits SCLC and our data very well. In addition, we have
restructured the introduction paragraphs to focus less on the genetics and subtypes of
SCLC, and rephrased it to make it clear how the subtypes and putative cell of origin may
influence the interactions with neurons. We also now mention how the contacts
described by Hanahan *et al.* are perisynaptic in nature. The relevant section now reads:

“Comprehensive genomic profiling has revealed that SCLC displays an almost
universal bi-allelic loss of the tumor suppressors *TP53* and *RB1*^{11,12}. Accordingly,
SCLC develops in mice after bi-allelic deletion of *Trp53* and *Rb1* in lung epithelial
cells⁸. Several studies have shown that pulmonary neuroendocrine cells (PNECs)
are a permissive cell type of origin for SCLC, but other cell types can also give rise
to SCLC upon loss of *Trp53* and *Rb1*, especially when oncogenic *Myc* is
concomitantly overexpressed^{7,13–15}. Interestingly, these non-neuroendocrine lung
epithelial cells acquire a PNEC-like phenotype and express neuroendocrine
markers during tumor development¹³. Similarly, human *EGFR*-driven
adenocarcinoma can transdifferentiate into SCLC and start expressing
neuroendocrine markers upon loss of *RB1*^{16,17}. PNECs develop from lung
epithelial progenitors of endodermal lineage and are innervated by different types
of nerve fibers originating from the nodose, jugular and dorsal root ganglia^{5,18–20}.
The strong association between SCLC and the neuroendocrine phenotype

suggests that SCLC-neuron interactions may be a key element in the formation
and progression of SCLC and may provide clues for the development of novel
therapeutic strategies.

Currently, three well defined molecular subtypes of SCLC are widely accepted
based on the expression levels of ASCL1 (subtype SCLC-A), NEUROD1 (subtype
SCLC-N), and POU2F3 (subtype SCLC-P). A fourth, more disputed subtype is
variably described as inflamed (SCLC-I) or YAP1-expressing (SCLC-Y) ²¹⁻²³.
Among these subtypes, SCLC-A and SCLC-N show high expression of
neuroendocrine features, while SCLC-P and SCLC-Y show lower neuroendocrine
differentiation, suggesting that they may differ in their interactions with the nervous
system ²¹.

In recent years, evidence suggesting that innervation impacts tumor initiation
and plasticity has accumulated ²⁴⁻²⁷. The most compelling evidence for neuron-
tumor cross talk originates from central nervous system (CNS) tumors and
metastases. For instance, neurotransmitter and growth factor secretion into the
microenvironment of gliomas upon neuronal activity has been shown to promote
tumor cell proliferation ²⁸. Similarly, glutamate spillover from the synaptic cleft of
neuron-to-neuron synapses has been reported to stimulate breast cancer cells
located in perisynaptic position ²⁹. In addition, direct synaptic contacts between
presynaptic neurons and postsynaptic cancer cells have been reported in gliomas
555 ^{28,30}. These synapses mediate membrane depolarizations of the postsynaptic
glioma cells through AMPA glutamate receptors, resulting in increased cell division
and enhanced tumor invasion ^{24,30,31}. Importantly, synaptic contacts in brain
tumors are located to cancer microtubes and tumor cell networks ^{30,32}, as well as
non-connected infiltrating cells that display pronounced microanatomical neuronal
features ³³, suggesting that the expression of a neuron-like phenotype may be a
determinant for synapse formation. In contrast, no *bona fide* synapses have been
described between neurons and cancers that arise outside of the nervous system.
The reported consequences of cancer-neuron interactions outside of the CNS are
dependent on tumor type. For example, parasympathetic cholinergic peripheral
nerves appear to promote metastatic spread of prostate cancer cells, whereas
breast cancer metastasis appears to be repressed by these fibers ^{24,34,35}."

Referee #2 (Remarks to the Author):

The study by Schmitt et al uses an innovative genome-wide piggyBac transposase system in
the context of the Rb1/p53 (RP) model of small cell lung cancer (SCLC) to identify genes
involved in synaptogenesis as potential regulators of SCLC. Importantly, the analysis is
rigorous using two distinct transposon models, analyzing 303 tumors from 58 mice under
various treatment conditions including metastases. The screen is clearly successful, evident
by the identification of key genes known to be functionally relevant in SCLC biology,
including Trp73, Pten, and Crebbp (Cui et al, Mol Can Res, 2014; Jia et al, Cancer Discovery
2018; George et al, Nature, 2015). The new information provided by this study is that
synaptogenesis genes such as Nlgn1, Nrnx1, Reln, and Dcc exhibit recurrent transposon
integrations suggesting a potential functional role in SCLC—although these genes are never
functionally interrogated. However, based on these findings, the authors determine that
SCLC cells form functional synaptic inputs using a co-culture system of murine cortical
neurons and human SCLC cell lines. They demonstrate that the presence of neurons
promotes SCLC growth with multiple cell lines. Interestingly, they suggest that in the RP
mouse model, small SCLC tumors have evidence of nerve fibers connecting with tumors, but
large tumors exclude nerve fibers. The absence of nerve fibers in large tumors was
confirmed in an independent allograft experiment as well. This may suggest that innervation
of SCLC is particularly relevant to tumor initiation, but not necessarily progression. The
authors also find that a large class of glutamate receptors have transposon insertions, and
they focus on GRM8, an inhibitory metabotropic glutamate receptor, but do not genetically
perturb GRM8 to test its function. They conclude showing that the pharmacological inhibitor
riluzole has a modest impact on tumor growth in the RP model alone and in combination
with standard-of-care chemotherapy.

We thank the reviewer for the positive evaluation of our work and for the insightful
suggestions below. These comments helped us to improve several aspects of our
manuscript, in particular our genetic, co-culture and preclinical experiments.

The authors employ innovative approaches toward an interesting topic not well understood
in SCLC, and some of the experiments are rigorous. However, a number of conclusions are
premature without further studies, and overall lack functional interrogation or mechanistic
insights into the phenomenon described:

Major Points:

1. The phrase in the Results section following Figure 1, “Therefore, our genetic screen....
suggests a novel role of synaptic genes in SCLC” is over-stated at this point in the text. It is
possible that expressed genes may have more integrations than non-expressed genes,
without a functional role for these genes, which should be demonstrated. Genetic
perturbation of Nlgn1, Nrnx1, Reln, or Dcc and its consequences on SCLC growth or
phenotype (in cell lines or a relevant tumor initiation setting such as the pre-SC system; Kim
et al, Genes Dev, 2016) would support these conclusions. Otherwise, it seems possible that

mutations in these genes are simply passenger effects in genes that are highly expressed
due to cell fate, but not necessarily functionally relevant.

1) We thank the reviewer for this important point. We acknowledge that the conclusion of
the first paragraphs of the results may have been premature. We have adjusted it to now
read:

“Therefore, our genetic screen identified recurrent transposon integrations
into synaptic genes, as well as several genes that recapitulate much of the known
biology of SCLC.”

We would like to thank the reviewer for suggesting a link between the levels of gene
expression and transposon integrations, which we have now investigated in our dataset.
Indeed, the background level of transposition in our screen does correlate with
expression (**Extended Data Figure 5a**). However, we also observed the opposite pattern
for human somatic mutations, with the background mutation levels showing a clear anti-
correlation to the expression levels (**Extended Data Figure 5a**). Therefore, we believe that
the two analyses ideally complement and validate each other, altogether indicating that
genes identified in both datasets are likely true positives. In support of this notion, the
genes that are significantly mutated in human samples but not identified in our screen
show a clear enrichment in nucleotides characterized by a low evolutionary conservation,
suggesting that they are not functionally relevant. In contrast, genes identified in both
cohorts show a depletion of mutated nucleotides with low evolutionary conservation,
suggesting that they are functionally relevant (**Extended Data Figure 5b**).

In addition, as requested, we validated the tumor-suppressing function of *Reln*, a gene
we identified in all our datasets. For this purpose, we employed *Rb1^{fl/fl};Trp53^{fl/fl};Rbl2^{fl/fl};Rosa26^{tdTomato};H11^{LSL-Cas9}* mice, which have been shown to produce SCLC-A tumors that
express functional *Cas9*³⁶. We induced tumorigenesis using lentiviral vectors that carry
*Cre*, as well as a non-targeting sgRNA or one of two distinct sgRNAs against *Reln*. As shown
in **Extended Data Figure 5c-g**, both sgRNAs resulted in the development of significantly
larger tumors, suggesting that loss of *Reln* causes faster tumorigenesis. The relevant
sections describing these new findings reads as follows:

“Importantly, the rate of *piggyBac* transposon insertions and the rate of
mutations in human samples show an opposite correlation to gene expression,
suggesting that these two datasets ideally complement each other (**Extended**
**Data Fig. 5a**). Consistent with this notion, mutations in genes that are significantly
mutated in human samples but not identified in the *piggyBac* screen are enriched
for non-conserved nucleotides, while mutations that are identified in both datasets
are depleted of non-conserved nucleotides, suggesting a functional role for genes
identified in both datasets (**Extended Data Fig. 5b**). We further confirmed the
effectiveness of this cross-validation strategy, using the autochthonous SCLC

mouse model *Rb1^{fl/fl};Trp53^{fl/fl};Rb12^{fl/fl};R26^{LSL-tdTomato};H11^{LSL-Cas9}* (RPR2TC) ³⁶,
combined with lentiviral delivery of Cre and sgRNAs against *Reln*, a gene
identified in both datasets. Strikingly, two different sgRNAs targeting *Reln*
produced significantly larger tumors compared to non-targeting controls, thus
functionally validating our cross-species verification approach (**Extended Data**
**Fig. 5c-g).**”

2. As previously shown by multiple groups, neuroendocrine and neuronal genes are more
commonly associated with the ASCL1 and NEUROD1 subtypes of SCLC (Rudin et al, Nat Rev
Cancer, 2019). It is not mentioned what subtypes of cell lines are used for this study such as
COR-L88, DMS273, H146, H69, H524, etc. It would be valuable to know whether functional
synapses are formed with all subtypes, or just neuroendocrine/neuronal subtypes of SCLC?
Does co-culture with neurons promote growth of all SCLC subtypes, or just SCLC-A and SCLC-
N? This question speaks to the applicability of these findings.

2) We thank this reviewing expert for this insightful comment. To directly address this
question, we have substantially expanded our cell line panel, which now includes cell
lines from the SCLC-A, SCLC-N and SCLC-P subtypes. The panel now consists of 8 cell lines:
CORL88, H69, H146 and H1836 for the ASCL1 subtype
DMS273 and H524 for the NEUROD1 subtype
H211 and H526 for the POU2F3 subtype.

As shown in the **Figure 5f** and **Extended Data Figure 14a-h**, all three subtypes derive a
significant growth advantage when co-cultured with neuronal cells.

The revised text now reads as follows:

“To evaluate whether neuronal co-culture stimulates proliferation in all well-
defined SCLC subtypes, we monitored the growth of eight distinct cell lines with
live cell imaging: COR-L88, H1836, H69 and H146 for the SCLC-A subtype,
DMS273 and H524 for the SCLC-N subtype and H211 and H526 for the SCLC-P
subtype. All tested lines derived a significant proliferation advantage when co-
cultured with cortical neurons (**Fig. 5e, f, Extended Data Fig. 14a-h).**”

3. The authors show in multiple cell lines that the presence of neurons promotes
proliferation of SCLC cell lines, but the mechanisms are still entirely unknown. Is physical
connection with neurons required for the pro-growth effect, or is supernatant from neurons
or neuron+SCLC co-cultures sufficient to promote SCLC growth? This clue toward
mechanism may be important for interpreting pharmacological in vivo experiments where
effects on small, innervated tumors may differ from large, nerve-excluded tumors. Likewise,
is the pro-growth effect specific to neurons? Have the authors studied the growth rate of
SCLC cells in fibroblast co-culture, upon blockade of neuronal contacts, and/or as a function

of cell:cell density? Is the pro-growth effect specific to neuroendocrine-high SCLC, or do
neurons also promote the growth of lung adenocarcinoma or non-neuroendocrine SCLC?

3) We truly appreciate this valuable comment. We have performed a series of experiments
to address the questions raised.

First, comparing the growth of 8 distinct SCLC cell lines covering the SCLC ASCL1,
NEUROD1 and POU2F3 subtypes, we demonstrate that growth in co-culture is
significantly enhanced compared to growth in conditioned media derived from cultures
of cortical neurons (**Figure 5f, Extended Data Figure 14a-h**). Of note, while conditioned
media is significantly less efficient in promoting SCLC cell growth than co-cultures for all
cell lines, it still provided a small, but significant benefit compared to unconditioned
media in 5 out of 8 cell lines (**Figure 5f, Revised Extended Data Figure 14a-h**). These data
indicate that physical contact between neurons and SCLC cells appears to be required for
maximum growth support, while neuron-derived factors secreted into the media are also
capable of providing a growth benefit.

Second, we now included fibroblast co-cultures as a positive control, as it has previously
been shown that adult lung fibroblasts robustly promote SCLC cell proliferation *in vitro*⁴.
As expected, the fibroblasts did provide a significant growth advantage to all 8 SCLC cell
lines tested. However, neurons outperformed fibroblasts in 4 out of 8 cell line, whereas
in 2 out of 8 cell lines we detected a small but significant growth advantage provided by
fibroblasts compared to neurons. In the remaining 2 out of 8 cell lines no significant
difference could be detected between the growth effect provided to SCLC cells by
fibroblasts or neurons (**Extended Data Figure 14a-h**).

Third, we tested the effect of high-density cultures by performing co-culture experiments
of mNeonGreen- or tdTomato- labelled SCLC cells with non-fluorescent cells of the same
cell line. These experiments revealed that in 7 out of 8 tested cell lines, high-density
culture provided a growth benefit that was significantly less pronounced than that
provided by neuronal co-cultures. In only 1 out of 8 cell lines, high-density seeding
provided a growth advantage to SCLC cells that was larger than that provided by neuronal
co-cultures (**Extended Data Figure 14a-h**).

Fourth, as mentioned in our response to reviewer comment 2, all three SCLC subtypes
tested derive a significant growth advantage when co-cultured with neuronal cells
(**Extended Data Figure 14**).

Lastly, as suggested by the reviewer, we also assessed the potential growth advantage
provided by neurons to four distinct lung adenocarcinoma (LUAD) cell lines (H1975,
HOP62, HCC44, H2291). As shown in **Figure 5g**, neuron co-culture provided a small but
significant proliferation advantage to 3 out of 4 LUAD cell lines. However, this effect was
less pronounced than that observed in SCLC-neuron co-cultures.

In addition, as requested by reviewer 3, we show that treating SCLC-neuron co-cultures
with the voltage-gated sodium channel blocker TTX, which represses neuronal activity, is
sufficient to significantly impair the growth advantage of cancer cells (**Figure 5c, d**). The
revised text sections read as follows:

“To directly investigate the functional relevance of the contacts between SCLC
cells and neurons, we tested whether SCLC cells derive a growth advantage when
kept in co-culture with neurons, as compared to mono-cultures. To this end, we

compared the proliferative capacity of DsRed-expressing human COR-L88 cells
seeded at low density and maintained either alone (mono-culture) or in co-culture
with cortical neurons, followed by EdU labeling 2h prior to analysis. While only few
scattered EdU-positive SCLC cells were found in mono-cultures, SCLC cells in co-
cultures often appeared as larger proliferating clusters (**Fig. 5a**). Importantly, this
effect was significantly reduced when the co-cultures were treated with TTX,
suggesting that the co-culture-specific proliferative advantage of SCLC cells is at
least partially mediated by neuronal activity (**Fig. 5c, d**).

To evaluate whether neuronal co-culture stimulates proliferation in all well-
defined SCLC subtypes, we monitored the growth of eight distinct cell lines with
live cell imaging: COR-L88, H1836, H69 and H146 for the SCLC-A subtype,
DMS273 and H524 for the SCLC-N subtype and H211 and H526 for the SCLC-P
subtype. All tested lines derived a significant proliferation advantage when co-
cultured with cortical neurons (**Fig. 5e, f, Extended Data Fig. 14a-h**). Importantly,
some cell lines (COR-L88, H69, DMS273, H524, H211) also derived a minor
proliferation advantage when cultured in conditioned media derived from neuronal
cultures. However, for all cell lines, the physical presence of the neurons conferred
a significantly stronger proliferation advantage (**Fig. 5f, Extended Data Fig. 14a-**
**h**). The proliferative advantage appears to be specific for neurons, as it vastly
exceeded that observed in high-density mono-cultures (**Extended Data Fig. 14a-**
**h**) and it was even stronger in four of the cell lines than that conferred by co-
cultures with fibroblasts, which have been shown to strongly promote SCLC
growth ⁴ (**Extended Data Fig. 14a-h**). Moreover, the effect appeared to be largely
specific for SCLC, as four non-small cell lung cancer (NSCLC) lines (H1975,
HCC44, HOP62, H2291) derived only a minor proliferation advantage when co-
cultured with neurons (**Fig. 5g, Extended Data Fig. 14i-l**).”

4. Related to Figure 6, pharmacological studies are used in an attempt to modulate GRM8
function, specifically with DCPG and riluzole. While DCPG is a selective GRM8 agonist,
riluzole appears to be quite non-specific and has many consequences, and its function in this
context is unknown. This is important because the in vivo studies suggest that riluzole slows
tumor growth and extends survival more than DCPG, especially when used in combination
with chemotherapy where DCPG doesn't have a significant impact on tumor growth (Fig.
6c). It would be much more compelling to perform genetic knockout or overexpression

studies with GRM8, and to test the impact of riluzole on SCLC tumor cells in the presence
and absence of GRM8. It would also be informative to determine the impact of the GRM8
agonist on synaptic connections and growth rate in the neuron/SCLC co-culture model.

4) We thank the reviewer for this comment. To select a cell line model for *in vitro*
experiments, we first assessed the expression level of GRM8 in our panel of SCLC cell
lines. As shown in **Reviewer Figure 1** below, GRM8 expression was absent or very faint in
all our cell lines. While this would be perfectly in line with a potential role of GRM8 as a
tumor suppressor, it also indicates that further experimental repression of GRM8 would
likely be futile. Nevertheless, we did attempt to validate the effect of DCPG in the CGRP-
driven model (driven by the request in reviewer comment #5). As shown in **Extended**
**Data Figure 19**, these experiments did not reveal a significant survival effect of single-
agent DCPG compared to vehicle control, potentially as a result of the low and
heterogeneous expression of *Grm8* in the cancer cells, which we have now measured in
the new murine snRNAseq dataset requested by reviewer 1 (**Extended Data Figure 16h**).
Given this experimental dilemma and the time constraints imposed by the accompanying
manuscript, we felt that further pursuit of GRM8 and DCPG was unlikely to be a
constructive avenue. We hence de-emphasized the GRM8 / DCPG angle of the
manuscript, by merging the original figure 5 and the original Extended Data Figure 12 into
a more compact supplementary figure (**Extended Data Figure 16**), particularly as we have
now further substantiated the effect of Riluzole in the CGRP-Cre model. These new data
provide *in vivo* proof of concept which warrants further clinical exploration.

The revised text sections read as follows:

“Among the possible molecular targets within this system are the glutamate
receptors, which we also identified as individual genes targeted by transposon
insertions in our *piggyBac* screen (*Grid1*, *Grik2*, *Grin3a*, *Grm1*, *Grm3*, *Grm5*,
*Grm8*), in human mutation data (*GRIA1*, *GRIA2*, *GRIA3*, *GRIA4*, *GRID2*, *GRIK2*,
*GRIK3*, *GRIK4*, *GRIN2A*, *GRIN2B*, *GRIN3A*, *GRM1*, *GRM3*, *GRM5*, *GRM8*) and
at the expression level in human samples (*GRIA2*, *GRIN3A*, *GRIK3*, *GRIK5*,
*GRM2*, *GRM4*, *GRM8*). Prominent among them was *GRM8*, a gene encoding an
inhibitory metabotropic glutamate receptor that has been shown to counteract
glutamate signaling by negatively regulating cAMP-dependent sensitization of
inositol 1,4,5-trisphosphate receptors, thereby limiting glutamate-induced
calcium release from the ER³⁷. In our human datasets, *GRM8* shows specific
expression in SCLC and a few other tumor types (**Extended Data Fig. 16c**) and
a statistically significant enrichment of both non-synonymous mutations and
more severe loss of function mutations (stop, frameshift, start-loss, and canonical
splice-site mutations) (**Extended Data Fig. 16d**). Re-analysis of scRNA-seq data
from Chan *et al.*⁶ confirmed that *GRM8* is specifically expressed in SCLC cells
(**Extended Data Fig. 16f, g**). We also find specific expression of *Grm8* in our

murine SCLC snRNA dataset, although at a substantially reduced fraction,
compared to the human dataset (**Extended Data Fig. 16h**)."

5. The RP mouse model used employs Ad-CMV-Cre to initiate SCLC, which may hit some NE
cells, but undoubtedly hits other non-NE cell types. This is consistent with the observation
that some tumors develop in peripheral locations that should lack normal NE cells (as in Ext.
Data Fig. 1c). Some studies demonstrate that cell of origin impact the metastatic phenotype
of SCLC (Yang et al, Cancer Discovery, 2018), while others demonstrate that cell of origin
impacts tumor latency, transcriptional profile, and histopathology (Sutherland, Cancer Cell,
2011; Olsen et al, Genes Dev, 2021; Chen et al, Oncogene, 2022). The use of CMV-Cre
contributes to tumor heterogeneity and potentially to pharmacological responses in Figure
6. Performing the studies in Figure 6 using CGRP-Cre to restrict tumorigenesis to NE cells
would significantly control for these effects.

5) We thank the reviewer for raising this important point. We fully agree that reassessment
of our pharmacological response data in the CGRP-Cre driven model would add to the
quality of our manuscript and have hence treated a cohort of mice harboring CGRP-Cre-
driven tumors with riluzole or DCPG and vehicle controls. While DCPG treatment did not
produce a significant survival benefit compared to controls, riluzole produced a highly
significant survival benefit, similar to that observed in the CMV-Cre-driven SCLC model
(**Extended Data Figure 19**). The revised text reads as follows:

"To further confirm these findings, we tested the efficacy of riluzole and DCPG
in a second cohort of mice using a CGRP-driven Cre, which has been shown to
more selectively induce transformation in PNECs ^{7,13-15}. In this cohort,
administration of DCPG did not significantly improve response or survival, while
treatment with riluzole resulted in a significantly higher response and provided a
significant survival advantage, compared to treatment with vehicle controls
(**Extended Data Fig. 20**)."

6. As the survival impact of riluzole in Figure 6 is relatively modest, it would be helpful to see
representative images of MRI data, and to analyze the average tumor burden per animal
rather than just the combination of individual tumors in Fig. 6a/c. It would be helpful to
convey the amount of tumor heterogeneity per animal, as differing cells of origin (as in Point
#5) may be one source of heterogeneity.

6) We thank the reviewer for this comment, which prompted us to improve the analysis of
our MRI data. As requested, we have now included representative MRI images for the
different treatments. In addition, in order to investigate total tumor burden and
heterogeneity per mouse, we have comprehensively reanalyzed our MRI data, including
9 control mice that had not yet been analyzed as well as tumors smaller than 1 cubic mm,
which were not included in the initial submission. We also followed tumor progression

beyond the time of best response. As a result, we now evaluate 2339 data points. While
this improved dataset did not change the results of the best response calculated per
tumor (**Revised Figure 6a, c**), it allowed us to investigate tumor response in much greater
detail. First, we now provide an overview of tumor burden per mouse over time, which
shows a significantly reduced tumor growth in the DCPG and riluzole cohorts, compared
to controls, as well as the chemotherapy plus riluzole, compared to the chemotherapy
alone (**Extended Data Figure 18a, Extended Data Figure 19a**). Second, we investigated
the time required for the mice to reach a total tumor burden 5 times greater than the
initial burden. This time is significantly longer for riluzole and DCPG compared to the
vehicle controls and for riluzole plus chemotherapy compared to chemotherapy alone
(**Extended Data Figure 18b, Extended Data Figure 19b**). Third, we analyze the best
response as well as the response at 28 and 56 days in terms of total tumor burden per
mouse. These data now better show that the response caused by single agents is
significant but short-lived, while the improved response caused by riluzole in
combination with chemotherapy is prolonged (**Extended Data Figure 18c-e, Extended**
**Data Figure 19c-e**). Finally, we now provide an overview of the best response of individual
tumors, grouped by mouse. This analysis revealed significant heterogeneity in the
response of individual tumors within each mouse, which does not seem to differ from
the heterogeneity across the whole cohorts, suggesting that the degree of response is
intrinsic of each individual tumor (**Extended Data Figure 18f, Extended Data Figure 19f**).
The text describing this improved analysis reads as follows:

“The response was evaluated by longitudinal MR imaging every two weeks. While
all tumors from vehicle-treated mice progressed during treatment, the response
was significantly improved both in the DCPG cohort and in the riluzole cohort, with
several tumors showing short-term stable disease or slower growth and a small
subset of tumors showing modest shrinkage (**Fig. 6a, Extended Data Fig. 17a-c,**
**Extended Data Fig. 18a-f**). The DCPG- and riluzole-treated mice also showed a
significantly improved survival, with a median survival of 66 days (DCPG) and 71.5
891 days (riluzole), compared to 54 days in the controls (**Fig. 6b**). We then compared
animals treated with frontline cisplatin/etoposide chemotherapy with mice that
received chemotherapy with the addition of DCPG or riluzole. Tumors exposed to
chemotherapy showed a mixed response, which included shrinkage, stable
disease and progressive disease. In the context of chemotherapy, the inclusion of
DCPG did not significantly improve the response of SCLC tumors (**Fig. 6c,**
**Extended Data Fig. 17d, e, Extended Data Fig. 19a-f**). In contrast, the
combination of chemotherapy and riluzole resulted in a significantly improved
response, with almost all tumors displaying partial response or stable disease and
slower growth for more than two months (**Fig. 6c, Extended Data Fig. 17d, f,**
**Extended Data Fig. 19a-f**). Similarly, the inclusion of DCPG into the frontline

chemotherapy regimen in our mice did not result in significantly improved survival,
while the inclusion of riluzole resulted in significant and substantial improvement
of 21 days (**Fig. 6d**).”

Minor:

1. While Rudin et al is a thorough review, it may not be sufficient to cover all of the points in
the opening paragraph.

1) We thank the reviewer for raising this point. We have now modified the introductory
paragraph to include three different reviews: Rudin *et al*, Poirier *et al*. and Ditmanns *et*
*al*. Together, these comprehensive works cover all aspects mentioned in the paragraph.
The revised version now reads:

“Small cell lung cancer (SCLC) constitutes approximately 13-15 % of all lung
cancers in humans and is associated with a very poor prognosis³⁸⁻⁴⁰. In the
majority of cases, SCLC patients are diagnosed with metastatic extensive stage
disease^{38,40}. The typical treatment is combination chemotherapy, consisting of a
platinum salt, etoposide and immune checkpoint blockade plus optional
prophylactic cranial irradiation^{38,40}. This intensive induction regimen induces
clinical responses in more than 60 % of the cases³⁸. However, these responses
are almost always transient, resulting in a median overall survival of approximately
one year for patients with extensive stage disease^{38,40}.”

2. Introduction: “SCLC derives from pulmonary neuroendocrine cells” is over-stated and
needs references. It has been shown that other cells can serve as a cell of origin for SCLC
and this is highly influenced by genetic changes (Sutherland et al, Cancer Cell, 2011; Olsen et
al, Genes Dev, 2021; Chen et al, Oncogene, 2022). It is more accurate that SCLC highly
resembles PNECs, and may often derive from them, but we also know that adenocarcinoma
can transition to SCLC (Oser et al, Lancet Oncology, 2015), making it clear that a tumor does
not have to originate in a PNEC to become PNEC-like.

2) We thank the reviewer for this important comment and we have now adjusted our
introduction, results, discussion and figure legends to avoid the erroneous idea that SCLC
only derives from PNECs. Instead, as suggested, we introduce the different potential cells
of origin in the introduction and we focus on the fact that SCLC highly resembles PNECs
in the other sections. The relevant paragraphs now reads:

“Comprehensive genomic profiling has revealed that SCLC displays an almost
universal bi-allelic loss of the tumor suppressors *TP53* and *RB1*^{11,12}. Accordingly,
SCLC develops in mice after bi-allelic deletion of *Trp53* and *Rb1* in lung epithelial

cells⁸. Several studies have shown that pulmonary neuroendocrine cells (PNECs)
are a permissive cell type of origin for SCLC, but other cell types can also give rise
to SCLC upon loss of *Trp53* and *Rb1*, especially when oncogenic *Myc* is
concomitantly overexpressed^{7,13-15}. Interestingly, these non-neuroendocrine lung
epithelial cells acquire a PNEC-like phenotype and express neuroendocrine
markers during tumor development¹³. Similarly, human *EGFR*-driven
adenocarcinoma can transdifferentiate into SCLC and start expressing
neuroendocrine markers upon loss of *RB1*^{16,17}.”

“This analysis indicated that the SCLC-specific expression of neuronal and
synaptic gene sets is part of the PNEC-like phenotype of SCLC”

“Second, the high expression of neuronal and synaptic gene sets, which are
part of the PNEC-like phenotype of SCLC and are strikingly overlapping with the
GO terms we identified at the genetic level.”

“Similar to the overall expression of synaptic genes, we speculate that the ability
to form synapses is part of the PNEC-like phenotype of SCLC.”

“A high fold change indicates that the upregulated genes within the gene set are
also upregulated in healthy PNECs.”

3. A number of figure legends are missing statistical tests used including Fig 2c/f, Fig 3i, Fig
5, etc.

3) We have now included the information about the statistical tests used in all figure
legends

4. Figure 4e: typo in the figure “colture” instead of “culture”

4) In order to answer the several questions regarding our co-culture experiments raised by
all three reviewers, we have now replaced our initial FACS experiments with live cell
imaging measurements using the more efficient Incucyte system. As a result, panel 4e
and the typo have been removed.

5. Figure 4e: This form of analysis as a group would be more accurate to provide individual

cell line comparisons, with each cell line by itself plus and minus neurons. Statistical method
and p value or q value (figure vs legend) should be clarified.

5) We thank the reviewer for this comment. As mentioned above, we have switched to live
cell imaging instead of FACS measurements. As requested, we have now included an
overview of the individual cell lines, in **Figure 5f**, as well as the complete view of all
replicates for all cell lines in **Extended Data Figures 14, 15**. We also included the details
of the statistical test in the legend of **Extended Data Figures 14 and 15**.

6. Figure 5a: Can you apply the NE signature used in Zhang et al, TLCR, 2018 which is a 50
gene signature that is more robust than the current 5 gene NE signature?

6) We thank the reviewer for this comment. We have now replaced the 5 gene NE signature
with the more robust 50 gene signatures described in Zhang *et al*. This change positively
affected the separation of the two subtypes and caused SCLC to become the top-scoring
type of cancer, as would be expected by a lung-specific neuroendocrine signature
(**Extended Data Figure 16a**). The relevant paragraph now reads:

“The SCLC samples that we analyzed at the expression level can be broadly

classified into classic SCLC with strong neuroendocrine features and variant

SCLC with lower expression of neuroendocrine features, using a lung-specific

neuroendocrine score ⁴¹ (**Extended Data Fig. 16a**).”

7. Extended Data Figure 1a: The unrecombined Trp53 allele from Meuwissen et al, Cancer
Cell, 2003 has loxP sites flanking exons 2-10, thus the last exon after the LoxP site should be
Exon 11 instead of 10.

7) We thank the reviewer for noticing this mistake and we have corrected the erroneous
panel.

8. Extended Data Figure 1d: The authors did not comment on whether any Piggybac
mutations were enriched in metastases compared to primary tumors. This may be
particularly interesting, as some genes do have migration-related functions such as Reelin
and Netrin signaling.

8) Similar to the analysis across treatment cohorts, we used a permutation test and FDR
correction to investigate potential differentially inserted genes between metastatic and
primary samples. While we were hoping to detect differences between primary and
metastatic samples, no gene showed a significantly different insertion pattern. As also
mentioned in our response to reviewer 1, we believe that the bulk of the signal in our
screen is established in the early phases of tumorigenesis, in the latency period between
recombination and tumor detection. We have included the q values of the comparison in
**Extended Data Table 3** and we now mention this analysis in the relevant results and
methods sections which now read:

“Initial analysis using a random permutation test did not reveal any gene with a
significantly different number of insertions between untreated,
cisplatin/etoposide-exposed or anti-PD1 antibody-treated tumors or between
primary and metastatic tumors (**Extended Data Tables 1-3**).”

“We used a permutation test to compare the distribution of the transposon
insertions in different sub-cohorts: untreated versus chemotherapy, untreated
versus immunotherapy and lung tumors versus metastatic tumors. For each
comparison, the union of samples included in the comparison was shuffled
1000000 times, while maintaining the same number of samples from each mouse
line in each sub-cohort (RPLH and RPLS). For each gene, we then counted the
number of iterations in which the absolute difference in the fraction of samples
carrying an insertion is greater than in the real configuration. We then calculated
the FDR-corrected q values for each gene.”

9. Discussion: “As SCLC is a cancer entity...high degree of inter- and intratumor
heterogeneity and plasticity” is missing Ireland et al, Cancer Cell, 2020 as a key reference.

9) We thank the reviewer for this comment and we have now included the requested
reference.

Referee #3 (Remarks to the Author):

Schmitt and colleagues examine genetic parameters contributing to the growth of small cell
lung cancer using mouse and human models/data. The use an elegant piggyBAC
mutagenesis screen in a mouse model to confirm previously known genes with
contributions to tumor growth and identify – surprisingly – neuronal axon guidance and
synaptogenic signaling components. Using in vitro experiments, the authors demonstrate
that mouse cortical neurons can form synapse-like contacts onto human SCLC cell lines.

The notion that neuron-tumor interactions contribute to tumor growth has recently
attracted a lot of attention. The strongest evidence supporting such a role is derived from
work on gliomas. Thus, glioma cells have been demonstrated to express critical
synaptogenic molecules and glutamate release from neurons has been demonstrated to
promote tumor cell proliferation and enhance invasion (Venkataramani et al. 2019;
Venkatesh et al., 2019; Venkataramani et al., 2019, 2022) and there is data published in

preprint form suggestion a related mechanism for SCLC tumors (Savchuk et al., biorxv).
However, almost all of this work relies on gain-of-function manipulations such as non-
physiological optogenetic stimulation. The study by Schmitt and colleagues stands out as it
uses both genetic and pharmacological loss-of-function approaches to investigate molecular
and functional components that modify tumor progression.

The work is very interesting and seems overall well-controlled, well-documented and the
manuscript is very well written. There are several points where the individual observations
of the study are not well connected. Addressing these points would significantly improve
the study:

We thank the reviewer for the extremely positive evaluation of our manuscript and for the
helpful and constructive comments, which helped us improve several aspects of our
manuscript related to the interactions between cancer cells and neurons.

1) A major question is regarding the logic connecting the loss-of-function screen and follow-
up experiments. Identifying synapse-components in the piggyBAC screen is intriguing – but
this would indicate that loss of synapse proteins enhances tumor progression. However, the
remainder of the study suggests the opposite, i.e. that synapse formation and synaptic
transmission onto SCLC cells promotes tumor growth. How can this be reconciled? Does this
relate to differential action of glutamatergic versus GABAergic receptors? Many
components of the NRXN-NLGN system are shared between glutamatergic and GABAergic
synapses. Moreover, the authors also identify GABA-specific molecules (CLSTN2, GABRB1).

1) We thank the reviewer for this insightful question. Indeed, both the *piggyBac* system and
the missense mutations in human samples rarely allow clear conclusions regarding the
effects of the genetic events. *PiggyBac* insertions can have a suppressing or an activating
function (Rad *et al.* Science, 2010⁴²), we now include this point in the description of the
model. The relevant text now reads as follows:

“To allow for *in vivo* transposon mutagenesis, we crossed RPL mice with *ATP1-*
*S2* (S, harboring 20 transposon copies on chromosome 10) or *ATP-H39* (H,
harboring 80 transposon copies on chromosome 5) mice (**Extended Data Fig.**
**1b**). Depending on the integration site relative to the target gene, the transposons
can intercept transcription and functionally knock out the gene by splicing into a
transposon splice acceptor followed by a poly-A signal, or activate gene
expression of different isoforms through the CAG promoter⁴².”

For the human mutation data, while an enrichment in clear loss of function mutations
suggests a tumor suppressor role for some genes, such as *GRM8* and *RELN*, missense
mutations can potentially have both gain-of function and loss-of-function effects.
Therefore, we see our genetic data as a tool to identify relevant gene sets and we rather
rely on our functional investigations for the conclusion that the interactions with neurons
have a tumor-promoting effect. In line with this view, while the genes that we identified

at the genetic level are often enriched in the same gene sets as the genes we identified
at the expression level, they do not actually significantly overlap (e.g., for the
*Glutamatergic Synapse* term, 20 genes are identified at the genetic level and 54 at the
expression level, but only 3 overlap). Therefore, the highly expressed genes are usually
not significantly mutated, even though they are part of the same gene sets as mutated
genes. Importantly, the two types of analysis reflect two different snapshots in tumor
biology: genetic data relates to initiation, while expression data reflects the steady state
of developed tumors. This suggests that the neuronal pathways are highly expressed and
active overall and that genetic modulation of individual genes within these gene sets
provides tumorigenic stimulus.

We also thank the reviewer for pointing out the overlap between genes we identified
within the glutamatergic and GABAergic synapses as well as the presence of GABA-
specific genes. The potential relevance of GABAergic synapses is further underscored by
the reduction in sPSCs in our new *in vitro* experiments in the presence of bicuculline (see
the response to reviewer point 5) as well as the identification of both excitatory and
inhibitory neurons in our new *in vivo* rabies experiments (see the response to reviewer
point 4). We have therefore now included a mention of this pathway in the description
of the results of our *piggyBac* screen, as well as in the discussion. The relevant sections
now read:

“To functionally categorize the composition of the genes detected in the
human sequencing data sets and in the *piggyBac* screen, we first asked which
gene ontology (GO) gene sets^{43,44} are significantly enriched in both data sets (**Fig.**
**1b, Extended Data Fig. 5h, Extended Data Tables 6, 7**). Unexpectedly, the vast
majority of the enriched terms were related to neuronal phenotypes and synaptic
functions, such as *Synaptic Membrane*, *Glutamatergic Synapse*, *Glutamate*
*Receptor Activity*, *GABA-ergic synapse* and *Transsynaptic Signaling*, among
others.”

“Similarly, while we focused mainly on glutamatergic contacts, the *GABA-ergic*
*synapse* GO term was also identified in our genetic screen and the potential for
GABA-ergic communication between neurons and SCLC cells was corroborated
by our electrophysiological and tracing experiments. A systematic exploration and
cataloguing of synaptic and non-synaptic (perisynaptic and paracrine neuro-ligand
signaling) interactions between neurons and SCLC cells will constitute an
important next step forward.”

2) The authors hypothesize that GRM8 is a major factor and provide evidence that it might
serve as a therapeutic target.

- a. For coherence of the data presentation, it would be valuable to show the insertion data
for GRM8 in Figure 1 and Extended data figure 2.
b. Are the insertions thought to result in a gain or loss of function?
c. Do the SCLC cell lines used express GRM8 and does GRM8 inhibition modify growth of
these cells in co-culture?

2) We thank the reviewer for these questions. We have now marked *Grm8* in **Figure 1a** and
included the insertion pattern of *Grm8* in **Extended Data Figure 16**.

There are not enough insertions in *Grm8* to predict their effect with certainty, but the
ones that are present point in both directions and are therefore consistent with a loss of
function. In addition, the human mutations include a statistically significant number of
loss-of-function mutations, suggesting that the loss of *GRM8* may be a selected event
(**Revised Extended Data Figure 16d**).

As also discussed in our response to reviewer 2, the cell lines that we tested express no
GRM8 or very low levels of GRM8 (see below, **Reviewer Figure 1**). In addition, while
*GRM8* is expressed at higher levels in human bulk and scRNAseq samples compared to
other tissues and cancers, the expression is rarer in the advanced murine tumors that we
sequenced (**Extended Data Figure 16g, h**). While these findings are compatible with loss
of *GRM8* being a selected event, they also greatly complicate its experimental
investigation. As the response to DCPG *in vivo* is inferior to riluzole, especially in the new
cohort of mice requested by reviewer 2, we chose to de-emphasize the *GRM8* and DCPG
angle of the manuscript, by merging the original figure 5 and the original Extended Data
Figure 12 into a more compact supplementary figure (**Extended Data Figure 16**).

The revised text section reads as follows:

“Among the possible molecular targets within this system are the glutamate
receptors, which we also identified as individual genes targeted by transposon
insertions in our *piggyBac* screen (*Grid1*, *Grik2*, *Grin3a*, *Grm1*, *Grm3*, *Grm5*,
*Grm8*), in human mutation data (*GRIA1*, *GRIA2*, *GRIA3*, *GRIA4*, *GRID2*, *GRIK2*,
*GRIK3*, *GRIK4*, *GRIN2A*, *GRIN2B*, *GRIN3A*, *GRM1*, *GRM3*, *GRM5*, *GRM8*) and
at the expression level in human samples (*GRIA2*, *GRIN3A*, *GRIK3*, *GRIK5*,
*GRM2*, *GRM4*, *GRM8*). Prominent among them was *GRM8*, a gene encoding an
inhibitory metabotropic glutamate receptor that has been shown to counteract
glutamate signaling by negatively regulating cAMP-dependent sensitization of
inositol 1,4,5-trisphosphate receptors, thereby limiting glutamate-induced
calcium release from the ER³⁷. In our human datasets, *GRM8* shows specific
expression in SCLC and a few other tumor types (**Extended Data Fig. 16c**) and
a statistically significant enrichment of both non-synonymous mutations and
more severe loss of function mutations (stop, frameshift, start-loss, and canonical
splice-site mutations) (**Extended Data Fig. 16d**). Re-analysis of scRNA-seq data
from Chan *et al.*⁶ confirmed that *GRM8* is specifically expressed in SCLC cells

(Extended Data Fig. 16f, g). We also find specific expression of *Grm8* in our
murine SCLC snRNA dataset, although at a substantially reduced fraction,
compared to the human dataset (Extended Data Fig. 16h).”

3) Fig.2a: These experiments are not straightforward to interpret. The neuronal density
applied in this study, light microscopy does not allow to unambiguously identify direct
contacts and to exclude the presence of dendritic processes. It is visible from the images
provided that there are many vGLUT1-positive structures forming on the dendrites, i.e. non-
tumor cells in the field of view. Ultrastructural investigations would be ideal to resolve this.
At the very least, the authors should include the MAP2 marker in their high magnification
views to confirm that there are no MAP2-positive dendrites between neuron and tumor cell.

3) We thank the reviewer for this important comment, which was raised in a similar fashion
by reviewer 1. As also stated in our response to reviewer 1, we have now been able to
visualize *bona fide* synaptic structures by deploying super-resolution imaging, expansion
microscopy as well as EM, and corroborated these with additional high-resolution
imaging of synaptic contacts in brain allografts.

In co-cultures of human SCLC cells and neurons, we visualized the synaptic contacts using
three types of super-resolution microscopy, with increasing resolution: STED microscopy
(40-50 nm), X10 expansion microscopy (20-25 nm resolution), and ONE microscopy (1-5
1211 nm resolution). These tools enabled us to demonstrate the co-localization of pre- and
1212 post-synaptic markers (Vglut1 and Homer1) on neurons and cancer cells, respectively.
Within these contacts, the distance between the pre-synaptic Vglut1 and the post-
synaptic Homer1 fell in the range of that observed in *bona fide* synapses between
neurons present in these co-cultures (Figure 3g, h). Importantly, we now also show the
presence of colocalizing axons and post-synaptic Homer1 in the absence of MAP2-
positive dendrites in SCLC-neuron co-cultures (Extended Data Figure 10e). Altogether,
these new data unambiguously show that pre-synaptic axonal structures contact the
post-synaptic cancer cells.

The revised text describing these new data reads as follows:

“Lastly, we demonstrate that these points of contacts on cancer cells mostly
occurred with neuronal axons marked by phosphorylated neurofilaments (anti-
SMI-312 antibody) and not with dendrites immunoreactive for MAP2 (Extended
Data Fig. 10e).”

“To investigate the nature of these contacts, we performed stimulated
emission depletion (STED) microscopy of SCLC-neuron co-cultures.
Immunostainings for glutamatergic vesicles (anti-VGLuT1) and the postsynaptic
protein HOMER1 revealed colocalizing formations at the contacts between
neurons and cancer cells (Fig. 3a, b). To better inspect the structure of these

contacts, we next turned to 10-fold expansion microscopy, which reaches ~25 nm
resolution ¹. The 3D reconstruction of these samples disclosed a spatial
organization consistent with synaptic structures, with VGlut1-positive puncta
outside of the cancer cells juxtaposed to HOMER1 immunoreactivity within the
cancer cells (**Fig. 3c**). To visualize synapses in even higher detail, with single-digit
nanometer resolution, we employed ONE microscopy ². 3D and 2D ONE images
presented the clear separation of the pre- and post-synaptic elements, and
demonstrated that their localization resembles classical synapses between CNS
neurons (**Fig. 3d, e**). Importantly, the distance between the VGlut1- and the
HOMER1-positive puncta was comparable with that from *bona fide* synapses
(neuron-to-neuron synapses) within the same cultures (**Fig. 3f-h**).”

In addition to the super-resolution experiments detailed above, we also conducted
additional EM experiments, as suggested by this reviewer and reviewer number 1. As a
result, we can now show direct juxtaposition of presynaptic axonal boutons with
postsynaptic cancer cells identified by CLEM. Crucially, we found several examples of
vesicles docked or fusing towards the cancer cells both *in vitro* in co cultures and *in-vivo*
in allografts (**Figure 3i-k, Extended Data Figure 11b**). Collectively, we believe these new
data to provide a strong support for the establishment of *bona fide* synaptic contacts.
The revised text describing these new data reads as follows:

“To further characterize these putative synaptic contacts at the ultrastructural
level, we next deployed Correlative Light Electron Microscopy (CLEM) in brain
allografts and co-cultures. Inspection of DsRed-positive SCLC cells confirmed the
presence of synaptic boutons filled with vesicles contacting the plasma membrane
of cancer cells (**Extended Data Fig. 11b**). Importantly, additional ultrastructural
hallmarks of stereotypical synapses were identified, including the presence of a
synaptic cleft and vesicles docked to or undergoing fusion with the presynaptic
membrane (**Fig. 3i-k, Extended Data Fig. 11b**).”

4) Fig.2b. Rabies tracing is not a suitable method to assess function of synapses as stated in
the text and not even a good method to confirm synaptic structures. There is increasing
evidence that rabies viruses can also spread through non-synaptic membrane contacts, and
in *in vitro* preparations this is an even bigger issue. The authors need to modify the text
accordingly and / or consider moving this information to the supplement to not
unintentionally mis-lead an uninformed reader.

4) We thank the reviewer for this insightful comment. We acknowledge the limitations of
this approach and have thus de-emphasized the rabies tracing experiments in the main
text as well as pointed out the potential for non-specific transmission. Moreover, we have
switched the order of our description of the rabies and electrophysiological data. We now
place the electrophysiology data upfront and use the rabies tracing experiments merely
to corroborate our electrophysiology data. In addition, we performed important further
control experiments using a retrovirus devoid of the glycoprotein G and only expressing
TVA to be able to quantify any spurious labelling of RABV due to non-specific spread,
which could potentially take place via non-synaptic mechanisms. The minor labeling
observed with the use of this control retrovirus support the specificity of the labeling
observed when the G-TVA retro was used, both *in vitro* and *in vivo* (**Figure 4g-k, Extended**
**Data Figure 13b**), suggesting the RABV spread to predominantly occur via synaptic
contacts.

In addition, we also performed *in vivo* rabies tracing experiments in brain transplants.
Here, we injected rabies virus in the hippocampus together with murine cancer cells
engineered to express either only the TVA receptor or both TVA and the glycoprotein G.
These experiments revealed a low-level unspecific uptake of rabies virus by glial cells and
neurons in the TVA-only experiment and a strong and significant increase in the number
of infected neurons in the TVA-G experiment (**Figure 4j, k, Extended Data Figure 13**),
suggesting that cancer cells can integrate into neuronal circuits *in vivo*. Importantly, both
inhibitory and excitatory neurons were labeled by the rabies virus, revealing the ability
of cancer cells to form synapses with different types of neurons (**Figure 4j, k, Extended**
**Data Figure 13**).

The revised text describing these new data reads as follows:

“To further substantiate these findings, we performed retrograde
monosynaptic tracing experiments of SCLC cells using an established replication-
incompetent EnvA-pseudotyped Δ G RABV-GFP, which can only infect cells
expressing the avian viral receptor TVA. Upon initial infection, cells that
complement the RABV glycoprotein (G) expression are then able to transmit the
virus to their first-order presynaptic partners⁴⁵. After transduction of SCLC cells
with a DsRed retrovirus encoding for TVA and G (G-TVA), we co-cultured them
with cortical neurons and then added RABV-GFP to the culture media (**Fig. 4g**).
In line with a retrograde RABV-GFP spread from DsRed-positive SCLC “starter”
cells into synaptically connected neurons, we detected DsRed and GFP double-
positive SCLC cells being surrounded by clusters of GFP-positive neurons (**Fig.**
**4g, Extended Data Fig. 12a, b**). We also found strong VGluT1 immunoreactivity
in presynaptic RABV-infected neuronal fibers impinging onto SCLC starter cells in
these samples (**Extended Data Fig. 12c**). We further validated the expected
transsynaptic RABV-GFP spread using time-lapse experiments of these co-

cultures, showing that neurons acquire GFP fluorescence only starting from 48hrs
after the appearance of DsRed/GFP double-positive SCLC starter cells
(**Extended Data Fig. 12d**). Assessment of GFP-positive neurons in co-cultures
with SCLC cells lacking any prior retroviral transduction, or expressing only TVA
and/or DsRed (but not G) as negative controls revealed a low and quantifiable,
spurious labelling by RABV in our settings (**Fig. 4h**). In contrast, co-expression of
G in SCLC cells resulted in a net increase of neuronal labelling of 10-fold or more,
corroborating the reliability of this transsynaptic approach (**Fig. 4h**). Analysis of
co-cultures containing either COR-L88 cells (SCLC-A subtype) or DMS273 cells
(SCLC-N subtype) displayed a connectivity ratio of 3 to 12 neurons per “starter”
cancer cell (**Fig. 4i**).

We next conducted transsynaptic tracing experiments in brain allografts *in*
*vivo*, by stereotactically co-injecting G-TVA-expressing- or TVA-only-expressing-
SCLC cells and EnvA-pseudotyped Δ G RABV-GFP in the mouse hippocampus
(**Fig. 4j**). In G-TVA injected animals, the DsRed/GFP double-positive SCLC starter
cells were typically surrounded by GFP-only-positive axonal fibers (**Fig. 4j**,
**Extended Data Fig. 13a**). Consistently, GFP-positive neurons were found in the
regions (hippocampus and subiculum) adjacent to the grafted G-TVA-expressing
SCLC cells, while control experiments with cancer cells expressing exclusively
TVA resulted in absent or minor neuronal labelling (**Fig. 4k, Extended Data Fig.**
**13b**). Classification of neurons according to their morphology and layer positioning
in the traced anatomical regions near the injected area disclosed both putative
excitatory and inhibitory neurons, further indicating that SCLC cells can be
innervated by distinct neuronal sub-types *in vivo* (**Extended Data Fig. 13c**).”

5) Fig.2d-f. The electrophysiological assessments are much more convincing than the
anatomical and rabies data. Considering that several of the genes identified in the piggyBAC
screen are GABAergic synapse components, the authors should perform additional
recordings to separately isolate glutamatergic and GABAergic currents. This is important
with regards to the interpretation of the data overall (see below).

5) We thank the reviewer for this important comment. As requested, we performed further
electrophysiological recordings in the presence of the GABA inhibitor bicuculline to
isolate GABAergic currents, as well as additional recordings with CNQX to inhibit AMPA
glutamatergic currents, AP5 to inhibit NMDA glutamatergic currents and TTX to inhibit
overall neuronal activity. With various degree of efficiency, each of these agents was able

to partially but significantly reduce the sPSCs in cancer cells (**Figure 4c**), suggesting that
in the co-culture settings, SCLC cells can engage in promiscuous synaptic contacts with
excitatory and inhibitory GABAergic neurons. This was corroborated by monosynaptic
tracing experiments conducted in brain allografts, in which both presynaptic excitatory
and inhibitory GABAergic neurons were identified, although a tendency towards
excitatory inputs was observed (**Figure 4j-k** and **Extended Data Figure 13**). The revised
text describing these new data reads as follows:

“To assess the functionality of synaptic contacts between neurons and cancer
cells, we next conducted electrophysiological recordings of cancer cells in co-
culture with cortical neurons. While whole-cell patch-clamp recordings of COR-
L88 mono-cultures revealed the absence of any spontaneous inputs, the same
cells developed spontaneous postsynaptic currents (sPSCs) when co-cultured
with cortical neurons (**Fig. 4a, b**). These currents were reduced when the co-
cultures were treated with the voltage-gated sodium channel blocker TTX, with the
AMPA receptor antagonist CNQX, with the NMDA receptor antagonist AP5, with
the glutamate release inhibitor 2-amino-6-trifluoromethoxy benzothiazole (riluzole)
³ and, interestingly, also with the GABA receptor inhibitor bicuculline, suggesting
that cancer cells in co-culture are able to promiscuously form contacts with both
excitatory and inhibitory neurons (**Fig. 4c**). Importantly, optogenetic stimulation of
co-cultured neurons expressing channelrhodopsin-2 reversibly elicited fast inward
currents in recorded SCLC cells, further corroborating the existence of direct
synaptic contacts (**Fig. 4d**).“

6) The authors provide in vivo data on SCLC innervation using allograft experiments. The
data on nerve fibers and endings in proximity of tumors in the lung are less compelling. I
appreciate the inclusion of EM data – but at present these are not very convincing and it
remains unclear at what frequency such structures were observed. Any experiments
strengthening this would greatly increase the impact of this work. This could be done by
additional ultrastructural investigations or (if a suitable cre line is available) re-deploying the
rabies approach which should have more validity in an in vivo setting than in vitro.

6) We thank the reviewer for this important comment. We performed three sets of
experiments to address this point.

First, we performed additional immunohistochemistry experiments on primary murine
lung tumors, which now more clearly show intraepithelial arborization of vglut1-positive
fibers between the cancer cells. These fibers show a morphology that is consistent with
the Vglut1-positive vagal fibers from the nodose ganglion that innervate healthy PNECs
(Brouns *et al. Histochem Cell Biol* 2009 ⁴⁶), which constitute the most permissive cell of

origin of SCLC (Sutherland *et al. Cancer Cell* 2011 ⁷). Of note, we can distinguish regular
PNECs from incipient SCLC tumors not only morphologically (disarrayed epithelial
structures and larger cluster size), but also through Cre-mediated GFP expression in the
cancerous lesions (brought about by a *Rosa26^{LSL-Cas9-IRES-EFGP}* allele) (**Figure 2d-f**). The
revised text describing these new data reads as follows:

“Using mice that additionally carry a GFP-marked allele
(*Rb1^{fl/fl};Trp53^{fl/fl};Rosa26^{Cas9-GFP}*, RPC), we detected VGluT1-positive fibers
arborizing between the neuroendocrine cells starting from the very initial stages of
transformation up to small and medium size tumors (**Fig. 2d-f**).”

Second, we performed additional EM experiments and 3D reconstructions of synaptic
contacts in SCLC cell allografts. These experiments are now depicted in **Figure 3i, j** and
**Extended Data Figure 11b**, and illustrate the presence of vesicles docked or fusing with
the membrane of the presynaptic axonal bouton contacting the reporter-positive cancer
cell identified by CLEM. The text describing these new data reads as follows:

“To further characterize these putative synaptic contacts at the ultrastructural
level, we next deployed Correlative Light Electron Microscopy (CLEM) in brain
allografts and co-cultures. Inspection of DsRed-positive SCLC cells confirmed the
presence of synaptic boutons filled with vesicles contacting the plasma membrane
of cancer cells (**Extended Data Fig. 11b**). Importantly, additional ultrastructural
hallmarks of stereotypical synapses were identified, including the presence of a
synaptic cleft and vesicles docked to or undergoing fusion with the presynaptic
membrane (**Fig. 3i-k, Extended Data Fig. 11b**).”

Lastly, as detailed above, we performed *in vivo* rabies monosynaptic tracing experiments,
using appropriate controls, which showed excitatory and inhibitory pre-synaptic neurons
being innervating grafted cancer cells (**Figure 4j-k** and **Extended Data Figure 13**).
Altogether, these data now provide stronger evidence for *in vivo* innervation.

7) Fig.4. The cell proliferation effect observed in mono- versus co-culture is interesting.
However, this analysis remains somewhat preliminary. There is no evidence that this effect
is specific to SCLC cells (would another cell line also increase proliferation in co-culture?) or
that it relates to synaptic transmission from neurons. One way to assess the latter possibility
would be blocking sodium channel-dependent action potentials with TTX, or – in line with
the subsequent experiments – apply GRM8 inhibitors to test whether these effects are
related. I appreciate that in this reductionist system inhibitors may not provide the data that
are expected based on the other observations in more intact system. In this case, it might be

worth considering to remove this experiment from the manuscript. As is, it seems too
preliminary for a main figure.

7) We thank the reviewer for this comment. We have performed several experiments to
improve figure 4 (now **Figure 5**). We can now show that the effect is not completely
specific of SCLC, as adenocarcinoma lines also derive a small but significant advantage.
We could also show that the advantage is much greater when the cancer cells are in direct
presence of the neurons, as opposed to being exposed to media conditioned by neurons
(**Figure 5f, Extended Data Figure 14a-h**). Finally, we could show that neurons provide a
stronger proliferation advantage than fibroblasts or higher seeding density (**Extended**
**Data Figure 14a-h**). The revised text describing these new findings reads as follows:

“To evaluate whether neuronal co-culture stimulates proliferation in all well-
defined SCLC subtypes, we monitored the growth of eight distinct cell lines with
live cell imaging: COR-L88, H1836, H69 and H146 for the SCLC-A subtype,
DMS273 and H524 for the SCLC-N subtype and H211 and H526 for the SCLC-P
subtype. All tested lines derived a significant proliferation advantage when co-
cultured with cortical neurons (**Fig. 5e, f, Extended Data Fig. 14a-h**). Importantly,
some cell lines (COR-L88, H69, DMS273, H524, H211) also derived a minor
proliferation advantage when cultured in conditioned media derived from neuronal
cultures. However, for all cell lines, the physical presence of the neurons conferred
a significantly stronger proliferation advantage (**Fig. 5f, Extended Data Fig. 14a-**
**h**). The proliferative advantage appears to be specific for neurons, as it vastly
exceeded that observed in high-density mono-cultures (**Extended Data Fig. 14a-**
**h**) and it was even stronger in four of the cell lines than that conferred by co-
cultures with fibroblasts, which have been shown to strongly promote SCLC
growth ⁴ (**Extended Data Fig. 14a-h**). Moreover, the effect appeared to be largely
specific for SCLC, as four non-small cell lung cancer (NSCLC) lines (H1975,
HCC44, HOP62, H2291) derived only a minor proliferation advantage when co-
cultured with neurons (**Fig. 5g, Extended Data Fig. 14i-l**).”

As suggested, we also investigated whether blocking sodium channel-dependent action
potentials with TTX can influence the proliferation advantage provided by neurons using
EdU incorporation measurements. TTX had no substantial effects on the proliferation of
cancer cells in mono-cultures, but significantly reduced the proliferation of cancer cells
in co-cultures with neurons (**Figure 5c, d**). The revised text describing these new findings
reads as follows:

“To this end, we compared the proliferative capacity of DsRed-expressing
human COR-L88 cells seeded at low density and maintained either alone (mono-
culture) or in co-culture with cortical neurons, followed by EdU labeling 2h prior to
analysis. While only few scattered EdU-positive SCLC cells were found in mono-
cultures, SCLC cells in co-cultures often appeared as larger proliferating clusters
(Fig. 5a). Importantly, this effect was significantly reduced when the co-cultures
were treated with TTX, suggesting that the co-culture-specific proliferative
advantage of SCLC cells is at least partially mediated by neuronal activity (Fig.
5c, d).”

These new data suggest that physical proximity and neuronal activity are required for the
full proliferation advantage conferred by neurons to cancer cells.

Reviewer Figure 1: SCLC lines show low and heterogeneous expression of GRM8

SCLC lines of different subtypes were stained with antibody against GRM8 (Thermofisher, # PA5-
11697) and actin as a loading control (Santa Cruz, sc-47778). Mouse brain lysate was used as
positive control. The cell lines were, in order: H146, COR-L88, H69 and H1836 for the SCLC-A
subtype, H524 and DMS273 for the SCLC-N subtype and H526 and H211 for the POU2F3 subtype.
A faint band was visible only at high exposure.

**References**

- 1. Truckenbrodt *et al.* X10 expansion microscopy enables 25-nm resolution on
conventional microscopes. *EMBO Rep* (2018).
- 2. Shaib *et al.* Visualizing proteins by expansion microscopy. Preprint at
<https://doi.org/10.1101/2022.08.03.502284> (2023).
- 3. Martin *et al.* The neuroprotective agent riluzole inhibits release of glutamate and
aspartate from slices of hippocampal area CA1. *European Journal of Pharmacology*
(1993).
- 4. Sen *et al.* Development of a small cell lung cancer organoid model to study cellular
interactions and survival after chemotherapy. *Front Pharmacol* (2023).
- 5. Brouns *et al.* Neurochemical pattern of the complex innervation of neuroepithelial
bodies in mouse lungs. *Histochemistry and Cell Biology* (2009).
- 6. Chan *et al.* Signatures of plasticity, metastasis, and immunosuppression in an atlas of
human small cell lung cancer. *Cancer Cell* (2021).
- 7. Sutherland *et al.* Cell of Origin of Small Cell Lung Cancer: Inactivation of Trp53 and Rb1
in Distinct Cell Types of Adult Mouse Lung. *Cancer Cell* (2011).
- 8. Meuwissen *et al.* Induction of small cell lung cancer by somatic inactivation of both
Trp53 and Rb1 in a conditional mouse model. *Cancer Cell* (2003).
- 9. Brouns *et al.* Pulmonary Sensory Receptors. *Adv Anat Embryol Cell Biol* (2021).
- 10. Doerr *et al.* Targeting a non-oncogene addiction to the ATR/CHK1 axis for the treatment
of small cell lung cancer. *Scientific Reports* (2017).
- 11. George *et al.* Comprehensive genomic profiles of small cell lung cancer. *Nature* (2015).
- 12. George *et al.* Evolutionary trajectories of small cell lung cancer under therapy. *Nature*
(2024).

- 13. Chen *et al.* Lineage-restricted neoplasia driven by Myc defaults to small cell lung cancer
when combined with loss of p53 and Rb in the airway epithelium. *Oncogene* (2022).
- 14. Olsen *et al.* ASCL1 represses a SOX9+ neural crest stem-like state in small cell lung
cancer. *Genes Dev.* (2021).
- 15. Yang *et al.* Intertumoral heterogeneity in sclc is influenced by the cell type of origin.
*Cancer Discovery* (2018).
- 16. Oser *et al.* Transformation from non-small-cell lung cancer to small-cell lung cancer:
molecular drivers and cells of origin. *Lancet Oncol* (2015).
- 17. Niederst *et al.* RB loss in resistant EGFR mutant lung adenocarcinomas that transform to
small-cell lung cancer. *Nat Commun* (2015).
- 18. Hockman *et al.* Evolution of the hypoxia-sensitive cells involved in amniote respiratory
reflexes. *eLife* (2017).
- 19. Kuo *et al.* Formation of a Neurosensory Organ by Epithelial Cell Slithering. *Cell* (2015).
- 20. Brouns *et al.* The pulmonary neuroepithelial body microenvironment represents an
underestimated multimodal component in airway sensory pathways. *The Anatomical*
*Record* (2023).
- 21. Rudin *et al.* Molecular subtypes of small cell lung cancer: a synthesis of human and
mouse model data. *Nature Reviews Cancer* (2019).
- 22. Gay *et al.* Patterns of transcription factor programs and immune pathway activation
define four major subtypes of SCLC with distinct therapeutic vulnerabilities. *Cancer Cell*
(2021).
- 23. Ng *et al.* Molecular and Pathologic Characterization of YAP1-Expressing Small Cell Lung
Cancer Cell Lines Leads to Reclassification as SMARCA4-Deficient Malignancies. *Clin*
*Cancer Res* (2024).

- 24. Pan *et al.* Insights and opportunities at the crossroads of cancer and neuroscience.
*Nature cell biology* (2022).
- 25. Venkataramani *et al.* Disconnecting multicellular networks in brain tumours. *Nature*
*reviews. Cancer* (2022).
- 26. Keough *et al.* Neural Signaling in Cancer. *Annual review of neuroscience* (2022).
- 27. Zahalka *et al.* Nerves in cancer. *Nature reviews. Cancer* (2020).
- 28. Venkatesh *et al.* Neuronal Activity Promotes Glioma Growth through Neuroligin-3
Secretion. *Cell* (2015).
- 29. Zeng *et al.* Synaptic proximity enables NMDAR signalling to promote brain metastasis.
*Nature* (2019).
- 30. Venkataramani *et al.* Glutamatergic synaptic input to glioma cells drives brain tumour
progression. *Nature* (2019).
- 31. Venkatesh *et al.* Electrical and synaptic integration of glioma into neural circuits. *Nature*
(2019).
- 32. Osswald *et al.* Brain tumour cells interconnect to a functional and resistant network.
*Nature* (2015).
- 33. Venkataramani *et al.* Glioblastoma hijacks neuronal mechanisms for brain invasion. *Cell*
(2022).
- 34. Kamiya *et al.* Genetic manipulation of autonomic nerve fiber innervation and activity
and its effect on breast cancer progression. *Nature neuroscience* (2019).
- 35. Magnon *et al.* Autonomic nerve development contributes to prostate cancer
progression. *Science* (2013).
- 36. Lee *et al.* A multiplexed in vivo approach to identify driver genes in small cell lung
cancer. *Cell Rep* (2023).

37. Woo *et al.* Neuronal metabotropic glutamate receptor 8 protects against
neurodegeneration in CNS inflammation. *Journal of Experimental Medicine* (2021).

38. Rudin *et al.* Small-cell lung cancer. *Nature Reviews Disease Primers* (2021).

39. Poirier *et al.* New Approaches to SCLC Therapy: From the Laboratory to the Clinic.
*Journal of Thoracic Oncology* (2020).

40. Dingemans *et al.* Small-cell lung cancer: ESMO Clinical Practice Guidelines for diagnosis,
treatment and follow-up☆. *Annals of Oncology* (2021).

41. Zhang *et al.* Small cell lung cancer tumors and preclinical models display heterogeneity
of neuroendocrine phenotypes. *Transl Lung Cancer Res* (2018).

42. Rad *et al.* PiggyBac transposon mutagenesis: A tool for cancer gene discovery in mice.
*Science* (2010).

43. Ashburner *et al.* Gene ontology: Tool for the unification of biology. *Nature Genetics*
(2000).

44. Carbon *et al.* The Gene Ontology resource: Enriching a GOld mine. *Nucleic Acids*
*Research* (2021).

45. Wickersham *et al.* Monosynaptic Restriction of Transsynaptic Tracing from Single,
Genetically Targeted Neurons. *Neuron* (2007).

46. Brouns *et al.* Neurochemical pattern of the complex innervation of neuroepithelial
bodies in mouse lungs. *Histochem Cell Biol* (2009).

University Hospital of Cologne | Translational Genomics
Dr. Filippo Beleggia Weyertal 115-b, 50931 Cologne

To:

Barbara Marte, PhD
Senior Editor, Nature
The Campus, 4 Crinan Street N1 9XW – London
UNITED KINGDOM

Dr. nat. med.

Filippo Beleggia
Junior Group Leader
Resident physician

University Hospital of Cologne
Department of
Translational Genomics

Department I
of Internal Medicine

Center for Integrated Oncology

Mildred Scheel
School of Oncology

Weyertal 115-b
50931, Köln

Tel.: +49 221 478 96703

Fax: +49 221 478 97719

filippo.beleggia@uk-koeln.de

Revision plan for manuscript #: 2022-12-20232B

Dear Barbara, dear members of the editorial team,

Thank you so much for discussing a possible second round of revision with us on Tuesday. We really appreciate your guidance.

Our manuscript makes two fundamental contributions to the field of cancer neuroscience, along with several ancillary findings that have all been positively evaluated by the reviewers:

1. We provide the first report of synapses forming between neurons and extracranial tumors, specifically in small cell lung cancer (SCLC). Following our first extensive revision, this evidence has now been very positively received by all reviewers.
2. We demonstrate that these synapses are functional and generate post-synaptic currents. Two reviewers are convinced by these findings, and we believe we can readily address the remaining concerns of reviewer #3 based on well-established neurobiological principles. Our electrophysiological data align perfectly with the current literature regarding GABA action reversal based on intracellular chloride concentration¹⁻³, as well as with the data on SCLC produced by Humsa Venkatesh and colleagues in the accompanying manuscript. In hindsight, we should have discussed these experiments in more detail in the previous version of our manuscript.

Importantly, reviewer #3 has suggested specific and appropriate experiments that would resolve their concerns, and we are prepared to conduct these studies. We are confident that these additional experiments will provide conclusive support for our findings and underscore the fundamental significance of our work.

Please find below in red our plan to address the reviewers' comments.

Referees' comments:

Referee #1 (Remarks to the Author):

I would like to thank the authors for answering my questions and for addressing my concerns with a large series of additional, meaningful experiments which made the manuscript significantly stronger.

We are thankful for the positive reception of our revised manuscript.

My main remaining concern would be that the existence and/or role of synaptic communication (or other form of communication) between cancer cells and neurons in the primary tumor, the site of relevance for the genetic screening, are still undefined. The ultrastructural and electrophysiological data (and rabies tracing data) have all been performed in cultured brain neurons or in the mouse brain. The authors now add co-culture experiments with nodose ganglia, which is laudable, but this data is restricted to co-culture experiments (Fig. 5j,k). Do the authors find indications for synaptic (or other) contacts between these peripheral neurons and cancer cells? Did they try to detect neuron-cancer synapses in SCLC tumors, and/or performed electrophysiological experiments in these conditions? They should at least discuss these points/limitations.

We fully acknowledge that we do not provide direct evidence for the existence of synaptic contacts in the lung or in the co-cultures with nodose ganglia. As requested by the reviewer, we will be happy to discuss this limitation and refer to future studies on this matter.

Other points: can you provide the exact statistics for Fig. 4c and f?

We will provide the exact statistics.

How do the authors know that the cell depicted in the EM image is a cancer cell? Can they e.g. provide clear evidence from correlative light/electron microscopy? Can they provide quantifications of spatial parameters and number of synapses detected (how many per cancer cell / region)?

The cancer cells in figure 4 were identified based on unique morphological and structural hallmarks that distinguish them from neurons, while the synapse in figure S11 was identified

via correlative light/electron microscopy (CLEM). We will be happy to provide additional information on this point. We will also provide a quantification of the number of contacts identified via EM, even though in our opinion this technique is not well-suited for quantification, due to the extreme sub-sampling of the cellular surface in ultrathin sections. We have already included quantifications of synaptic contacts with techniques that we consider more reliable for this purpose: superresolution (Figure 3a-f) and functional experiments (electrophysiology and rabies tracing, Figure 4).

Referee #2 (Remarks to the Author):

Reviewer #2:

Overall, the authors have been thoroughly responsive to prior critique and made substantial improvements to the manuscript including:

- an improved organization and flow;
- clarified and more precise language that improves the accuracy of the claims;
- Related to Figure 5 and Ext Data Fig 14, they provide a more comprehensive assessment of SCLC subtype in neuronal co-culture including comparison to high density conditions and fibroblasts, non-neuronal cancer types, and examining conditioned media vs direct contact. These data suggest direct contact with neurons facilitates SCLC growth more than cell density or fibroblasts or paracrine factors alone. The figure legends still lack key details, which should be added.
- Related to Figure 1 and Extended Data Fig 5, they provide new data validating a gene from the Sleeping Beauty genetic screen (Reelin), suggesting Reelin is a tumor suppressor in SCLC, but animal numbers are lacking from this analysis, so it's hard to assess rigor.
- There is an appropriate de-emphasis on Grm8. The Grm8 data is somewhat preliminary (lack of functional genetic tests), and given the lack of effects of the GRM8 agonist in vivo in two mouse models, it is appropriate to de-emphasize any claims on GRM8.
- Related to Figure 6, the in vivo drug studies are now more rigorous, and the effects of riluzole (inhibitor of glutamate release) are now tested in RP mice from both CMV and neuroendocrine cells of origin with large animal numbers. These results show a modest but consistent and significant impact on tumor burden and overall survival. While it is not clear what this drug is doing, the results are overall consistent with the notion that neuronal interactions support SCLC growth.

We are thankful for the positive evaluation of our revised manuscript

I find minor errors/omissions that can be easily corrected:

1. New Extended Data 5c-g is missing animal numbers analyzed, so there's a lack of rigor without those details.

In figure 5c,d,e only one representative mouse per experiment is shown. The number of mice is visible in figure 5f, where each dot is an independent mouse (7, 7 and 5) and the same mice are analyzed in figure 5g. We will include this information in the figure caption.

2. Extended Data Figure 20a appears to denote tumor number but does not include animal number.

We will include this information

3. Figure 5 legend lacks key details such as time points (Fig 5a-d), replicates (Fig 5e), etc.

We will include the requested information

4. Figure Legend Fig 1d, two typos, should be "ChIP-seq"

We will correct the typos.

5. The authors may want to note that GRM8 was identified as an ASCL1 and NEUROD1 ChIP-seq target gene in human SCLC cell lines from Borromeo et al, Cell Rep, 2016. Moreover, analysis of data from Olsen et al, Genes Dev, 2021, suggest that Grm8 expression is significantly lost upon Ascl1 deletion in RPM tumors, consistent with being an ASCL1 target gene. Analysis in NIH SCLCCellMiner also suggests a correlation with ASCL1 and GRM8 in human SCLC cell line datasets.

We will include this very helpful information, which fully supports our hypotheses, in the manuscript.

6. SCLC-P is usually much less neuroendocrine/neuronal but still responds quite well to neurons in culture (Fig 5f) in some cases, so cells may not necessarily be neuronal in phenotype to benefit from neuronal interactions.

We agree with this point and we will mention it in the manuscript.

Referee #3 (Remarks to the Author):

The authors have made a substantial effort to address the comments made in the review. In particular, the EM and light microscopy analysis is substantially improved and now provides good evidence for synapse-like structures between neurons and SCLC tumor cells. On the other hand, some of the newly added electrophysiology experiments are raising new questions about the validity of the working hypothesis.

We are thankful that the reviewer is now convinced by our imaging data of the presence of synapse-like structures. We believe that the concerns of the reviewer on our electrophysiology data can easily be addressed with the experiments we propose below.

Specific Points:

Extended Data Figure 10: the new data on AMPA- and NMDA-receptor expression is not adding much value. From the staining presented, it is unclear whether this is specific immune-reactivity or some unspecific labeling. Examining these receptors would make sense when probing their concentration at neuron-tumor cell contacts. However, this was not done here. I suggest extending or removing this data.

We will remove these data as requested

The new EM analysis is a valuable extension of the previous dataset. The authors refer to “docked or fusing vesicles” in the text. I strongly recommend removing such statements. The experiments were done with chemical fixation. Over the past decade, it has become clear that this method produces substantial artifacts with respect to the positioning of vesicles in the presynaptic terminal. Only high-pressure freezing allows for evaluation of vesicle docking. This does not detract from the value added regarding synaptic ultrastructure. However, docking or fusion state cannot be evaluated.

We acknowledge this point and will remove all mentions of docking and fusing vesicles.

The new electrophysiological analysis of the co-cultures is difficult to interpret. The bath application of pharmacological inhibitors can have (at least) two effects: (1) it inhibits ion channels at the surface of the tumor cells, (2) it inhibits ion channels in the cultured neurons. The reduction of spontaneous events by all blockers (targeting glutamatergic and GABAergic postsynaptic receptors) is puzzling and raises concern. The authors need to use recording conditions that separate sIPSCs and sEPSCs. If the majority of events are glutamatergic, then one would actually expect bicucullin to increase the frequency (as it would block

GABAergic transmission in the culture and increase network activity). These experiments raise concern as to the validity of the experiment.

We acknowledge that the interpretation of our electrophysiology data requires careful consideration of the complex interactions in our co-culture system and we apologize for not having discussed our observations in more detail. The reviewer raises an important point about the dual effects of bath-applied pharmacological inhibitors on both tumor cells and cultured neurons.

The apparent discrepancy between our findings and the effects expected by the Reviewer (that is, an increased frequency of glutamatergic inputs upon bicuculline addition) can be explained by the model system in our experiments (neuronal cultures at DIV12-14). The key consideration is that the effect of GABA on post-synaptic cells (hyperpolarizing vs. depolarizing) depends on their intracellular chloride concentration ¹. Young neurons undergo a switch in the response to GABA, which is initially depolarizing at immature stages and becomes hyperpolarizing upon maturation, as the intracellular chloride concentration decreases ¹. In cultured murine embryonic neurons, the switch has been reported to happen, e.g., around DIV 13-14 ³ or after two weeks ². The cultures we use for electrophysiology (DIV 12-14) are proficient for synaptic and network activity, but they are not expected to have fully undergone the switch and to respond to GABA inhibition with increased network activity. We will provide measurements on neurons to demonstrate this point.

In the recorded cancer cells, the precise nature of the measured inputs cannot be inferred from in our previous recordings. However, our prediction is that GABAergic inputs onto cancer cells would be depolarizing, on account of a high intracellular chloride concentration. This is also clearly shown in the accompanying manuscript by Humsa Venkatesh and colleagues, and would explain why we see a reduction in the overall number of inputs upon bath application of bicuculline.

We completely agree that further experiments will help to support our initial findings showing that the identified synapses are both GABAergic and glutamatergic. Therefore, as requested, we will perform additional electrophysiological recordings with ion concentrations in the internal and external solutions adjusted, to isolate GABAergic and glutamatergic currents by reversal potential in addition to pharmacological blockage. Specifically, we will employ a low chloride, Cs-based internal solution, which allows us to record sIPSCs as outward currents at the reversal potential of sodium ($V_h = 0$ mV). The GABAergic nature of the detected sIPSCs will be confirmed by bath application of GABA-A receptor antagonist. Similarly, to isolate sEPSCs, we will voltage-clamp the cells at -70 mV and confirm the glutamatergic nature of the remaining currents through bath application of AMPA and NMDA receptor antagonists. This two-pronged approach will allow us to define the neurotransmitter systems underlying the identified synaptic communication.

Figure 4d (optogenetic stimulation) seems anecdotal. A single trace is shown, there is no information on replication, success rate of evoked responses or fraction of responsive cells.

We acknowledge that this experiment was preliminary and we can provide additional traces and a quantification.

Figure 5a-d: The only modest reduction of cancer cell number and EdU+ cells after TTX addition is not strongly supportive of synaptic transmission as a major component of neuron-derived growth stimulation. TTX does not block mEPSCs. Would glutamate receptor blockers have a more substantial effect in this assay? At present, these experiments rather suggest a non-synaptic neuronal contribution to tumor cell proliferation.

In our opinion, a reduction of 30-40% in cancer cell proliferation upon TTX addition alone (Fig. 5c-d) is quite remarkable. We fully agree that other forms of neuron-cancer communication also stimulate cancer cell proliferation. The incomplete negation of the proliferation stimulus with TTX is in line with our live cell imaging data, which show that conditioned media alone can also influence proliferation (Fig. 5f). We will rephrase the text to better reflect this point. We nonetheless fully agree that quantification of proliferation in the presence of glutamatergic blockers will prove informative and we will perform this additional analysis as requested.

References

1. Peerboom *et al.* The postnatal GABA shift: A developmental perspective. *Neuroscience & Biobehavioral Reviews* (2021).
2. Opitz *et al.* Spontaneous Development of Synchronous Oscillatory Activity During Maturation of Cortical Networks In Vitro. *Journal of Neurophysiology* (2002).
3. Khirug *et al.* Distinct properties of functional KCC2 expression in immature mouse hippocampal neurons in culture and in acute slices. *European Journal of Neuroscience* (2005).

Yours sincerely,

Universitätsklinikum Köln (AöR)
Translationale Genomik
Dr. Filippo Beleggia
Weyertal 115-b, 50931 Köln

Filippo Beleggia

Referees' comments:

Referee #1 (Remarks to the Author):

I would like to thank the authors for answering my questions and for addressing my concerns with a large series of additional, meaningful experiments which made the manuscript significantly stronger.

We are thankful for the positive reception of our additional data and for the helpful expert guidance during the first revision, which we believe greatly helped to strengthen our results.

My main remaining concern would be that the existence and/or role of synaptic communication (or other form of communication) between cancer cells and neurons in the primary tumor, the site of relevance for the genetic screening, are still undefined. The ultrastructural and electrophysiological data (and rabies tracing data) have all been performed in cultured brain neurons or in the mouse brain. The authors now add co-culture experiments with nodose ganglia, which is laudable, but this data is restricted to co-culture experiments (Fig. 5j,k). Do the authors find indications for synaptic (or other) contacts between these peripheral neurons and cancer cells? Did they try to detect neuron-cancer synapses in SCLC tumors, and/or performed electrophysiological experiments in these conditions? They should at least discuss these points/limitations.

We thank the reviewer for raising this important point, which we addressed with two independent lines of experiments, performed by multicolor super-resolution imaging.

First, we examined co-cultures of human SCLC cells with nodose ganglia. As shown in **revised Extended Data Fig. 11b-e**, 62 % of the total presynaptic HOMER1 signals were localized within the mNeonGreen-expressing SCLC cells. Strikingly, these signals co-localized very significantly with postsynaptic VGlut1 signals (Pearson correlation test, $p = 0.5 * 10^{-102}$), indicating the presence of putative synaptic contacts on the cancer cells.

Second, we analyzed lung sections from tumor-bearing autochthonous mice from the *Rb1^{fl/fl}; Trp53^{fl/fl}; R26^{Cas9-IRES-EGFP}* model of SCLC. Here, cancer cells were marked by the expression of EGFP after Cre recombination and contained the vast majority of the HOMER1 signals, as would be expected in the lung, which is highly innervated but poor in neuronal somata and dendrites. Also in this setting, HOMER1 signals co-localize very significantly with VGlut1 signals (Pearson correlation test, $p = 0.8 * 10^{-43}$, **revised Extended Data Fig. 12b-e**), again suggesting the presence of putative synapses.

These new data further expand the experimental settings in which we identify putative neuron-to-SCLC synapses to the primary site and to sensory vagal neurons. The relevant results section now reads:

"To investigate the nature of these contacts, we performed confocal and stimulated emission depletion (STED) microscopy of SCLC cells in five distinct experimental settings. First, in co-cultures of SCLC cells and cortical neurons, immunostainings for glutamatergic vesicles (anti-VGlut1) and the postsynaptic protein HOMER1 revealed colocalizing formations at the contacts between neurons and cancer cells (Fig. 3a, b). Second, we identify similar contacts in co-cultures with human iPSC-derived cortical

neurons, which were characterized by the expression of the presynaptic protein bassoon (BSN) in neuronal fibers and HOMER1 in cancer cells (**Extended Data Fig. 11a**). Third, in co-cultures with murine nodose ganglia, which physiologically innervate the PNECs in the lung and are the most likely origin of the VGluT1-positive fibers observed in tumors in vivo (**Fig. 2a, d-f**)¹³, we again identify juxtaposition of HOMER1 and VGluT1 on cancer cells (**Extended Data Fig. 11b-e**). Fourth, we detected putative HOMER1-positive postsynaptic structures within cancer cells in close proximity to GFP-positive axonal boutons in brain allografts (**Extended Data Fig. 12a**). Fifth, we detected HOMER1-VGluT1 proximity at the interface of Cre-exposed, recombined EGFP-positive cancer cells in lung sections from autochthonous RP mice (**Extended Data Fig. 12b-e**).”

As suggested, we also discuss that the exact nature and functionality of these putative synaptic contacts in the lung is still to be determined. The relevant section in the discussion now reads:

“Here, we also detect colocalization of presynaptic VGluT1 and postsynaptic HOMER1 in cancer cells, suggesting that synapses may also form in this primary setting. The precise nature and functionality of these contacts in the lung, as well as a potential role in cancer initiation, remain to be determined.”

Other points: can you provide the exact statistics for Fig. 4c and f?

We include the exact statistics for previous figure 4c and 4f (**Extended Data Fig. 14c, e** in this version) in **Reviewer Tables 1 and 2** below. The p-values were calculated using the python package scipy v1.13.1, with the function stats.mannwhitneyu(alternative="two-sided"). FDR correction was performed with the Benjamini Hochberg method using the python package statsmodels v0.14.2 and the function stats.multitest(method='fdr_bh'). We note that the equality between the lowest p-value and its corresponding q-value, as well as the occurrence of repeated q-values, are consistent with the expected behavior of the Benjamini-Hochberg procedure due to its rank scaling and monotonicity adjustments.

Culturing conditions	P-value versus untreated co-culture (Mann-Whitney test)	Q-value versus untreated co-culture (FDR)
Mono-culture	0.005613251564512701,	0.01282026428251593
TTX	0.007539095118451109,	0.01282026428251593
CNQX	0.002352139817269465,	0.008232489360443128
D-AP5	0.00014924716195009947,	0.0010447301336506964
CNQX D-AP5	0.009157331630368522,	0.01282026428251593
Riluzole	0.015299702573694576,	0.017849653002643674
Bicuculline	0.05779012793801531	0.05779012793801531

Reviewer Table 1 – Statistics for previous Fig 4c / revised Extended Data Fig. 14c

Culturing conditions	P-value versus untreated slices (Mann-Whitney test)	Q-value versus untreated slices (FDR)
TTX	0.14807326904280482	0.29614653808560965
CNQX	0.4248611904415801	0.4248611904415801
D-AP5	0.14662053964149438	0.29614653808560965
CNQX D-AP5 bicuculline	0.3210924825999877	0.4248611904415801

Reviewer Table 2 – Statistics for previous Fig 4f / revised Extended Data Fig. 14e

How do the authors know that the cell depicted in the EM image is a cancer cell? Can they e.g. provide clear evidence from correlative light/electron microscopy? Can they provide quantifications of spatial parameters and number of synapses detected (how many per cancer cell / region)?

The EM images previously shown in Figure 3 were obtained from samples in which cancer cells had been treated with retroviruses expressing DsRed prior to the allograft. Given the partial efficiency of this approach in transducing cancer cells for fluorescence expression, we also relied on morphological and structural hallmarks to unequivocally distinguish them from neurons (small size, location at the borders of the tumor mass, high nuclear/cytoplasm ratio, irregular, indented nuclear membranes). Wherever feasible, we matched these hallmarks with fluorescence in the cancer cell via correlative light/electron microscopy (CLEM) (as shown in the **previous Extended Data Fig. 11a**, now **revised Extended Data Fig. 13a**). However, for some examples (like the one shown in **previous Fig. 3**) we lacked sufficiently high levels of fluorescence to conduct CLEM. As requested — and to rule out potential ambiguities — we have now replaced the prior example in Figure 3 with new experiments using cancer cells stably expressing tdTomato, allowing CLEM for virtually all examined cells (**revised Fig. 3i**).

As requested, we have also examined the percentage of cancer cells (tdTomato+) exhibiting synaptic contact sites with apposed axonal boutons in these new samples. As innervation occurs preferentially at the periphery of the tumor mass, we quantified the presence of direct contacts in a total of 280 cell perimeters obtained from allografts in 3 mice (90-96 tdTomato+ cells per mouse, examined in 10-14 consecutive ultrathin sections per mouse by EM). We found that, on average, 8.2 % of cancer cell perimeters are contacted by a synapse (**revised Extended Data Fig. 13c**). Importantly, each individual putative contact identified by 2D EM was validated via acquisition of an electron tomogram.

We now also provide the CLEM images of the reconstructed example obtained in co-cultures (**revised Extended Data Fig. 13d**).

The relevant paragraph in the results section now reads:

“To further characterize these putative synaptic contacts at the ultrastructural level, we next deployed Electron Microscopy (EM) and Correlative Light Electron Microscopy

(CLEM) in brain allografts and co-cultures. Electron tomograms and 3D reconstructions of DsRed- or tdTomato- positive SCLC cells confirmed the presence of synaptic boutons filled with vesicles contacting the plasma membrane of cancer cells (Fig 3i, Extended Data Fig. 13a, b, d). Detailed examination of 280 cell perimeters located at the periphery of the allografts in ultrathin sections revealed that an average of 8.2% of the cancer cells exhibited synapses with axonal boutons (Extended Data Fig. 13c). Importantly, additional ultrastructural hallmarks of stereotypical synapses were identified, including the presence of a synaptic cleft and a pool of vesicles close to the presynaptic membrane (Fig. 3i, Extended Data Fig. 13a, b, d). These data demonstrate that neuron-SCLC cell contacts exhibit structural features of bona fide synapses.”

Referee #2 (Remarks to the Author):

Reviewer #2:

Overall, the authors have been thoroughly responsive to prior critique and made substantial improvements to the manuscript including:

- an improved organization and flow;
- clarified and more precise language that improves the accuracy of the claims;
- Related to Figure 5 and Ext Data Fig 14, they provide a more comprehensive assessment of SCLC subtype in neuronal co-culture including comparison to high density conditions and fibroblasts, non-neuronal cancer types, and examining conditioned media vs direct contact. These data suggest direct contact with neurons facilitates SCLC growth more than cell density or fibroblasts or paracrine factors alone. The figure legends still lack key details, which should be added.
- Related to Figure 1 and Extended Data Fig 5, they provide new data validating a gene from the Sleeping Beauty genetic screen (Reelin), suggesting Reelin is a tumor suppressor in SCLC, but animal numbers are lacking from this analysis, so it's hard to assess rigor.
- There is an appropriate de-emphasis on Grm8. The Grm8 data is somewhat preliminary (lack of functional genetic tests), and given the lack of effects of the GRM8 agonist in vivo in two mouse models, it is appropriate to de-emphasize any claims on GRM8.
- Related to Figure 6, the in vivo drug studies are now more rigorous, and the effects of riluzole (inhibitor of glutamate release) are now tested in RP mice from both CMV and neuroendocrine cells of origin with large animal numbers. These results show a modest but consistent and significant impact on tumor burden and overall survival. While it is not clear what this drug is doing, the results are overall consistent with the notion that neuronal interactions support SCLC growth.

We are thankful for the recognition of our effort and for the positive evaluation of our revised manuscript, as well as for the expert guidance during the first revision, which we believe greatly helped to strengthen our results.

I find minor errors/omissions that can be easily corrected:

1. New Extended Data 5c-g is missing animal numbers analyzed, so there's a lack of rigor without those details.

In **Extended Data Figure 5c-e** only one representative mouse per experiment is shown. The number of mice is visible in **Extended Data Figure 5f**, where each dot is an independent mouse (n=7, 7 and 5) and the same mice are analyzed in **Extended Data Figure 5g**. We have included this information in the figure caption, which now reads:

“c-e) Representative hematoxylin-eosin stains of individual RPR2TC mice induced with lentiviral vectors carrying a non-targeting sgRNA or sgRNAs targeting Reln. f) The mean area of tumors identified in mice induced with sgRNAs targeting Reln is significantly larger than the area of tumors induced with the non-targeting sgRNA. N = 7 mice for non-targeting sgRNA and sgReln-1, n = 5 mice for sgReln-2. g) The size of individual tumors from the mice in f is significantly greater in mice induced with sgRNAs targeting

Reln. **h)** Force-directed graph of gene ontology analysis, showing gene sets enriched in both the piggyBac dataset and the analysis of human genetic data. Most gene sets are related to synaptic and neuronal functions (light blue). **b, f, g)** Mann-Whitney test followed by FDR correction.”

2. Extended Data Figure 20a appears to denote tumor number but does not include animal number.

We have included the information about the number of mice. The legend of **previous Extended Data Figure 20a**, now **revised Extended Data Figure 23a** now reads:

“**a)** Best response of individual tumors from RP mice induced with CGRP-Cre. The mice were treated with DCPG ($n = 31$ tumors from $n = 18$ mice), riluzole ($n = 19$ tumors from $n = 13$ mice), or the relative controls ($n = 23$ tumors from $n = 17$ mice for PBS plus $n = 20$ tumors from $n = 12$ mice for riluzole vehicle). The best response is shown as percentage relative to the tumor size in the last MRI scan before therapy. The red area indicates progression ($>120\%$), the yellow area indicates stable disease ($70-120\%$) and the green area indicates response ($<70\%$). Treatment with riluzole results in a significantly improved response, mainly characterized by slower growth or stable disease. The benefit provided by DCPG is not statistically significant. Mann-Whitney test followed by FDR correction.”

3. Figure 5 legend lacks key details such as time points (Fig 5a-d), replicates (Fig 5e), etc.

We have included the requested details in the legend of **Figure 5**, which now reads:

“**a,b)** Example of DsRed-expressing human SCLC cells (COR-L88) cultured for three days in presence (**a**) or absence (**b**) of cortical neurons and following treatment with EdU to label dividing cells. Bar, $30\ \mu\text{m}$. **c)** Quantification of COR-L88 cells cultured for three days in the absence or presence of neurons and/or TTX ($n = 3$ experiments, paired t -test, error bars represent the standard deviation). **d)** Quantification of EdU-positive COR-L88 cells cultured for three days in absence or presence of neurons and/or TTX ($n = 3$ experiments, paired t -test, error bars represent the standard deviation). **e)** Representative growth curves of COR-L88 cells cultured with and without cortical neurons from one experiment ($n = 10$ wells with neurons and $n = 30$ wells in mono-culture). **f)** Quantification of live cell imaging experiments of SCLC cell lines of different subtypes (in order: COR-L88, H69, H146, H1836, DMS273, H524, H211, H526). The cancer cells are cultured with cortical neurons or with medium conditioned by cortical neurons. The growth is normalized to the growth of mono-cultures in the same plates. Each bar is the median of ≥ 20 wells across at least 4 batches of neurons. Error bars represent the median absolute deviation from the median. The p value is calculated with a Wilcoxon signed-rank test. **g)** Quantification of the growth of NSCLC cell lines (in order: H1975, HCC44, HOP62, H2291) over five days in co-culture with cortical neurons, relative to the growth of mono-cultures. Each bar is the median of ≥ 20 wells across at least 4 batches of neurons. Error bars

represent the median absolute deviation from the median. **h)** Representative image of COR-L88 cells grown for 6 days in co-culture with a nodose ganglion. Bar, 200 μm . **i)** Representative image of COR-L88 cells grown for 6 days in the absence of a nodose ganglion. Bar, 200 μm . **j)** Representative growth curves of COR-L88 cells cultured with and without nodose ganglia from one experiment ($n = 4$ wells with ganglia and $n = 16$ wells in mono-culture) **k)** Quantification of the growth of SCLC cell lines (in order: COR-L88, DMS273, H211) over five days in co-culture with nodose ganglion explants, relative to the growth of mono-cultures. Each bar is the median of ≥ 4 wells across at least 4 batches of ganglia. Error bars represent the median absolute deviation from the median.”

4. Figure Legend Fig 1d, two typos, should be “ChIP-seq”

Thank you for picking up these typos, which we have corrected. The relative section now reads:

“**d)** Scatter plot of the gene sets in **c**. On the y axis is the RB-E2F score, calculated using ChIP-seq data from the CISTROME database. A high score indicates strong ChIP-seq signal in experiments with antibodies against RB1, RBL2, E2F1, E2F2, E2F3, E2F4, or E2F5 near the promoter of the upregulated genes included in the gene set. On the x axis is the fold change in the log₂ scale of PNECs versus other lung cells in scRNA-seq data downloaded from Travaglini et al. ⁵⁷. A high fold change indicates that the upregulated genes within the gene set are also upregulated in healthy PNECs. Fisher’s exact test with FDR correction.”

5. The authors may want to note that GRM8 was identified as an ASCL1 and NEUROD1 ChIP-seq target gene in human SCLC cell lines from Borromeo et al, Cell Rep, 2016. Moreover, analysis of data from Olsen et al, Genes Dev, 2021, suggest that Grm8 expression is significantly lost upon Ascl1 deletion in RPM tumors, consistent with being an ASCL1 target gene. Analysis in NIH SCLCCellMiner also suggests a correlation with ASCL1 and GRM8 in human SCLC cell line datasets.

We have included this very helpful information, which fully supports our hypotheses, in the manuscript. The relevant section now reads:

“Prominent among them was GRM8, a gene encoding an inhibitory metabotropic glutamate receptor that has been shown to counteract glutamate signaling by negatively regulating cAMP-dependent sensitization of inositol 1,4,5-trisphosphate receptors, thereby limiting glutamate-induced calcium release from the ER ⁶⁵. GRM8 has been identified as an ASCL1 and NEUROD1 ChIP-seq target in human SCLC cell lines ⁶⁶. The expression of GRM8 correlates with the expression of ASCL1 in cell lines reported in the SCLC-CellMiner ⁶⁷ and is reduced in autochthonous SCLC mice in which Ascl1 is deleted specifically in cancer cells ⁹. In our human datasets, GRM8

*shows specific expression in SCLC and a few other tumor types (**Extended Data Fig. 19c**) and a statistically significant enrichment of both non-synonymous mutations and more severe loss of function mutations (stop, frameshift, start-loss, and canonical splice-site mutations) (**Extended Data Fig. 19d**)."*

6. SCLC-P is usually much less neuroendocrine/neuronal but still responds quite well to neurons in culture (Fig 5f) in some cases, so cells may not necessarily be neuronal in phenotype to benefit from neuronal interactions.

We agree with this point, which is relevant for the overall interpretation of our findings. We have modified the text to highlight this. The relevant paragraph now reads:

*"All tested lines derived a significant proliferation advantage when co-cultured with cortical neurons, including the SCLC-P lines, which display a less pronounced neuroendocrine transcriptional phenotype (**Fig. 5e, f, Extended Data Fig. 17a-h**)."*

Referee #3 (Remarks to the Author):

The authors have made a substantial effort to address the comments made in the review. In particular, the EM and light microscopy analysis is substantially improved and now provides good evidence for synapse-like structures between neurons and SCLC tumor cells. On the other hand, some of the newly added electrophysiology experiments are raising new questions about the validity of the working hypothesis.

We are thankful that the reviewer is now convinced by our imaging data of the presence of synapse-like structures. We are also grateful to the reviewer for providing his/her expertise to enhance our electrophysiology experiments and data. We have addressed the reviewer's concerns as detailed below and we believe that the new electrophysiology data now convincingly shows the presence of synaptic transmission between neurons and cancer cells.

Specific Points:

Extended Data Figure 10: the new data on AMPA- and NMDA-receptor expression is not adding much value. From the staining presented, it is unclear whether this is specific immune-reactivity or some unspecific labeling. Examining these receptors would make sense when probing their concentration at neuron-tumor cell contacts. However, this was not done here. I suggest extending or removing this data.

Following the reviewer's advice, we have now removed these data as requested.

The new EM analysis is a valuable extension of the previous dataset. The authors refer to "docked or fusing vesicles" in the text. I strongly recommend removing such statements. The experiments were done with chemical fixation. Over the past decade, it has become clear that this method produces substantial artifacts with respect to the positioning of vesicles in the presynaptic terminal. Only high-pressure freezing allows for evaluation of vesicle docking. This does not detract from the value added regarding synaptic ultrastructure. However, docking or fusion state cannot be evaluated.

We acknowledge this point and have now removed all mentions of docking and fusing or fused vesicles. This affects the results section, **revised Data Figure 3** and **revised Extended Data Figure 13**). The relevant paragraphs now read:

"Importantly, additional ultrastructural hallmarks of stereotypical synapses were identified, including the presence of a synaptic cleft and a pool of vesicles close to the presynaptic membrane (Fig. 3i, Extended Data Fig. 13a, b, d)."

"i) CLEM of SCLC cells (expressing tdTomato) grafted in the mouse hippocampus. The left panels depict the registered overlay between fluorescence signal and EM image. The third panel shows the electron tomogram of an identified synaptic contact."

The tomogram (single slice) illustrates a presynaptic bouton (yellow pseudocolor) filled with vesicles, contacting the tdTomato-positive cancer cell (red pseudocolor). Blue pseudocolor indicates the nucleus. The right panel shows an enlarged view of the synaptic cleft and a pool of vesicles located within 20 nm of the plasma membrane (green pseudocolor)."

*"a) CLEM of grafted murine SCLC cells stably expressing tdTomato. Left panel shows a confocal acquisition of the SCLC cells grafted in the hippocampus of a Thy1-GFP mouse. The central panel depicts the registered overlay between fluorescence signal and EM image acquired at the border of the tumor mass, where GFP-positive fibers can be identified (boxed region). On the right, the electron tomogram of an identified synaptic contact is shown. The tomogram (single slice) illustrates a presynaptic bouton (yellow pseudocolor) filled with vesicles, contacting the identified tdTomato-positive cancer cell (red pseudocolor). Vesicles located within 20 nm of the plasma membrane (PM) are depicted in green. b) 3D reconstruction of the whole electron tomogram (250 nm thick) of the synapse depicted in **Fig. 3i**, showing the cancer cell (red pseudocolor), axonal bouton (yellow pseudocolor) with vesicles (white pseudocolor), and vesicles located within 20 nm from the synaptic cleft (green pseudocolor)."*

"d) CLEM of COR-L88 SCLC cells (expressing DsRed) co-cultured with cortical neurons. The left panels depict the registered overlay between fluorescence signal and EM image. The third panel shows the electron tomogram of an identified synaptic contact. The tomogram (single slice) illustrates a presynaptic bouton (yellow pseudocolor) filled with vesicles, contacting the identified DsRed-positive cancer cell (red pseudocolor). The right panels show a 3D reconstruction of the whole electron tomogram (250 nm thick) of the synapse, depicting cancer cell (red pseudocolor), axonal bouton (yellow pseudocolor) with vesicles (white pseudocolor), and vesicles located within 20 nm from the plasma membrane (PM) (green pseudocolor)."

The new electrophysiological analysis of the co-cultures is difficult to interpret. The bath application of pharmacological inhibitors can have (at least) two effects: (1) it inhibits ion channels at the surface of the tumor cells, (2) it inhibits ion channels in the cultured neurons. The reduction of spontaneous events by all blockers (targeting glutamatergic and GABAergic postsynaptic receptors) is puzzling and raises concern. The authors need to use recording conditions that separate sIPSCs and sEPSCs. If the majority of events are glutamatergic, then one would actually expect bicucullin to increase the frequency (as it would block GABAergic transmission in the culture and increase network activity). These experiments raise concern as to the validity of the experiment.

We thank the reviewer for this important comment. We fully acknowledge that the interpretation of the electrophysiology data included in the previous version is limited by the complex interactions between the neuronal network and the cancer cells in our co-culture system, which are both affected by the bath application of the inhibitors, as expertly pointed out by the reviewer. The COR-L88 cells we used in these initial experiments appear to have a very fragile membrane that does not allow for long term recordings, especially when changing holding potentials as needed to fully address the reviewer's concerns. In the initial setup, we were therefore limited to experiments in

which different cells were measured in parallel with or without blockers. In order to perform the requested separation of sIPSCs and sEPSCs in a model that allows for the sequential blocking of the detected currents in the same cells, we identified a human SCLC cell line (H524) that has a much more stable membrane and can be efficiently patched for longer time periods at different holding potentials. As requested, this improved experimental system allowed us to isolate spontaneous post synaptic currents by measuring the currents at -70 mV, 0 mV and +40 mV membrane potential and applying the specific receptor blockers. Similar to the COR-L88 cells, H524 cells also displayed no detectable currents in mono-culture (**revised Extended Data Fig. 14g**). However, a sizable fraction of the cells (7 out of 28, from three independent co-culture experiments) presented currents when measured at +40 mV. These currents predominantly displayed kinetics consistent with NMDA receptor-mediated currents and could be inhibited by bath application of the specific NMDAR inhibitor D-AP5 (**revised Fig. 4b, c, revised Extended Data Fig. 14h**). In 2 of the patched cells, residual currents measured at +40 mV after D-AP5 application were fully inhibited by the additional application of gabazine, suggesting that GABA-A receptor-mediated currents are also present (**revised Fig. 4b, c**). GABAergic currents inhibited by gabazine were also visible in one cell at 0 mV (**revised Extended Data Fig. 14i**). The results section describing these new findings reads:

*“Similar to COR-L88, the H524 cell line (SCLC-N subtype) exhibited no sPSCs in mono-culture (**Extended Data Fig. 14f, g**). However, we detected sPCS in H524 cells cultured with cortical neurons when measuring with a holding potential of +40 mV. The majority of these currents could be inhibited with the NMDA receptor blocker D-AP5 (**Fig. 4a-c, Extended Data Fig. 14h**). In two cells, a small fraction of the currents remained after D-AP5 inhibition, presented a shape compatible with GABA-A receptor-mediated currents, and could be inhibited with the addition of the GABA-A receptor blocker gabazine (**Fig. 4b, c**). We also identified examples of synaptic events when measuring at 0 mV, a voltage at which mainly GABA-A-mediated chloride currents are observable (**Extended Data Fig. 14i**).”*

These new data are now part of **revised Fig. 4** and **revised Extended Data Figure 14** and replace the previous recordings, which we have moved to the supplement (**revised Extended Data Fig. 14a-e**). We also rephrased the text to make it clear that we are affecting both the cancer cells and the neurons with our pharmacological treatments. Finally, while we previously highlighted the decrease in frequency after treatment with bicuculline, the difference was not statistically significant (**q=0.058**). To better reflect the statistical analysis, we now clearly state the lack of a significant difference after treatment with bicuculline and refocus the text on the other inhibitors. The revised text now reads:

*“These currents were reduced when the cancer cells and neuronal networks in the co-cultures were treated with the voltage-gated sodium channel blocker TTX, with the AMPA receptor antagonist CNQX, with the NMDA receptor antagonist D-AP5, or with the glutamate release inhibitor 2-amino-6-trifluoromethoxy benzothiazole (riluzole)⁵⁹ but not with the GABA receptor inhibitor bicuculline (**Extended Data Fig. 14c**).”*

The lack of increased network activity upon bicuculline addition is a characteristic of the model system used in our experiments. The key consideration is that the effect of GABA on post-synaptic cells depends on their intracellular chloride concentration (Peerboom & Wierenga, 2021). Young neurons undergo a switch in the response to GABA, which is initially depolarizing at immature stages and becomes hyperpolarizing upon maturation, as the intracellular chloride concentration decreases (Peerboom & Wierenga, 2021). In cultured murine embryonic neurons, the switch has been reported to be initiated after approximately two weeks (Khirug et al., 2005; Opitz et al., 2002). Most of our recordings are performed at day in vitro (DIV) 12, a timepoint at which the cultures are proficient for synaptic and network activity, but are not expected to have undergone the switch and to respond to GABA inhibition with increased network activity. To confirm this point, we have performed a series of control measurements in neurons from our cultures before and after bath application of bicuculline and could not detect an increase in the overall network activity at DIV 12 (**Reviewer Figure 1a-d**). In contrast, we could detect the expected increase in network activity at DIV 16 (**Reviewer Figure 1e-h**).

Reviewer Figure 1 – network effect of bicuculline at different timepoints

Exemplary neurons were patched at -70 mV at different timepoints, relative to initial plating. **a, b**) First neuron patched before (a) or during (b) bath application of bicuculline at DIV12. No obvious increase in synaptic event frequency is observed. **c, d**) Second neuron patched as above at DIV12. **e, f**) First neuron patched as above but at DIV16. Here, the frequency of large hypersynchronous synaptic events (see event inset 1 and 2) increased after bicuculline application, with current amplitudes reaching the nano ampere scale. **g, h**) Second neuron patched as above at DIV16. An increase in event frequency is observed.

We believe that this dataset does not necessarily have to be included in the manuscript. Hence, we provide it as Reviewer Figure. Should the reviewer or the editorial team feel

that the manuscript would benefit from including these data, we would be happy to incorporate it as an additional Extended Data Figure.

The improved experimental system discussed above also allowed us to combine optogenetics stimulation with neurotransmitter receptor inhibition. Having this tool available allowed us to substantially expand our optogenetics efforts in order to exclude indirect effects of the inhibitors on the network. The evoked currents in the patched cancer cells after optogenetics stimulation of the neurons again showed the typical kinetics of NMDA currents and could be inhibited by the NMDA receptor blocker D-AP5 in all but one cell (**revised Fig. 4d-f**). We also saw one example of a cell whose evoked currents could only be partially blocked by D-AP5 and fully blocked by D-AP5 plus gabazine, again confirming that cancer cells have the ability to also receive GABAergic inputs (**revised Extended Data Fig. 14j**). In our opinion, the use of optogenetic stimulation on the neurons and the recordings in the same cell with and without neurotransmitter receptor blockers exclude potential indirect effects on the network and therefore provide solid evidence for the presence of synaptic transmission through NMDA receptors and GABA-A receptors in cancer cells. The results section detailing these new data now reads:

*“Importantly, optogenetic stimulation of co-cultured neurons expressing Channelrhodopsin-2 elicited postsynaptic events in recorded SCLC cells measured at +40 mV, which could be abolished with the NMDA-blocker D-AP5, further corroborating the existence of direct synaptic transmission (**Fig. 4d, e, f**). In one cell, we identified the presence of both evoked NMDA and evoked GABA currents, further suggesting that cancer cells in co-culture are able to form contacts with both excitatory and inhibitory neurons (**Extended Data Fig. 14j**).”*

We are extremely grateful for the thoughtful and balanced expert guidance provided by the reviewer, which clearly helped tremendously in improving the quality of our data and led us to more solidly show the presence of sPSCs on cancer cells in co-culture with neurons. Importantly, these results are fully in line with the electrophysiology recordings reported by Humsa Venkatesh, Michelle Monje, and colleagues in the accompanying manuscript, which show both glutamatergic and GABAergic currents in different SCLC models.

Figure 4d (optogenetic stimulation) seems anecdotal. A single trace is shown, there is no information on replication, success rate of evoked responses or fraction of responsive cells.

We acknowledge that our initial optogenetics results were anecdotal and as discussed in response to the previous comment, we have now substantially expanded them. We focused on measurements at +40 mV in order to detect both glutamatergic (AMPA or NMDA) and GABAergic (GABA-A) currents in combination with subsequent specific pharmacological receptor blockade. We can now show that 34% of the cells (n=13 out of 38 from three independent co-culture experiments) are connected and display

evoked currents when the neurons are stimulated optogenetically (**revised Fig. 4d-f, revised Extended Data Fig. 14j**).

Figure 5a-d: The only modest reduction of cancer cell number and EdU+ cells after TTX addition is not strongly supportive of synaptic transmission as a major component of neuron-derived growth stimulation. TTX does not block mEPSCs. Would glutamate receptor blockers have a more substantial effect in this assay? At present, these experiments rather suggest a non-synaptic neuronal contribution to tumor cell proliferation.

In our opinion, a reduction of 20-30% in cancer cell proliferation upon TTX addition (**Fig. 5c-d**) is quite remarkable and bears therapeutic potential that should be explored in this treatment-refractory disease. We fully agree that other forms of neuron-cancer communication, such as paracrine signaling or — as suggested by the reviewer — miniature currents, may also stimulate cancer cell proliferation and we would not expect full reversal based only on blockage of action potential-mediated currents. In fact, the incomplete negation of the proliferation stimulus with TTX is in line with our live cell imaging data, which show that conditioned media alone can also influence proliferation (**Fig. 5f**). Importantly, Humsa Venkatesh, Michelle Monje, and colleagues report similar partial effects of TTX in the accompanying manuscript in some of the SCLC cell lines they tested (16T, SCLC22H, H1048), and stronger abrogation of the proliferative advantage in others (H446, H69, CORL47). They also extensively test glutamate receptor and GABA receptor inhibitors and detect effects comparable to those of TTX, with partial effects in 16T and H1048 and more pronounced effects in H446 and H69. We have rephrased the relevant paragraphs in the results and discussion to better reflect the presence of action-potential-independent neuronal contributions to cancer cell proliferation:

“Importantly, this effect was significantly, but not completely, reduced when the co-cultures were treated with TTX, suggesting that the co-culture-specific proliferative advantage of SCLC cells is mediated by both neuronal activity-dependent and -independent mechanisms (Fig. 5c, d).”

“In line with a putative oncogenic role of SCLC-neurons interactions, all SCLC cell lines we tested derived a growth advantage when cultured in the presence of neurons. This advantage was at least partially dependent on direct neuronal innervation and neuronal activity. However, the advantage was not fully abolished by blocking action potentials with TTX, suggesting the presence of additional, action-potential-independent contributions to the proliferation of cancer cells. For example, we detected a small effect of media conditioned by neurons in vitro.”

Finally, please allow us to again state our heartfelt gratitude to the reviewers for their expert guidance during the revision period. Due to their fair and always constructive comments, we were able to substantially improve the quality of our data set. We have rarely experienced such a professional and goal-oriented review process and would like to thank everyone involved. We would also like to thank the editorial team for their guidance and the handling of both parallel manuscripts. We are looking forward to hearing back from you.

References

- Khirug, S., Huttu, K., Ludwig, A., Smirnov, S., Voipio, J., Rivera, C., Kaila, K., & Khiroug, L. (2005). Distinct properties of functional KCC2 expression in immature mouse hippocampal neurons in culture and in acute slices. *European Journal of Neuroscience*, *21*(4), 899–904. <https://doi.org/10.1111/j.1460-9568.2005.03886.x>
- Opitz, T., De Lima, A. D., & Voigt, T. (2002). Spontaneous Development of Synchronous Oscillatory Activity During Maturation of Cortical Networks In Vitro. *Journal of Neurophysiology*, *88*(5), 2196–2206. <https://doi.org/10.1152/jn.00316.2002>
- Peerboom, C., & Wierenga, C. J. (2021). The postnatal GABA shift: A developmental perspective. *Neuroscience & Biobehavioral Reviews*, *124*, 179–192. <https://doi.org/10.1016/j.neubiorev.2021.01.024>
- Travaglini, K. J., Nabhan, A. N., Penland, L., Sinha, R., Gillich, A., Sit, R. V., Chang, S., Conley, S. D., Mori, Y., Seita, J., Berry, G. J., Shrager, J. B., Metzger, R. J., Kuo, C. S., Neff, N., Weissman, I. L., Quake, S. R., & Krasnow, M. A. (2020). A molecular cell atlas of the human lung from single-cell RNA sequencing. *Nature*, *587*(7835), 619–625. <https://doi.org/10.1038/s41586-020-2922-4>